# Weaning drives microbiome-mediated epigenetic regulation to shape immune memory in mice

Li Yang[1], Robert C. Peery[1], Shirui Zhou[2], Xiaomin Chen[1], Leah M. Farmer[1], Fabiola Gutierrez[3], Stephanie Fowler[3], Lanjing Zhang[4,5], Julia M. Salamat[1], Karen Riggins[6], Jiejun Shi [2] ✉ & Lanlan Shen [1] ✉

During weaning, the transition to solid food diversifies the gut microbiome, triggering a programmed immune response critical for long-lasting mucosal immunity. Previous work showed that the gut microbiome mediates epigenetic development in intestinal stem cells (ISCs) during suckling, but what happens during weaning is unclear. Here, genome-wide profiling revealed that weaning-driven microbiome changes shape the DNA methylome and transcriptome of murine ISCs in an IFNγ-dependent manner. Specifically, we observe demethylation of enhancer elements essential for MHC class II genes, which results in a transcriptional memory that persists through differentiation into adulthood. IFNγ blockade, or low-dose penicillin to target Gram-positive bacteria, in early life impaired microbiome-mediated epigenetic control and mucosal immunity, and exacerbated colitis. Murine organoids primed with IFNγ showed rapid, amplified transcriptional responses upon secondary stimulations. These findings reveal that early-life events alter the gut microbiome and these changes reprogramme ISC epigenetic memory to shape mucosal immunity.

Communication among epithelial cells, immune cells and the gut microbiota establishes and maintains intestinal homeostasis, particularly in early postnatal life[1–4]. Epithelial cells form a physical barrier, whereas immune cells actively engage with the microbiota to establish tolerance to harmless antigens during maturation while retaining responsiveness to pathogenic threats. This symbiotic relationship between the microbiome and the host supports normal intestinal physiology and overall health.

The transition from milk to solid food during weaning introduces new antigens and microorganisms to the gut that trigger a controlled inflammatory response known as the weaning reaction[5]. The weaning reaction, programmed to occur during a specific window of time,

increases proinflammatory molecules such as IFNγ and TNF[5–8]. Disrupting this process, for example, with antibiotics, affects normal immune development[5,9,10]; however, little is known about the specific role that the weaning reaction plays in developing the mucosal immune system, which shapes and affects lifelong immunity.

Long-lived, Lgr5+ multipotent intestinal stem cells (ISCs) maintain gut function by continuously replenishing the intestinal epithelium every 2–6 days[11]. The inheritance of epigenetic mechanisms allows ISCs to retain transcriptional memory of both lineage decisions and past environmental exposures across successive generations. RNA-sequencing (RNA-seq) and single-cell RNA-seq studies show that the gut microbiota critically regulates the transcriptome of the

[1]USDA Children's Nutrition Research Center, Department of Pediatrics, Baylor College of Medicine, Houston, TX, USA. [2]Department of General Surgery, Shanghai Tongji Hospital, School of Life Sciences and Technology, Tongji University, Shanghai, China. [3]Gnotobiotic Facility, Molecular Virology and Microbiology, Baylor College of Medicine, Houston, TX, USA. [4]Department of Pathology, Princeton Medical Center, Plainsboro, NJ, USA. [5]Department of Chemical Biology, Earnest Mario School of Pharmacy, Rutgers University, Piscataway, NJ, USA. [6]Department of Medicine, Hematology and Oncology, Dan L. Duncan Comprehensive Cancer Center, Baylor College of Medicine, Houston, Texas, USA. ✉e-mail: shij@tongji.edu.cn; Lanlan.Shen@bcm.edu

intestinal epithelium[12,13]. We demonstrated that the gut microbiome modulates the epigenetic development of ISCs in suckling mice[14]; however, the effects of the microbiome on the ISC epigenome after weaning and into adulthood are unknown.

Here, using whole-genome bisulfite sequencing (WGBS) for single-base resolution profiling of CpG (5′-cytosine-phosphate-guanine-3′) methylation, we discovered that microbiome-induced epigenetic changes in ISCs not only promoted crypt growth and development but also coordinated immune function and the host defence mechanism. We showed that transient IFNγ stimulation via the weaning reaction resulted in epigenetic reprogramming of MHC class II (MHC-II)-associated enhancers that conferred transcriptional memory in ISCs. Exposure of mice to low-dose penicillin (LDP) early in life reduced Gram-positive bacteria with loss of epigenetic regulation in ISCs that resulted in the loss of MHC-II expression and increased disease susceptibility in adult mice. Thus, understanding the role of the gut microbiome in modulating ISC epigenetics during a critical period of postnatal development will accelerate the development of therapies to treat intestinal diseases.

## Results

### The gut microbiome modulates locus-specific epigenetic remodelling in ISCs

To assess microbiome-driven epigenetic changes in ISCs, we generated germ-free (GF) *Lgr5-GFP* reporter mice[11] from specific-pathogen-free (SPF) *Lgr5-GFP* mice (Fig. 1a). Consistent with previous reports in wild-type mice[15,16], 15-week-old adult GF animals had morphological disruptions, including reduced crypt length, fewer Ki-67⁺ proliferative cells and abnormal goblet cell distribution, particularly in the colon (Fig. 1a and Extended Data Fig. 1). Interestingly, there were similar numbers of colonic Lgr5⁺ ISCs (EpCAM⁺/GFP⁺) in GF and SPF mice (Fig. 1b), suggesting that the microbiota does not affect ISC production under steady-state conditions. There was also a similar expression of key epigenetic modifiers such as DNA methyltransferases (DNMTs) and demethylases (TETs) in ISCs from both groups (Extended Data Fig. 2a), indicating that housing conditions did not alter widespread epigenetic programmes. Unbiased WGBS of adult colonic Lgr5⁺ ISCs revealed no significant differences in global DNA methylation across promoters, gene bodies, CpG islands or transposable elements (LINE, SINE, LTR) due to housing conditions or sex (Extended Data Fig. 2b).

Although the gut microbiome did not alter global methylation patterns, we identified 1,463 differentially methylated regions (DMRs) and 821 DMR-associated genes in colonic Lgr5⁺ ISCs from SPF vs GF mice (Fig. 1c). To identify microbiome-induced epigenetic changes that propagate stably through differentiation, we compared these DMRs with those in EpCAM⁺/GFP⁻ cells, predominantly comprising differentiated intestinal epithelial cells (IECs) (see Supplementary Tables 1 and 2 for complete lists of DMRs identified in ISCs and IECs, respectively). We found that 683 DMRs were shared between ISCs and IECs, indicating that a subset of microbiome-responsive methylation marks is stably inherited during lineage progression. The remaining non-overlapping DMRs, however, suggest that not all methylation changes are maintained through differentiation and that some arise

transiently without contributing to long-term epigenetic memory. Interestingly, 51% of shared DMRs overlapped with signatures of enhancer activity as annotated by the Encyclopedia of DNA Elements (ENCODE)[17]. Further, in SPF mice, the majority of shared DMRs exhibited hypomethylation (Fig. 1d), with associated genes enriched for immune pathways and host defence responses, including the response to interferon, the innate immune response, and antigen processing and presentation (Fig. 1e). In contrast, genes associated with hypermethylation were primarily involved in developmental processes such as post-embryonic development, anterior/posterior pattern specification, Wnt signalling pathway and cell morphogenesis. Notably, in SPF mice, promoter-proximal enhancer-like regions of MHC-II genes (*Cd74*, *H2-Eb1*, *H2-Aa* and *Ciita*) and innate immune response genes (*Irf1* and *Ifi47*) were generally hypomethylated (Fig. 1f and Extended Data Fig. 2c). Similarly, host defence genes (*Plet1* and *Lpo*) showed hypomethylation at their promoter-proximal enhancers, whereas *Duoxa2* lacked enhancer signatures within its hypomethylated region (Fig. 1f and Extended Data Fig. 2c). Thus, our unbiased genome-wide methylation profiling revealed locus-specific enhancer regions in ISCs that were sensitive to microbiome influences, with methylation changes persisting through differentiation.

### Epigenetic remodelling increases epithelial MHC-II expression

As the intestinal epithelium is the primary interface with the gut microbiome, we analysed DMRs associated with immune and host defence genes. Hypomethylation in SPF vs GF mice was confined to the intestines, with no changes in the brain, kidney, liver or spleen (Extended Data Fig. 3a). Interestingly, MHC-II genes (*Cd74*, *H2-Eb1*, *H2-Aa* and *Ciita*) were hypomethylated across all gut segments, indicating systemic priming, whereas host defence genes (*Duoxa2* and *Plet1*) showed segment-specific hypomethylation in the distal small intestine and colon, reflecting regional microbial effects. Supporting our findings, littermate *Lgr5-GFP* and wild-type mice showed very consistent methylation patterns in colonic IECs ($R^2 = 0.86$; Extended Data Fig. 3b). The EpCAM⁺CD24⁺ subpopulation, which can act as non-professional antigen-presenting cells by expressing MHC-II[18,19], had nearly identical methylation patterns to those of EpCAM⁺ IECs for all analysed CpGs ($R^2 = 0.98$; Extended Data Fig. 3c).

To determine whether these DMRs regulate gene expression, we evaluated their methylation-dependent activity in an enhancer assay. We measured the effect of DNA methylation on enhancer activity by placing either in vitro methylated or unmethylated enhancer fragments before a human EF1 promoter driving expression of the Lucia reporter gene in a CpG-free vector (Extended Data Fig. 4a,b). The increased expression provided by all MHC-II-linked DMRs (*Cd74*, *H2-Eb1*, *H2-Aa* and *Ciita*) as well as by the predicted enhancer downstream of the *Plet1* promoter was abolished by CpG methylation (Extended Data Fig. 4c). The *Duoxa2* DMR lacking enhancer signatures showed no activity.

Using RNA-seq, we identified 486 differentially expressed genes between SPF and GF mice, including 222 downregulated and 264 upregulated genes. Notably, hypomethylated immune-related genes were among the top upregulated genes in SPF ISCs (Extended Data Fig. 4d). Gene enrichment analysis showed that downregulated genes were

---

**Fig. 1 | Persistent enhancer hypomethylation underlies microbiota-driven mucosal immune regulation. a**, Schematic of GF *Lgr5-GFP* mouse rederivation. Colonic crypts were shorter under GF conditions at 15 weeks of age. Scale bars, 50 µm. **b**, Flow cytometry showed a similar percentage of Lgr5⁺ ISC populations in 15-week-old SPF and GF mouse colons (*n* = 27–28). WT mice were used as negative controls for gating. Data were analysed using unpaired (two-tailed) *t*-test. **c**, Numbers of DMRs identified by WGBS. Shared DMRs persistently linked to the microbiota in ISCs and IECs were frequently located at enhancers. **d**, The majority of shared DMRs showed hypomethylation in SPF mice. **e**, GO analysis of shared DMR-associated genes revealed enrichment of distinct biological processes related to methylation gain or loss. Enrichment significance was assessed using

a hypergeometric test, and the *P* values were adjusted for multiple comparisons using the Benjamini–Hochberg method. Top 10 terms for genes with gain or loss of methylation (adjusted *P* < 0.0005) are shown. **f**, Genome browser views with ENCODE annotation showed MHC-II and host defence genes with DMRs at promoter-proximal enhancers (enhP), displaying persistent hypomethylation in SPF ISCs and IECs. **g**, Transcriptional activation of these hypomethylated genes was detected by RNA-seq in both ISCs and their differentiated progeny IECs. ISCs were pooled from multiple mice and analysed in two biological replicates, whereas IECs were analysed individually from 6 SPF and 8 GF mice. All data (**b**,**g**) are represented as mean ± s.e.m.

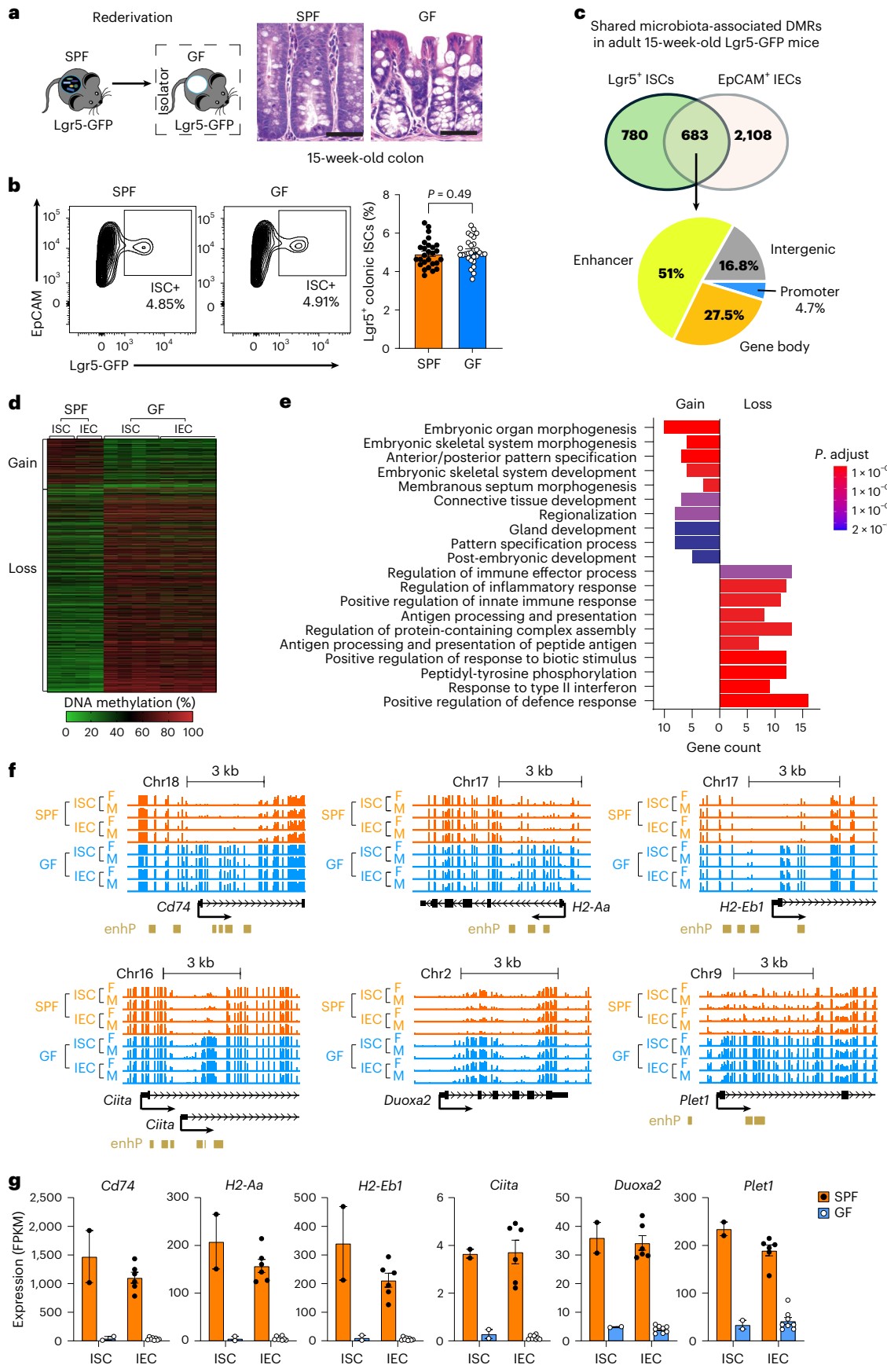

associated with developmental and metabolic processes, whereas upregulated genes were enriched for interferon response, inflammatory regulation and bacterial defence (Extended Data Fig. 4e). We also observed increased expression of MHC-II and host defence genes in SPF ISCs and IECs (Fig. 1g), which inversely correlated with DNA methylation levels (Extended Data Fig. 4f). Together, our data indicate that the gut microbiome is associated with decreased methylation at enhancer regions and increased gene expression, especially in genes involved in host defence and epithelial MHC-II signalling.

## A critical post-weaning window for microbiota-driven epigenetic reprogramming

WGBS analysis of adult-defined DMRs in SPF and GF mice at weaning (3 weeks old; Fig. 2a) revealed no methylation differences between housing conditions, with both ISCs and IECs exhibiting high methylation levels (Fig. 2b). In adult SPF mice, methylation in both cell types decreased significantly compared to weaning SPF mice, whereas adult GF mice showed no change. This suggested that the gut microbiota in SPF mice induced persistent hypomethylation after weaning. Therefore, we measured DNA methylation and gene expression in the colonic IECs of SPF mice before weaning (2 weeks), after weaning (4 and 6 weeks), in adolescence (8 and 10 weeks) and in adulthood (12 and 20 weeks) (Fig. 2c). MHC-II genes, particularly *Cd74*, were rapidly demethylated between 4 and 6 weeks of age, whereas the loss of methylation of host defence genes (for example, *Duoxa2* and *Plet1*) occurred later and more gradually. Thus, intrinsic properties of MHC-II genes may promote their early loss of methylation in both the small intestine and colon. Further, we demonstrated that the changes in methylation generally preceded or coincided with transcriptional changes (Fig. 2c).

To characterize temporal changes in microbiome-epigenetic interactions in ISCs, we performed gut microbiota transfer (GMT) by introducing the luminal contents from age-matched SPF mice into GF recipients after weaning (3–4 weeks), in adolescence (7–8 weeks) and in adults (12–13 weeks) (Fig. 2d). Metagenomic whole-genome sequencing (WGS) confirmed comparable engraftment efficiency across all post-GMT groups at the phylum level (Fig. 2e), and similar alpha diversity (abundance-based coverage estimator (ACE), Shannon, Simpson) and beta diversity (principal coordinate analysis (PCoA), non-metric multidimensional scaling (NMDS)) indicated no major differences in overall diversity at the phylum or genus level (Extended Data Fig. 5a,b). However, there were modest genus-level changes, with a greater abundance of *Bifidobacterium* in the post-weaning and adolescent groups (Extended Data Fig. 5c). Strikingly, GMT at 3–4 weeks after weaning restored epigenetic regulation of MHC-II gene transcription in ISCs most effectively at 15 weeks of age, whereas efficacy declined progressively when GMT was delayed to adolescence or adulthood (Fig. 2f). Even after extended exposure, such as in GMT-4 (defined in Fig. 2d), neither methylation patterns nor gene expressions were fully restored. Thus, the post-weaning period is a critical window for gut microbiota-induced epigenetic reprogramming in ISCs.

## Epithelial reprogramming requires IFNγ that is driven by weaning

Epithelial MHC-II expression depends on microbiota-induced IFNγ signalling[5,19–21]. During weaning, expansion and diversification of the gut microbiota induce a strong but transient IFNγ response[5,8], leading us to hypothesize that this cytokine surge mediates locus-specific hypomethylation of MHC-II regulatory regions. To test this, we administered an IFNγ-neutralizing antibody to SPF mice either immediately after weaning (3–5 weeks, Fig. 3a) or in adulthood (15–17 weeks, Fig. 3b) and assessed DNA methylation and gene expression of MHC-II genes. Strikingly, blocking IFNγ shortly after weaning, but not in adulthood, abolished epithelial MHC-II expression as measured by flow cytometry (Fig. 3a,b). Consistent with our hypothesis, blocking IFNγ after weaning disrupted microbiome-induced hypomethylation and transcriptional activation of MHC-II genes (Fig. 3c). Thus, transient weaning-associated IFNγ production is required to establish long-lasting, microbiome-dependent epigenetic regulation of epithelial MHC-II signalling.

## Gram-positive bacteria mediate epigenetic reprogramming

Because Gram-positive bacteria are potent inducers of IFNγ[22,23], we determined their role in epithelial MHC-II epigenetic programming in early life. To consistently reduce microbes in coprophagic mice, we treated pregnant dams and their offspring with LDP throughout the early postnatal period (Fig. 4a). Microbial communities expanded robustly from 2–4 weeks of age and stabilized after weaning at 4–12 weeks (Fig. 4b). WGS analysis revealed that LDP selectively reduced Gram-positive taxa without affecting overall microbial richness or expansion (Fig. 4b,c). In particular, the relative abundance of Gram-positive bacteria, including Bacillota and Actinomycetota, decreased in LDP-treated mice, whereas Gram-negative populations such as *Bacteroides* increased (Fig. 4c). Quantitative PCR confirmed the reduction in Gram-positive genera, including *Lactobacillus*, *Clostridium* and segmented filamentous bacteria (SFB) (Extended Data Fig. 6). With this decrease in Gram-positive genera, there was a corresponding reduction in CD4+ IFNγ+ T cells in the intestinal intraepithelial lymphocyte (IEL) compartment (Fig. 4d).

Indeed, LDP treatment increased methylation and reduced expression of MHC-II genes in the colonic epithelium (Fig. 4e), and flow cytometry confirmed decreased MHC-II expression on EpCAM+ IECs (Fig. 4f). Given conflicting reports on the impact of early-life antibiotic exposure on colitis[24,25], we next assessed whether early LDP exposure altered susceptibility to dextran sulfate sodium (DSS)-induced colitis. LDP-pretreated mice developed accelerated and more severe colitis upon DSS challenge, as evidenced by rapid weight loss, increased disease activity and persistent epithelial damage, whereas control mice largely recovered after initial DSS exposure (Fig. 4g–i and Extended Data Fig. 7). Moreover, LDP exposure exacerbated colitis-associated tumorigenesis in the azoxymethane (AOM)-DSS model, resulting in larger and more aggressive tumours (Extended Data Fig. 8). Together, these data indicate that the microbiota–IFNγ axis epigenetically programmes epithelial MHC-II during early life, thereby priming post-weaning ISCs to protect against intestinal inflammation and tumorigenesis.

## Microbiota-derived metabolites contribute to weaning-driven epigenetic reprogramming

To determine the role of microbiome changes driven by weaning on host epigenetic regulation, we analysed genus- and species-level WGS data.

**Fig. 2 | Post-weaning gut microbiota drives methylation dynamics and gene regulation in ISCs. a**, A timeline of the stages of postnatal development in mice. **b**, Microbiome-mediated persistent hypomethylation in adult ISCs and IECs emerged after weaning. Violin plots compare overall methylation levels of adult-defined microbiome-associated DMRs at weaning and in adulthood (ISCs, left; IECs, right). Each violin plot shows methylation distribution and density, with the center line indicating the median and dotted lines denoting the interquartile range. **c**, Comparison of methylation and gene expression changes before weaning (2 weeks old, *n* = 4) and after weaning (4–20 weeks old, *n* = 2–3 per age group). **d**, Schematic illustration of gut microbiota transfer (GMT) experiments performed in different age groups. **e**, Metagenomic whole-genome sequencing showed a similar phylum-level composition across post-GMT groups. *N* = 3–4 per group. **f**, GMT in post-weaning GF mice restored ISC epigenetic regulation. DNA methylation (bisulfite clonal sequencing) and gene expression (RT–qPCR) were quantified for the indicated genes in GF and GMT mice. Each circle represents a CpG site, with white indicating unmethylated sites and black indicating methylated sites. Expression data were analysed using unpaired (two-tailed) *t*-test and represented as mean ± s.e.m. of 3 biological replicates.

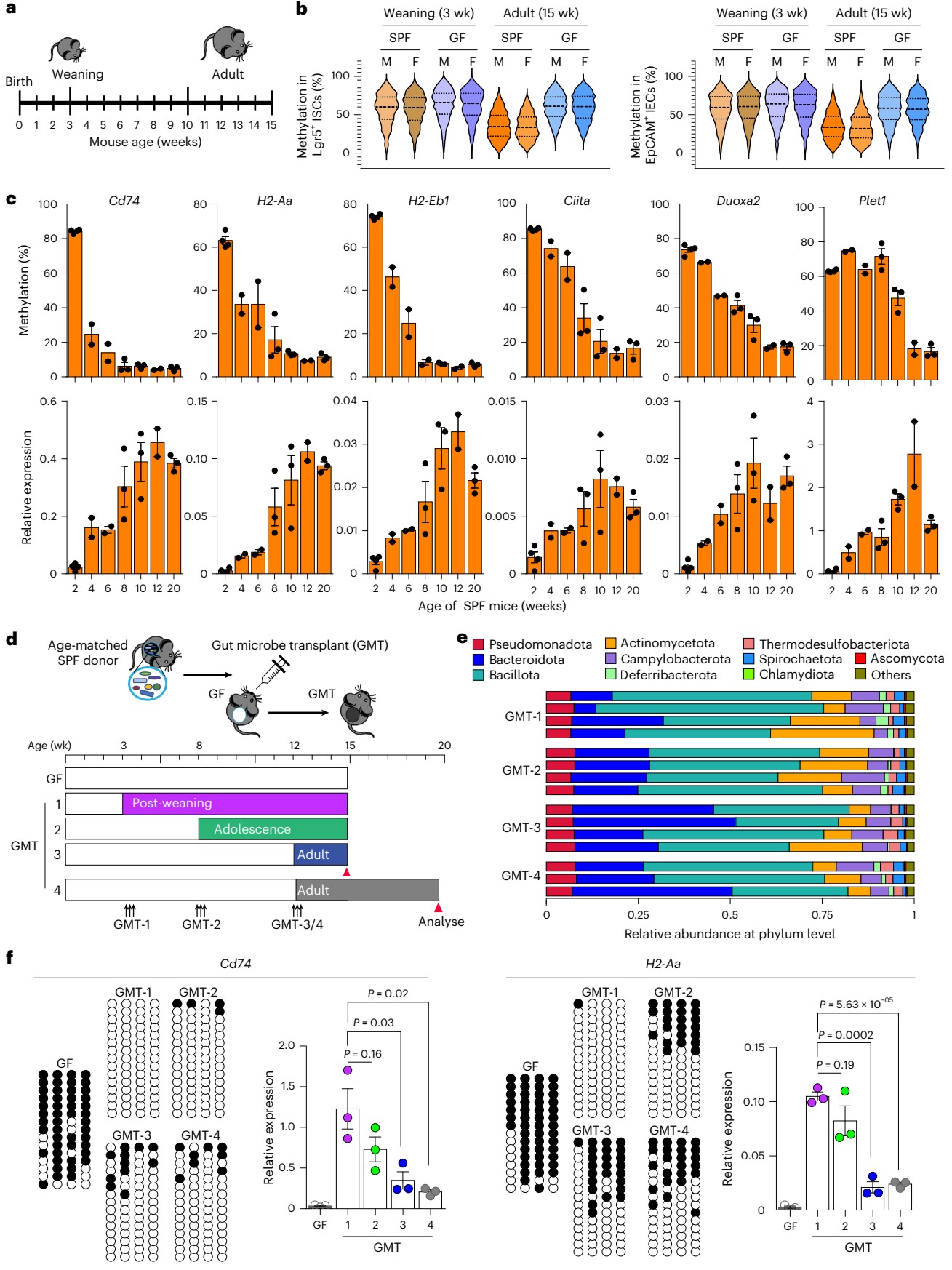

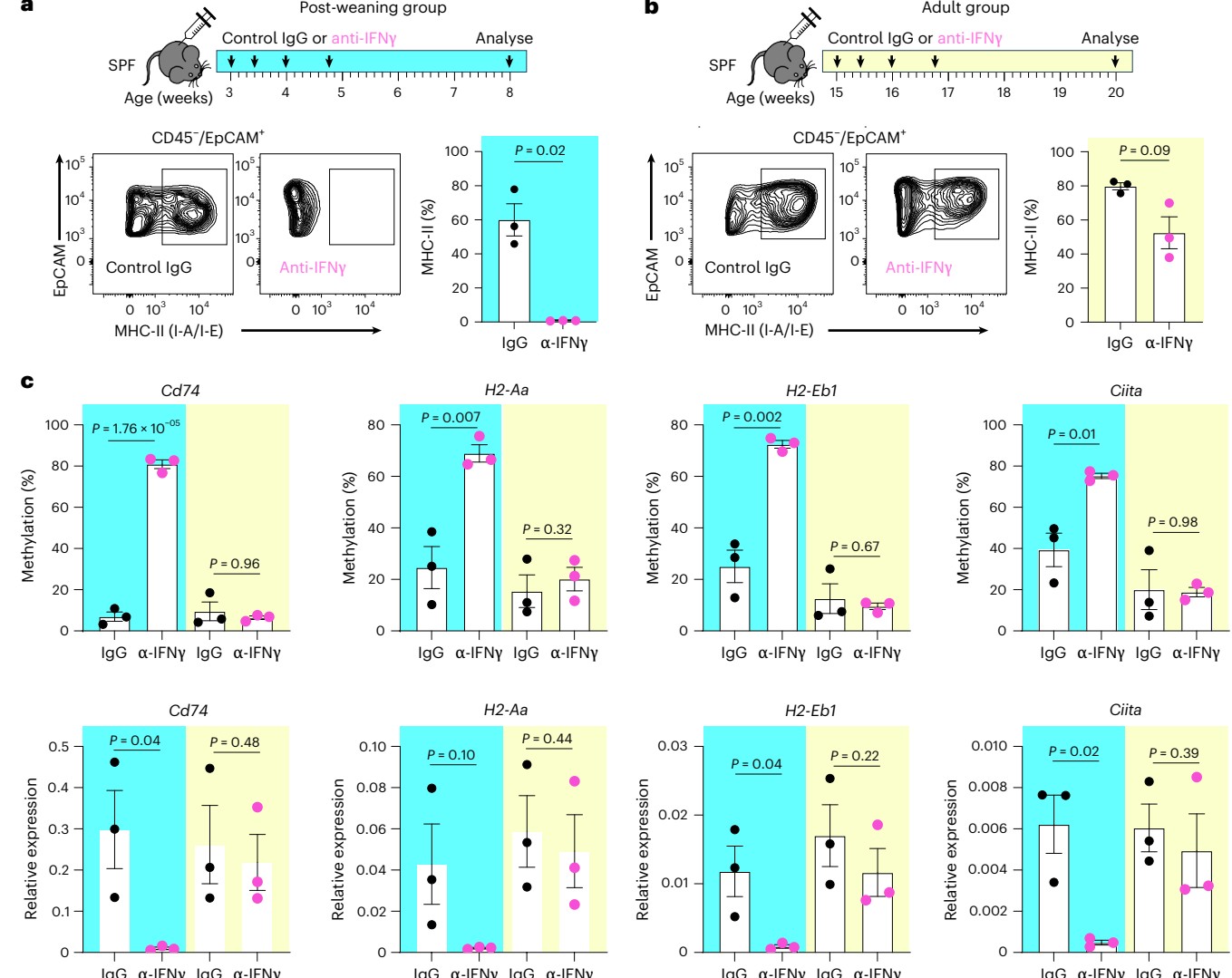

**Fig. 3 | IFNγ signalling after weaning is required for microbiome-induced epigenetic regulation of epithelial MHC-II. a**, Experimental design for post-weaning mice. Mice immediately after weaning (3 weeks of age) were treated with either control IgG or IFNγ neutralizing antibodies 4 times over the course of 2 weeks and analysed 3 weeks later. Flow analysis of MHC-II expression in colonic IECs showed that IFNγ blockade abolished epithelial MHC-II expression. **b**, Same treatment design, frequency and analysis timeline in adult mice at 15 weeks of age. In contrast to the post-weaning group, IFNγ neutralization in adult mice did not significantly alter epithelial MHC-II expression compared to controls. **c**, DNA methylation and gene expression of MHC-II genes were analysed in post-weaning (blue highlight) and adult (yellow highlight) groups, with or without anti-IFNγ antibody treatment. Blocking IFNγ shortly after weaning, but not in adulthood, disrupted microbiome-induced hypomethylation and transcriptional activation of MHC-II genes. All data (**a**–**c**) were analysed using unpaired (two-tailed) *t*-test and represented as mean ± s.e.m. of 3 biological replicates.

LDP diminished several bacterial genera, including *Bacillus*, *Faecalibaculum*, *Blautia*, *Roseburia*, *Allobaculum*, *Turicimonas*/*Turicibacter* and *Faecalibacterium*, that are associated with weaning (Fig. 5a). *Bacillus* species directly stimulate IFNγ production via CpG motifs in bacterial DNA, whereas *Turicibacter* abundance is linked to enhanced immune activation, probably mediated by short-chain fatty acid (SCFA) effects on CD4[+] T cells[26–28]. At the species level, LDP abolished the expansion of *B. catenulatum*, *B. wexlerae*, *R. intestinalis* and *A. hallii*, which normally occurs after weaning with the introduction of dietary fibre in solid foods (Fig. 5b). These microbes produce SCFAs—primarily butyrate, acetate and propionate—that act as histone deacetylase (HDAC) inhibitors to regulate host intestinal epigenetics[29,30]. Notably, LDP also reduced *Lacticaseibacillus rhamnosus* and *Clostridium sporogenes*, which enhance microbial production of α-ketoglutarate (α-KG) from amino acids[31,32], a cofactor for demethylation by host DNA demethylases (Fig. 5c). Gene Ontology (GO) analysis identified shifts in epigenetic

regulatory pathways (Fig. 5d), including isocitrate dehydrogenase activity affecting α-KG, ferrous iron transmembrane transporter activity for Fe[2+], and methionine γ-lyase activity that inhibits DNMT (Fig. 5e). Thus, SCFA- and α-KG-producing microbes induced by weaning contribute to IFNγ-mediated epigenetic reprogramming of MHC-II in ISCs, a gut process that is profoundly disrupted by LDP treatment after weaning.

**Epigenetic reprogramming as a mechanism of trained memory**

Our findings suggest that gut microbiota changes driven by weaning promote epigenetic reprogramming of epithelial MHC-II expression via (1) IFNγ signalling induced by immune crosstalk and (2) microbial metabolites that reinforce microbiome, epithelial cells and immune system interactions. To determine the role of IFNγ in epithelial-intrinsic epigenetic changes, we treated organoids derived from 6-week-old GF mice with IFNγ at a physiologically relevant concentration of 0.1 ng ml[−1], optimized on the basis of cell proliferation

and viability (Extended Data Fig. 9a). IFNγ-induced demethylation of the *Cd74* enhancer region, accompanied by increased *Cd74* expression, was detectable as early as 24 h (Fig. 6a). The rapid kinetics of methylation loss suggested that IFNγ–STAT signalling recruits active demethylases, such as TET proteins. Supporting this, chromatin immunoprecipitation (ChIP)–qPCR revealed enrichment of STAT3 and TET3 at the demethylated, but not the methylated *Cd74* enhancer (ChIP-2 and 3), and not at the non-DMR control region (ChIP-1) (Fig. 6b), whereas STAT1, TET1 and TET2 were not enriched (Extended Data Fig. 9b). To assess potential synergy with other cytokines or microbial metabolites, we treated organoids with TNF, sodium butyrate (NaB) or lipopolysaccharide (LPS); however, none altered *Cd74* methylation (Fig. 6c). Interestingly, co-treatment with IFNγ and the hypomethylating agent decitabine (DAC)[33] enhanced demethylation and further increased *Cd74* expression (Fig. 6c). Notably, DAC alone did not activate *Cd74* expression, indicating that although enhancer demethylation increased DNA accessibility, transcriptional activation required inductive signals such as IFNγ–STAT signalling.

On the basis of these results, we propose that microbiota-induced enhancer demethylation in ISCs functions as an epigenetic form of trained memory: it leaves basal gene expression unchanged but primes epithelial cells to mount faster and stronger responses to subsequent challenges. To test this, we treated organoids derived from GF mice with IFNγ for 3 passages, followed by a 7-day washout, and then restimulated them with either IFNγ or TNF (Fig. 6d). The 3 IFNγ passages removed *Cd74* enhancer methylation, confirming demethylation at the stem cell level (Fig. 6e). Compared to naïve organoids, IFNγ-primed organoids exhibited markedly elevated *Cd74* expression upon restimulation with IFNγ, peaking at 36 h in a time-course analysis (Fig. 6f). Consistent with the features of trained memory, IFNγ-primed organoids also showed rapid, amplified *Cd74* induction in response to non-specific stimulation with TNF (Fig. 6f). Therefore, the transient exposure to IFNγ established epithelial cell-intrinsic transcriptional memory through epigenetic reprogramming of MHC-II-associated gene enhancers.

To directly assess the influence of microbiota on methylation-dependent memory-like responses, we generated colon organoids from three groups of 15-week-old mice: (1) GF mice, (2) SPF mice and (3) GF mice that received GMT at weaning (Fig. 6g). Compared to GF organoids, SPF organoids exhibited significant demethylation of MHC-II genes *Cd74*, *H2-Eb1*, *H2-Aa* and *Ciita* (Fig. 6h). The methylation patterns of organoids from GMT mice were similar to the patterns in SPF organoids, confirming the direct effects of the microbiota. Following IFNγ stimulation, all organoids showed comparable proliferation and viability (Extended Data Fig. 10a). SPF organoids showed enhanced activation of all MHC-II genes vs GF organoids, detectable as early as 6 h after IFNγ stimulation (Fig. 6i and Extended Data Fig. 10b). We saw similar effects in GMT, reflecting the epigenetic restoration in MHC-II enhancers. Thus, the microbiota changes driven by weaning serve as key regulators of MHC-II-associated epigenetic remodelling in ISC enhancers and have lasting effects on epithelial-intrinsic immune and tissue responses.

## Discussion

Here we demonstrate a previously unrecognized layer of microbiota-mediated epigenetic regulation that emerges during early life

and persists beyond transient responses to microbial exposures. There are three key features of this regulatory programme. First, microbiota-induced changes are highly locus specific: they do not alter global DNA methylation patterns, but rather are specific to enhancer elements. Second, these enhancer-associated modifications regulate intestinal immunity by controlling genes essential for epithelial MHC-II presentation (*Cd74*, *H2-Aa*, *H2-Eb1* and *Ciita*). Third, in contrast to the promoter-centric paradigm of DNA methylation-mediated transcriptional regulation, microbiota-driven enhancer demethylation establishes the long-term transcriptional potential rather than immediate gene activation. Similar to trained immunity, this mechanism primes epithelial cells for rapid and enhanced MHC-II expression by subsequent, even unrelated, challenges. Together, these findings define a molecular framework for understanding how early-life environmental exposures imprint lasting epigenetic memory and durably shape lifelong mucosal immunity.

Epithelial MHC-II antigen presentation is critical for the interaction of cells with microbial-specific CD4⁺ T cells, where it promotes tolerogenic responses[18,34–36]. The gut microbiota and their metabolites, such as SCFA, regulate epithelial MHC-II expression through the epigenetic modifier histone deacetylase 3 (HDAC3)[37], and intestinal TET enzymes are implicated in microbiota-dependent DNA demethylation and protection against inflammatory bowel diseases (IBDs)[38,39]. However, little is known about the mechanisms by which (1) genome-wide regulators exert locus-specific epigenetic effects, (2) changes are integrated into stable epigenetic memory or (3) early-life microbial colonization directs this trajectory. Using complementary approaches—including high-resolution DNA methylome profiling, microbiota transplantation and ablation, and intestinal organoid modelling—we found that the stepwise methylation reprogramming of MHC-II genes in ISCs by the IFNγ–TET3 axis was induced by the weaning-associated gut microbiota, thereby imprinting cells of the epithelium. This reprogramming coincides with the 'weaning reaction', a transient burst of inflammatory signals such as IFNγ and TNF, that protect against IBD and cancer later in life[5]. Here we established a direct mechanistic link between the weaning reaction and both the ontogeny and lifelong epigenetic regulation of the epithelial MHC-II pathway.

Our findings carry important clinical implications. IBD is typically diagnosed in late adolescence or young adulthood, between the ages of 15 and 30 years[40–42]. Epidemiologically, antibiotic use is linked to the development of IBD[43–47], probably as a result of dysbiosis and genetic predisposition[48–53]. We demonstrated that early-life exposure to LDP selectively suppressed Gram-positive bacteria, disrupted epigenetic regulation in ISCs—including the establishment of post-weaning MHC-II expression, exacerbated colitis symptoms, impaired recovery and reduced long-term resilience, with an increased risk of colon cancer. Consistently, in mouse models, diet-induced and microbiota-associated loss of epithelial MHC-II expression also promoted colon cancer[54]. Thus, epithelial MHC-II epigenetic regulation is important in health and could serve as a biomarker for identifying individuals at risk for IBD or early-onset colorectal cancer.

We also identified a critical window for gut microbiome-mediated epigenetic reprogramming of ISCs. Early-life exposure to commensal

---

**Fig. 4 | Early-life exposure to LDP disrupts post-weaning epigenetic regulation of MHC-II signalling and predisposes adolescent mice to DSS-induced colitis. a**, Experimental design for LDP treatment and subsequent analyses. **b**, Rarefaction curves of taxonomic richness in faecal samples from control mice (without LDP, w/o LDP; left) or LDP-treated mice (right), collected before weaning (2 weeks) and after weaning (4–12 weeks). Each line represents a faecal WGS sample, with colours indicating age groups. **c**, Relative abundance of microbial phyla in control or LDP-treated mice before and after weaning. **d**, LDP treatment reduced IFNγ-producing CD4⁺ T cells in 8-week-old mice. Data were analysed using unpaired (two-tailed) *t*-test and represented as mean ± s.e.m. of 4 biological replicates. **e**, LDP treatment altered the epigenetic reprogramming of MHC-II

genes that normally occurs in post-weaning SPF IECs at 8 weeks of age. All data were analysed using unpaired (two-tailed) *t*-test and represented as mean ± s.e.m. of 3 biological replicates. **f**, Representative flow analysis (top) and quantified data (bottom) show that LDP treatment reduced epithelial MHC-II expression in 8-week-old mice. Data were analysed using unpaired (two-tailed) *t*-test and represented as mean ± s.e.m. of 4 biological replicates. **g**, Experimental design for repeated challenges with DSS. **h,i**, Weight loss (**h**) and disease activity index (DAI) (**i**) during repeated DSS cycles in LDP-pretreated (*n* = 5) vs control (*n* = 10) mice. Data are represented as mean ± s.e.m., and *P* values were calculated using two-way analysis of variance (ANOVA) with Geisser–Greenhouse correction.

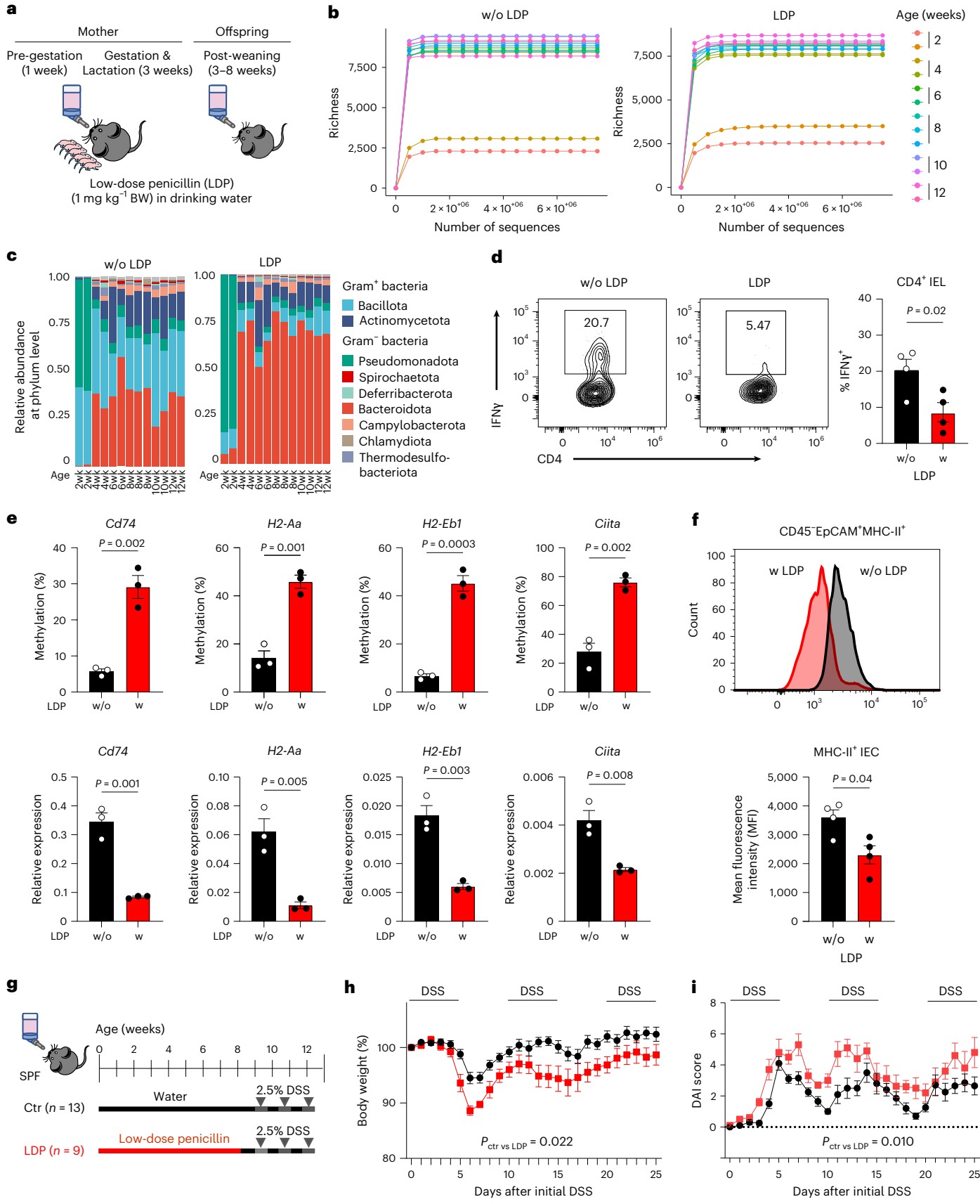

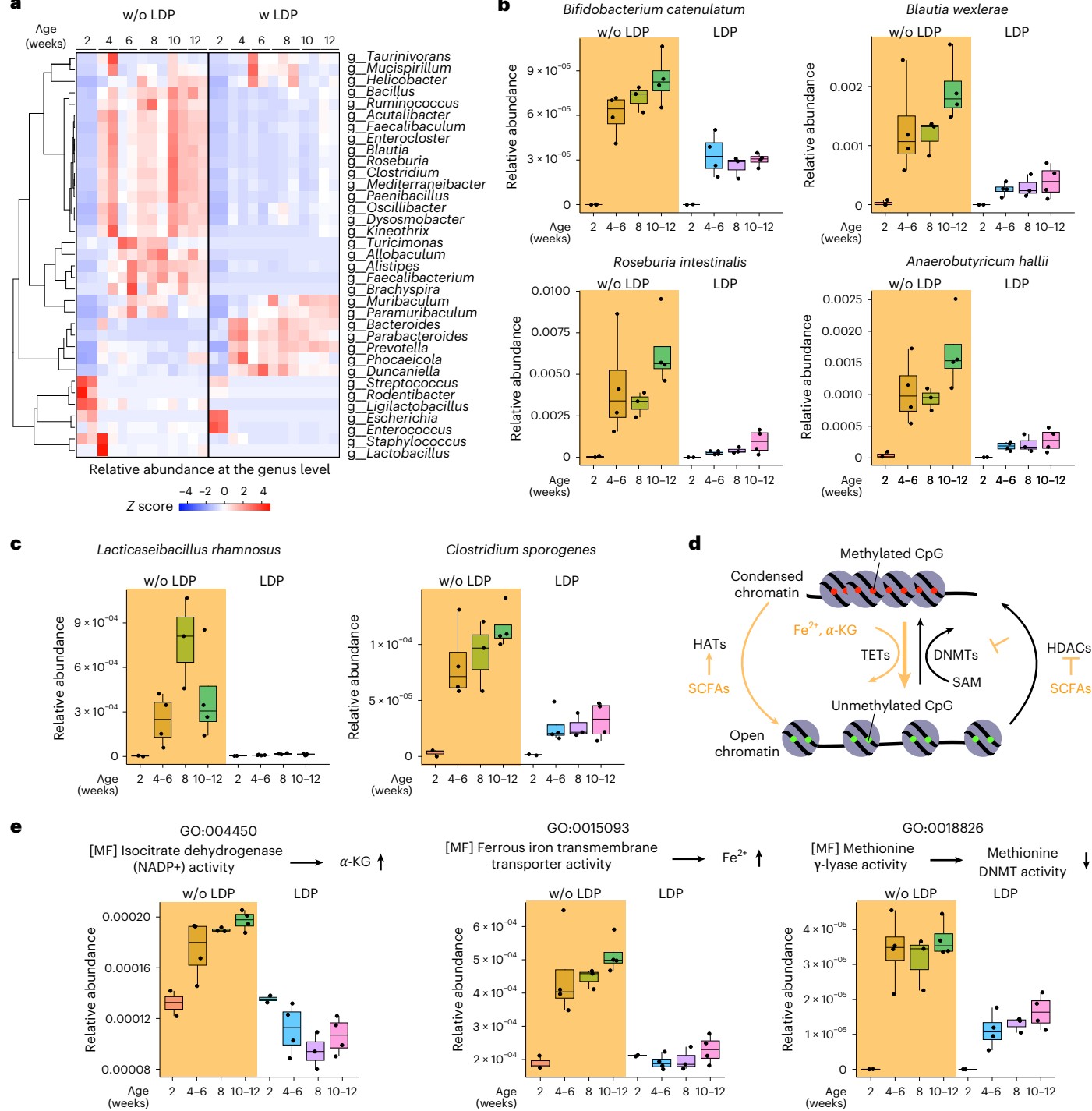

**Fig. 5 | Early-life LDP treatment alters gut microbiota composition and disrupts metabolite-mediated epigenetic reprogramming. a**, WGS at the genus level revealed that specific bacterial taxa normally colonizing the gut after weaning are depleted by early-life LDP exposure. **b**, Four SCFA-producing bacteria that typically expand after weaning were depleted in LDP-treated mice. **c**, The abundance of two bacteria that produce metabolites that directly or indirectly modulate epigenetic regulation, such as α-KG, lactate and indole-3-propionic acid (IPA), decreased following LDP treatment. **d**, Schematic model illustrating how microbial metabolites promote TET-mediated DNA demethylation and chromatin accessibility, linking post-weaning microbial diversification to host epigenetic regulation. HAT, histone acetyltransferase. **e**, GO enrichment analysis showed that activities associated with epigenetic processes normally increase after weaning but are disrupted by LDP treatment. In **a**–**c** and **e**, results are representative of 2 or 3 biological replicates. In **b**, **c** and **e**, boxplots show the median (centre line), upper and lower quartiles (boxes), and minimum to maximum values (whiskers); all individual data points are shown.

microbiota is associated with reduced susceptibility to colitis, in part through epigenetic regulation of *Cxcl16*, a chemokine ligand gene in natural killer T cells[55]. We demonstrated that changes in the microbiota induced by weaning shaped MHC-II epigenetic regulation in ISCs via IFNγ–TET3-mediated enhancer demethylation. We found that the microbiota coordinated with immune and epithelial cells to establish an epigenetic memory that conferred long-term protection against chronic intestinal disease. Importantly, neutralization of IFNγ

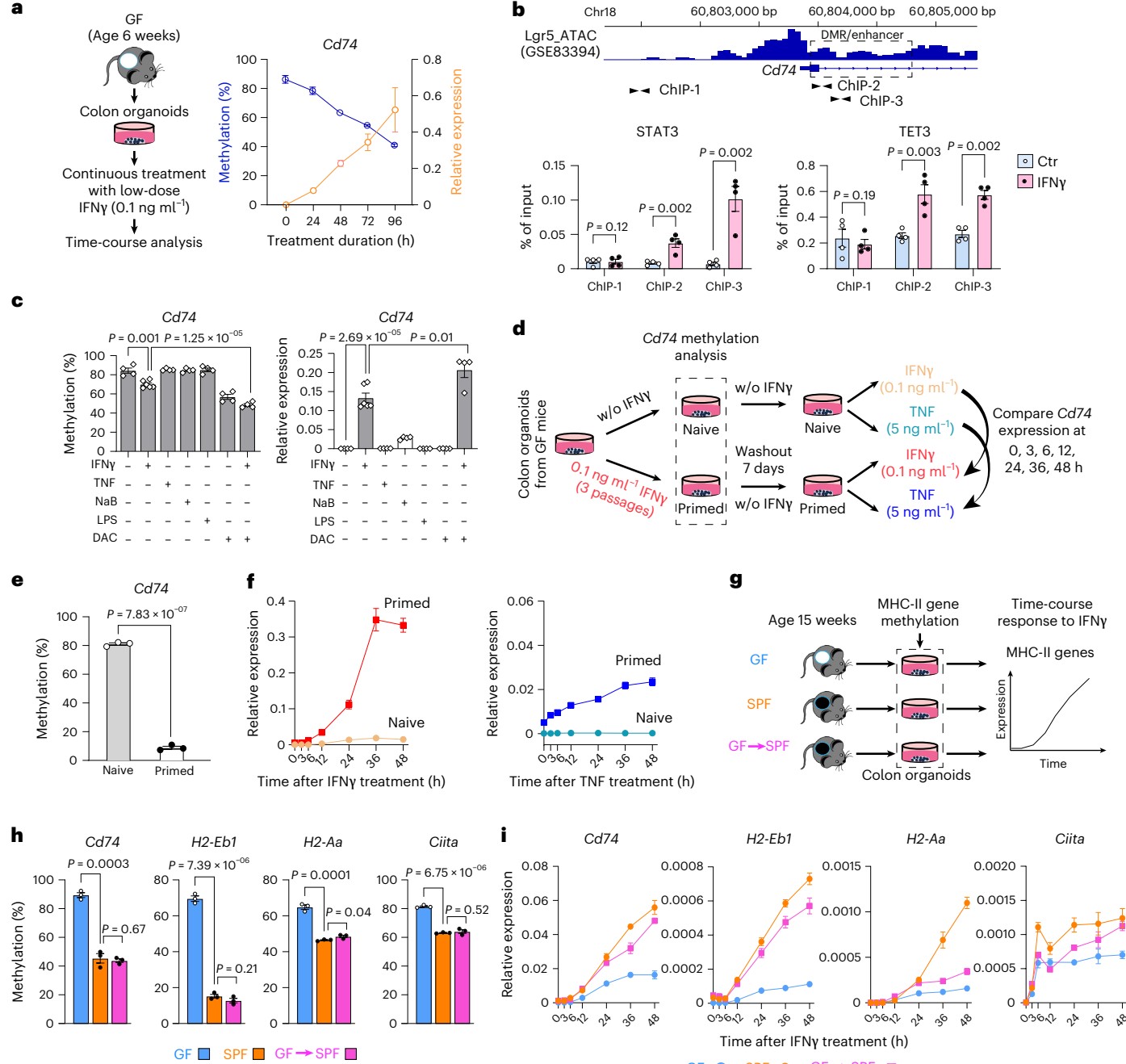

**Fig. 6 | IFNγ stimulation induces transcriptional memory in ISCs through enhancer-mediated epigenetic reprogramming of MHC-II genes. a**, Time-course analysis of *Cd74* methylation and expression in GF colonic organoids treated with a low dose of IFNγ (0.1 ng ml⁻¹) demonstrates rapid demethylation and concomitant gene activation. Data are presented as mean ± s.e.m. of 2 independent experiments. **b**, DNA demethylase TET3 mediates IFNγ-induced epigenetic reprogramming of *Cd74*. Shown is the *Cd74* locus with its genomic location, ATAC-seq peaks from sorted Lgr5-GFP⁺ ISCs (GSE83394), the microbial-induced DMR (dashed box) and the positions of primer sets used to assess STATs and TETs binding by ChIP–qPCR. Compared with a non-DMR control region (ChIP-1), STAT3 and TET3 showed an IFNγ-dependent increase in binding at the DMR/enhancer (ChIP-2 and ChIP-3). **c**, Combined treatment with IFNγ and the hypomethylation agent DAC synergistically enhanced IFNγ-induced demethylation and transcriptional activation. In contrast, treatment with TNF, sodium butyrate (NaB) or LPS alone did not alter *Cd74* methylation. **d**, Experimental design for testing transcriptional memory in organoids with

(primed) or without (naïve) previous IFNγ exposure. Pretreated organoids were rested for 7 days without IFNγ and then restimulated with IFNγ or TNF. Memory was indicated by faster and stronger induction of *Cd74* expression. **e**, Before restimulation, organoids exposed to IFNγ for 3 passages (primed) showed nearly complete loss of *Cd74* methylation. **f**, IFNγ (left) or TNF (right) restimulation of primed organoids accelerated and enhanced *Cd74* expression compared to naïve organoids. **g**, Experimental design to determine whether microbiota drive methylation-dependent transcriptional memory of epithelial MHC-II. **h**, Methylation of MHC-II genes in organoids derived from adult GF, SPF and GF→SPF (converted at weaning) mice at 15 weeks of age, with methylation assessed after passage 2. **i**, IFNγ stimulation induced methylation-dependent, memory-like transcriptional activation of the MHC-II genes *Cd74*, *H2-Eb1*, *H2-Aa* and *Ciita* in SPF and GF→SPF organoids, measured by RT–qPCR. In **b**, **c**, **e**, **f**, **h** and **i**, all data are presented as mean ± s.e.m. of at least 3 independent experiments. In **b**, **c**, **e** and **h**, *P* values were calculated using unpaired (two-tailed) *t*-test.

at weaning, but not in adulthood, greatly reduced microbial-driven demethylation of MHC-II enhancers. Similarly, gut microbiota transplantation was most effective immediately after weaning, whereas transplantation in adulthood had little effect on the epigenetic state. Interestingly, transplantation during adolescence resulted in partial epigenetic restoration, suggesting that the window is not fully closed after the weaning-induced changes and additional modulators of microbial epigenetic function probably remain to be identified. In this study, metagenomic GO- and pathway-based analyses were used to identify microbiome-associated functional signatures that align with known epigenetic regulatory processes. Future studies incorporating metabolomics, targeted manipulation of specific bacterial species and dietary modulation will be required to establish microbial causality and define the drivers of these long-lasting epigenetic programmes.

In conclusion, we demonstrate that the healthy gut microbiome, driven by dietary changes during weaning, reprogrammes ISC epigenetic states to shape enduring immune and tissue responses. These findings suggest that appropriately timed probiotic or microbial interventions could provide lifelong protection against chronic intestinal diseases.

## Methods

### SPF and GF *Lgr5-GFP* mice
We used a knock-in *Lgr5*[EGFP–IRES–CreERT2] mouse model (*Lgr5-GFP*) in which green fluorescent protein (GFP) expression is driven by endogenous *Lgr5* regulatory sequences[11]. *Lgr5-GFP* mice were backcrossed to the C57BL/6J strain for more than 20 generations to ensure a genetically identical background. Conventional *Lgr5-GFP* mice were maintained under SPF conditions in the Texas Children's Hospital animal facility. To establish a GF *Lgr5-GFP* mouse colony, SPF breeders were used to generate litters that were delivered by caesarean section into a GF isolator and fostered by GF BALB/c mothers. Faecal samples were analysed weekly for 6 months using 16S sequencing to verify GF status, and monthly thereafter. The GF colony was maintained in the Gnotobiotic Mouse Facility at Baylor College of Medicine. Mice were kept on a 12-h light/dark cycle at 20–22 °C with 30–70% humidity. All mice were fed an identical autoclaved chow diet (LabDiet 5V0F) to minimize dietary confounding. Both male and female mice were age and sex matched for all experiments. All animal procedures were conducted in accordance with the NIH Guide for the Care and Use of Laboratory Animals and were approved by the Baylor College of Medicine Institutional Animal Care and Use Committee (protocol no. AN-6775).

### Gut microbiota transfer
GF *Lgr5-GFP* mice were conventionalized by GMT at 3, 8 or 12 weeks of age. Luminal contents were collected from the colons of 2 age-matched SPF donor mice, suspended in 1 ml sterile phosphate-buffered saline (PBS) and centrifuged at low speed to remove insoluble materials. GF recipient mice were inoculated orally with 200 µl of freshly prepared microbiota suspension. Following inoculation, mice were transferred to the SPF facility and maintained under standard housing conditions. Colonic epithelial cells were collected at indicated time points for DNA methylation and expression analyses.

### Low-dose penicillin treatment
Maternal and early-life antibiotic exposure was performed using a subtherapeutic penicillin regimen[56,57]. Pregnant Lgr5-GFP mice received either regular drinking water or water containing penicillin G sodium salt (Sigma-Aldrich, P3032) at 6.67 mg l[−1], corresponding to ~1 mg kg[−1] body weight. Water was replaced weekly. Treatment continued throughout gestation and lactation to ensure pup exposure via maternal milk. After weaning, offspring remained on the same treatment until 8 weeks of age.

### Dextran sulfate sodium-induced colitis
To assess susceptibility to colitis, SPF *Lgr5-GFP* mice were treated with LDP until 8 weeks of age, followed by a 1-week antibiotic washout. Age-matched untreated mice served as controls. All mice received either 1 cycle or 3 cycles of 2.5% (w/v) DSS (molecular weight 36,000–50,000, MP Biomedicals, 9011-18-1) in drinking water as previously described[5]. Each cycle consisted of 5 days of DSS treatment followed by 5 days of recovery with regular water. Water consumption was recorded daily per cage, and bottles were replaced every 2 days. Because no sex-based differences were observed (Supplementary Fig. 1), data from both sexes were pooled for analysis.

### Azoxymethane–DSS-induced colitis-associated tumorigenesis
To induce colitis-associated colorectal cancer, 9-week-old mice were given a single intraperitoneal (i.p.) injection of AOM (10 mg kg[−1]; Sigma-Aldrich, A5486) followed by 3 cycles of 2.5% DSS as described above. Age-matched mice without LDP pretreatment served as controls. At 19 weeks of age, mice were euthanized, and tumour number and size were recorded. Colon tissues were collected for histological examination.

### Assessment of DSS-induced disease activity
During DSS treatment, disease progression was monitored daily by measuring body weight, stool consistency and rectal bleeding. Occult blood was detected using Hemoccult Faecal Occult Blood Slides (Beckman-Coulter, SK-61130). Colitis severity was scored according to established criteria[58,59]: weight loss (0 = none; 1 = 1%–5%; 2 = 5%–10%; 3 = 10%–20%; 4 = >20%); stool consistency (0 = normal; 1 = soft; 2 = loose; 3 = semi-liquid; 4 = diarrhoea); and bleeding (0 = normal; 1 = Haemoccult positive; 2 = Haemoccult positive and visual bleeding; 3 = gross bleeding). The disease activity index (DAI) was calculated as the sum of these three parameters. A subset of mice were euthanized at 7 days after inital DSS treatment, and colon length and histopathology were evaluated.

### Neutralization of IFNγ in mice
To block IFNγ signalling, mice received i.p. injections of Ultra-LEAF purified anti-mouse IFNγ antibody (clone XMG1.2, BioLegend, 505847) at 10 mg kg[−1] body weight. The antibody was administered immediately after weaning or in adulthood. Control mice received Ultra-LEAF purified Rat IgG1 κ isotype control antibody (clone RTK2071, BioLegend, 400457) at the same dose. A total of 4 injections were given to ensure sustained cytokine blockade. Colonic epithelial cells were collected 5 weeks after the first injection for downstream analyses.

### Organoid culture and treatment
Organoids were established from mouse colons[60,61]. Briefly, colons were dissected and incubated in ethylenediaminetetraacetic acid (EDTA) to dissociate epithelial crypts. The crypts were washed 4 times with cold washing buffer (comprising Ham's F-12 Nutrient Mix, 5% fetal bovine serum (FBS), 15 mM HEPES, 2.5 µM Rock-inhibitor Y-27632, 2 mM L-glutamine, 1% penicillin/streptomycin, 0.25 µg ml[−1] amphotericin B, 50 µg ml[−1] gentamicin and 100 µg ml[−1] Primocin) and filtered through a 70 µm strainer. The resulting cell suspension was embedded in Matrigel Matrix (Corning, 356231) and plated into pre-warmed 24-well plates. Cultures were maintained in WRNE medium containing Wnt3A, R-spondin 3 and Noggin. Medium was replaced every 3 days and organoids were passaged every 5–7 days.

For IFNγ treatment, GF-derived organoids were incubated with recombinant mouse IFNγ (R&D Systems, 485-MI-100) at 0.1 ng ml[−1] for 0, 24, 48, 72 or 96 h before collection. For single-agent treatment, organoids were treated for 48 h with 0.1 ng ml[−1] IFNγ, 5 ng ml[−1] TNF (R&D Systems, 410-MT-010), 2.5 µg ml[−1] LPS (Thermo Fisher, 00-4976-03), 2 mM sodium butyrate (Sigma-Aldrich, 303410) or 0.08 µM DAC (Sigma-Aldrich, A3656). For combination treatments,

organoids were pretreated with 0.08 µM DAC followed by 0.1 ng ml⁻¹ IFNγ for 48 h.

To assess trained memory responses, naïve and IFNγ-primed organoids were initially treated with vehicle or 0.1 ng ml⁻¹ IFNγ across 3 consecutive passages. Organoids were then washed and cultured in IFNγ-free medium for 7 days before restimulation with 0.1 ng ml⁻¹ IFNγ or 5 ng ml⁻¹ TNF. Organoids were collected at 0, 3, 6, 12, 24, 36 and 48 h post restimulation. Early-passage organoids were derived from 15-week-old SPF, GF and GMT mice and maintained in IFNγ-free medium for 20 days, then treated with 0.1 ng ml⁻¹ IFNγ and collected at the same time points for downstream analyses.

## Histology and immunohistochemistry (IHC)

Small intestines and colons were fixed in 4% paraformaldehyde, paraffin embedded, sectioned and stained with haematoxylin and eosin (H&E) following standard protocols at the Cellular and Molecular Morphology Core, Texas Medical Center Digestive Diseases Center. IHC was performed as described previously[62] using primary antibodies against Ki-67 (Thermo Fisher, MA5-14520, 1:100) and CD3 (Abcam, ab16669, 1:100). Alcian blue (AB) (pH 2.5) staining (Vector Labs, H-3501) was conducted following manufacturer instructions. Crypt length, the average number of Ki-67⁺ cells and the average number of AB⁺ cells per crypt–villus unit were quantified microscopically at ×20 magnification from at least 20 randomly selected crypts per sample.

## Isolation of Lgr5⁺ ISCs and EpCAM⁺ IECs

Colonic crypts were isolated as described above for organoid culture. Crypt-derived epithelial cells were dissociated into a single-cell suspension using TrypLE Express Enzyme (Thermo Fisher, 12605010) and filtered through a 40 µm strainer. Cells were stained with PE-conjugated anti-mouse CD326 (EpCAM) (Biolegend, clone G8.8, 118206, 1:200) and, when indicated, APC-conjugated anti-mouse CD24 (Biolegend, clone 30-F1, 138505, 1:100) for fluorescence-activated cell sorting. ISCs (EpCAM⁺/GFP⁺) and IECs (EpCAM⁺/GFP⁻ or EpCAM⁺/GFP⁻/CD24⁺) were sorted using a BD Aria Fusion cell sorter (BD Biosciences). Wild-type (WT) mice lacking the Lgr5-GFP knock-in served as negative controls (see gating strategy, Supplementary Fig. 2a).

## Colonic intraepithelial lymphocyte isolation

Colonic IELs were isolated following a modified protocol[63,64]. Briefly, colons were dissected, flushed with cold calcium/magnesium-free Hanks' balanced salt solution (HBSS) containing 2% FBS, cut longitudinally and minced into small pieces. Tissue fragments were incubated in EDTA-dithiothreitol (DTT) buffer (HBSS, 2% FBS, 1 mM EDTA and 1 mM DTT) for 30 min at 37 °C with gentle shaking. Crypts were mechanically released by pipetting and filtered through a 70 µm strainer. IELs were enriched using 40%/80% discontinuous Percoll (Sigma-Aldrich, GE17-0891-02) gradient centrifugation.

## Flow cytometry analysis

Colon epithelial cells were strained for viability using the Zombie Aqua Fixable Viability kit (BioLegend, 423101, 1:200) and blocked with anti-mouse CD16/CD32 (BioLegend, clone S17011E, 156603, 1:100). Surface staining was performed using fluorophore-conjugated antibodies against PE/Cyanine7 anti-mouse CD326 (EpCAM) (BioLegend, clone G8.8, 118215, 1:200), Brilliant Violet 650 anti-mouse I-A/I-E (MHC-II) (BioLegend, clone M5/114.15.2, 107641, 1:100) and PE/Dazzle 594 anti-mouse CD45 (BioLegend, clone 30-F11, 103145, 1:100) to exclude haematopoietic cells. MHC-II expression was quantified in live CD45⁻EpCAM⁺ epithelial cells (Supplementary Fig. 2b).

For intracellular IFNγ staining in IELs, isolated IELs were stimulated with Cell Activation Cocktail containing Brefeldin A (BioLegend, 423303). Cells were stained for viability and surface markers including CD45 (PE/Dazzle 594, BioLegend, clone 30-F11, 103145, 1:100), TCRβ (BUV395, BD, clone H57-597, 569248, 1:100), CD3 (PerCP/Cy5.5,

BioLegend, clone 17A2, 100217, 1:50), CD4 (APC/Cy7, BioLegend, clone RM4-5, 100525, 1:50), CD8a (Alexa Fluor 700, BioLegend, clone 53-6.7, 100729, 1:200), EpCAM/CD326 (PE/Cy7, BioLegend, clone G8.8, 118215, 1:200), CD19 (Brilliant Violet 785, BioLegend, clone 6D5, 115543, 1:100) and CD11b (Brilliant Violet 421, BioLegend, clone M1/70, 101235, 1:50). Cells were then fixed, permeabilized and stained for PE anti-mouse IFNγ (BioLegend, clone XMG1.2, 505807, 1:50). Samples were acquired on a BD Symphony A5 flow cytometer and data were processed using FlowJo software (Supplementary Fig. 2c).

## Whole-genome bisulfite sequencing analysis

WGBS was performed as previously described[14,61] using sorted Lgr5⁺ ISCs (EpCAM⁺/GFP⁺) and EpCAM⁺/GFP⁻ IECs from the colons of SPF and GF Lgr5-GFP mice at weaning (3 weeks old) and in adulthood (15 weeks old). To obtain sufficient material for stem cell samples, we pooled cells from multiple animals of the same sex. Genomic DNA (500 ng) was treated with sodium bisulfite using the EZ DNA Methylation-Direct kit (Zymo Research, D5021). The bisulfite-modified DNA was amplified using adaptor-specific primers, and fragments of 200–500 bp were isolated. The quantity and size of the libraries were measured with Pico-Green and the Agilent 2100 Bioanalyzer. Each library was sequenced using 150 bp paired-end reads, resulting in an average sequencing depth of ~30× for both female and male mice in each group and in the replicates (Supplementary Table 3). The reads were mapped to the mouse genome (mm10) using BSMAP. The average methylation ratio for each CpG site was calculated as the number of unconverted CpGs divided by the total reads covering that site.

The gene annotation and genomic positions of CpG islands (CGIs) and transposable elements (LINE, SINE and LTR) in mouse (mm10/GRCm38) were fetched using the Table Browser tool in the UCSC Genome Browser (http://genome.ucsc.edu). Gene promoter regions were defined as 1 kb upstream of transcription start sites (TSS). Gene body regions were defined as the region from the TSS to the transcription termination site (TTS). The methylation levels of these genomic features (promoters, gene bodies, CGIs, transposons) were calculated using the 'avgmod' module in BASALkit[65] (https://www.github.com/JiejunShi/BASAL).

The mouse enhancers were defined on the basis of the annotations of ENCODE candidate cis-regulatory elements (cCREs)[17] and downloaded from the UCSC Genome Browser website (https://hgdownload.soe.ucsc.edu/gbdb/mm10/encode3/ccre/encodeCcreCombined.bb). There are five cCRE classes: promoter-like signature (PLS), proximal enhancer-like signature (pELS), distal enhancer-like signature (dELS), DNase-H3K4me3 and CTCF-only. The promoter-proximal enhancers (enhP) in this study refer to pELS in cCREs.

DMRs were analysed using Metilene (v.0.2-862)[66] in 'de novo' mode with the option '-m 5' to identify regions containing at least five CpGs. A mean methylation difference cut-off of 0.15 was applied for both hypermethylated and hypomethylated DMRs. DMR-associated genes were defined as those with promoters or gene bodies overlapping a DMR. Shared DMRs were defined as regions exhibiting more than 50% reciprocal overlap, reflecting concordant microbiome-mediated epigenetic changes in ISCs and IECs.

## RNA-seq analysis

RNA-seq was performed as previously described[14] using 200 ng RNA extracted from colonic ISCs and IECs. Before sequencing, RNA quality was assessed using an Agilent 2100 Bioanalyzer with the Agilent RNA 6000 Nano kit. Library preparation and RNA-seq were performed by Novogene. BOWTIE2 software[67] was used for efficient realignment of RNA sequences, and gene expression levels for each sample were calculated with RSEM[68]. For differentially expressed gene analysis, we used DEseq2 (ref. 69) to rank genes by log2 fold change (>1.5) and adjusted the P values on the basis of multiple testing corrections (Bonferroni).

## Metagenomic whole-genome sequencing analysis

Luminal microbial samples were collected from the intestinal tract. For GMT groups, caecal contents were used, while faecal samples from the colon were collected for LDP groups. Microbial DNA was extracted using the QIAamp PowerFecal Pro DNA kit (Qiagen, 51804) and quantified with a Qubit Fluorometer. Library preparation and metagenomic sequencing were performed by Novogene. Genomic DNA (200 ng) was fragmented (~350 bp), end repaired, adapter ligated, PCR amplified and quality assessed. Libraries with effective concentration >3 nM were sequenced on an Illumina NovaSeq X Plus platform using 150 bp paired-end reads with ≥20 million reads (~6 Gb data) per sample.

Raw reads were trimmed and quality filtered using fastp (v.0.23.1), and host-derived sequences were removed by aligning to the mouse genome with Bowtie2 (v.2.5.4). Taxonomic classification across bacteria, archaea, fungi and viruses was performed with Kraken2 (v.2.1.3) and refined using Bracken (v.2.9). Functional annotation was carried out using DIAMOND (v.2.1.9) against the UniProt database and mapped to Gene Ontology databases. Alpha and beta diversity were calculated using the vegan and ade4 R packages. Differential abundance analysis at species and functional levels was conducted using metagenomeSeq, with significance determined by permutation tests and Benjamini–Hochberg FDR correction (FDR < 0.05).

## Quantitative DNA methylation analysis

Quantitative bisulfite pyrosequencing was used to assess DNA methylation, following protocols established in previous studies[70,71]. For clonal bisulfite sequencing, post-bisulfite PCR products were cloned into the TA vector pCR4-TOPO (Invitrogen, K457501). Plasmid DNA was extracted from 15–20 clones using a QIAprep Spin Miniprep kit (QIAGEN, 27106) and sequenced at the Sequencing Core Facility at Baylor College of Medicine. Primer sequences and sequencing assays are provided in Supplementary Table 4.

## Quantitative gene expression analysis

Gene expression was measured by quantitative PCR with reverse transcription (RT–qPCR) using *Taq*Man assays from Thermo Fisher. The mouse genes analysed were *Dnmt1* (Mm01151063_m1), *Dnmt3a* (Mm00432881_m1), *Dnmt3b* (Mm01240113_m1), *Dnmt3l* (Mm00457628_g1), *Tet1* (Mm01169087_m1), *Tet2* (Mm00524395_m1), *Tet3* (Mm00805756_m1), *Cd74* (Mm01262763_m1), *H2-Aa* (Mm00439211_m1), *H2-Eb1* (Mm00439221_m1), *Ciita* (Mm00482914_m1), *Duoxa2* (Mm00470560_m1) and *Plet1* (Mm01170995_m1). Target gene expression was normalized to *Actb* (Mm00607939_s1) expression using an ABI Step OnePlus Detection System.

## Quantification of intestinal bacterial populations by real-time PCR

Bacterial genomic DNA was extracted from faecal samples collected from mice with and without LDP treatment at 4 weeks of age, using the QIAamp Fast DNA Stool mini kit (QIAGEN, 51604) following manufacturer instructions. Quantitative PCR was performed using the ABI Step OnePlus Detection System with PowerUp SYBR Green Master Mix (Applied Biosystems, A25776). The following primer sets were used as described previously[72–76]: universal bacteria primers (5′-CGGCAACGAGCGCAACCC-3′ and 5′-CCATTGTAGCACGTGTGTAGCC-3′), segmented filamentous bacteria (SFB) primers (5′-GACGCTGAGGCATGAGAGCAT-3′ and 5′-GACGGCACGGATTGTTATTCA-3′), *Bacteroides–Prevotella* primers (5′-GAAGGTCCCCCACATTG-3′ and 5′-CAATCGGAGTTCTTCGTG-3′), *Clostridium* cluster XIVab primers (5′-GAWGAAGTATYTCGGTATGT-3′ and 5′-CTACGCWCCCTTTACAC-3′) and *Lactobacillus* primers (5′-AGCAGTAGGGAATCTTCCA-3′ and 5′-ATTYCACCGCTACACATG-3′). Relative abundances of individual bacterial species were normalized on the basis of the total bacterial load.

## CpGfree-enhancer Lucia reporter assay

To measure the effect of DNA methylation in enhancer elements, we used the reporter plasmid pCpGfree-promoter-Lucia vector (InvivoGen, pcpgf-promlc) and a firefly luciferase pGL3 promoter (CMV) vector[77]. Putative enhancer regions were amplified from mouse genomic DNA and cloned into the pCpGfree-promoter-Lucia vector, which had been digested with ApaI/BamHI for *Cd74*, *H2-Aa*, *Ciita*, *Duoxa2* and *Plet1*, or with ScaI/BamHI for *H2-Eb1*. All primer sequences used for generating plasmids for the Lucia reporter assays are listed in Supplementary Table 5. Plasmids were methylated using the CpG methyltransferase M.SssI (New England Biolabs, M0226L) according to manufacturer instructions. Plasmid DNA (10 μg) was incubated overnight at 37 °C with 160 μM *S*-adenosylmethionine (SAM) and 20 U of M.SssI. An additional 160 μM SAM and 20 U M.SssI were then added, followed by a 6-h incubation. Unmethylated plasmids were prepared similarly but without M.SssI.

The purified plasmids were quantified and co-transfected transiently into HEK-293T cells along with a firefly luciferase plasmid using Lipofectamine 3000 Transfection Reagent (Invitrogen, L3000008) following manufacturer protocol. After 24 h, cells were lysed, and Lucia and firefly luciferase activities were measured in a GloMax Discover plate reader (Promega) using the Dual-Luciferase Reporter Assay System (Promega, E1910). Lucia activity was normalized to firefly activity.

## Statistical analysis

All statistical analyses were performed using GraphPad Prism v.9 (RRID:SCR_002798). Quantitative DNA methylation and expression results are expressed as mean ± s.e.m. Significance was determined using Student's *t*-test with a two-tailed distribution or two-way ANOVA, and *P* < 0.05 was considered statistically significant.

## Reporting summary

Further information on research design is available in the Nature Portfolio Reporting Summary linked to this article.

## Data availability

All WGBS data are available in GEO under accession number GSE275219. All RNA-seq data are under accession number GSE275418. All shotgun metagenomic data are under accession number GSE310995. Source data are provided with this paper.

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

## Acknowledgements

This work was funded by grants from the March of Dimes (6-FY18-135 to L.S.), the US Department of Agriculture (USDA) (CRIS 3092-51000-060 to L.S.), and the National Institutes of Health (NIH) (R21HD101035, R01CA233472 and R01HD100914 to L.S.). The content is solely the responsibility of the authors and does not necessarily represent the official views of the USDA or the NIH. We thank A. Gillum for assistance with the figures. This work was also supported in part by the National Institute of Diabetes and Digestive and Kidney Diseases T32 fellowship (T32DK07664 to R.C.P.). We thank the shared resources provided by the Cellular and Molecular Morphology core at Texas Medical Center Digestive Disease Center (NIH/NIDDK P30 DK056338), the Gulf Coast Center for Precision Environmental Health (NIH/NIEHS P30ES030285), and the Cancer Center Core Grant (NIH/NCI P30CA125123). L.S. is a member of the Dan L Duncan Comprehensive Cancer Center at Baylor College of Medicine and the Texas Children's Research Institute at the Texas Children's Hospital.

## Author contributions

L.S. conceptualized the study and acquired funding. L.Y., J.S. and L.S. designed the research strategy. L.Y., R.C.P., S.Z., X.C., L.M.F., F.G., S.F., L.Z., J.M.S., K.R. and J.S. performed research and analysed data. L.Y., S.Z., J.S. and L.S. wrote the paper. All authors provided critical feedback on the manuscript.

## Competing interests

The authors declare no competing interests.

## Additional information

**Extended data** is available for this paper at https://doi.org/10.1038/s41564-026-02295-6.

**Correspondence and requests for materials** should be addressed to Jiejun Shi or Lanlan Shen.

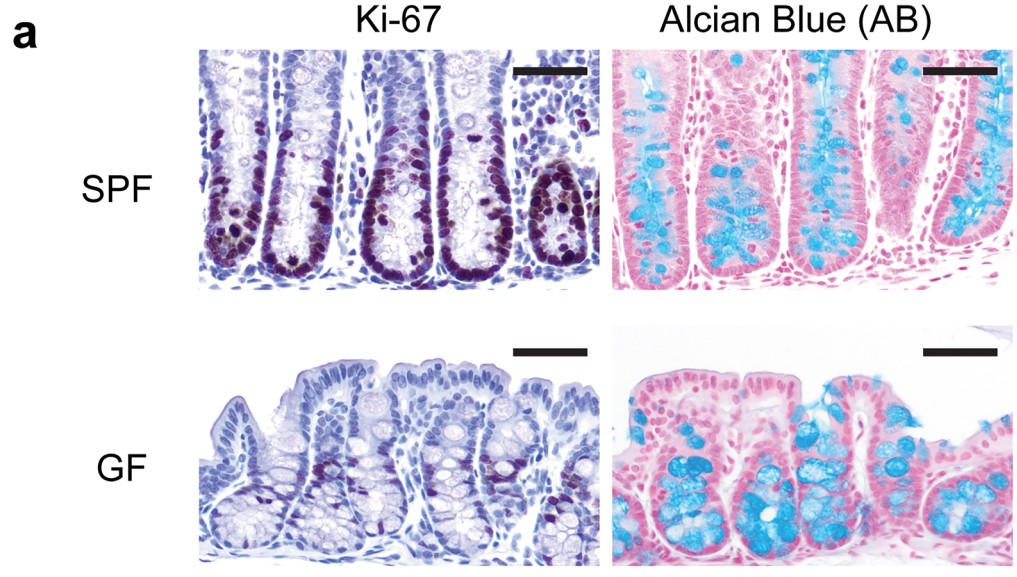

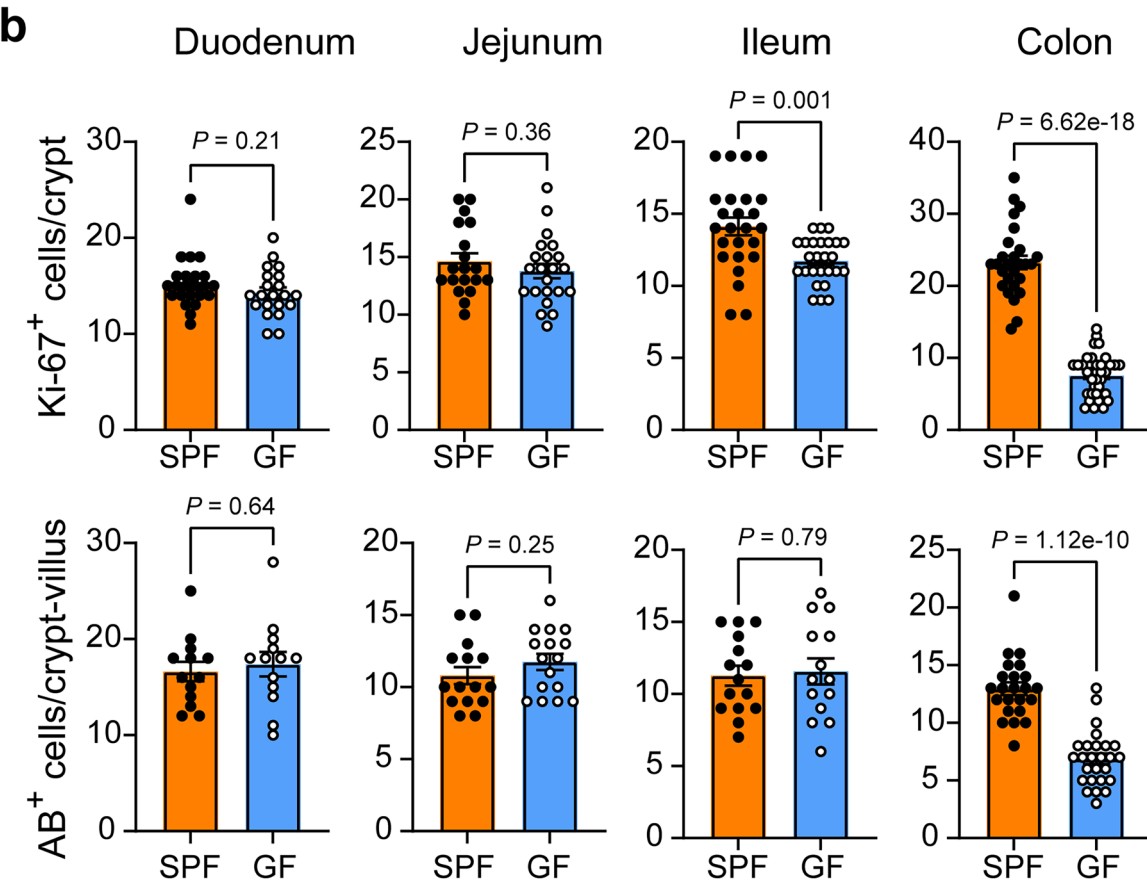

**Extended Data Fig. 1 | Impact of microbial status on Ki-67 expression and goblet cell abundance in the intestines of Lgr5-GFP mice.** (**a**) Representative images of IHC staining for Ki-67 and Alcian blue (AB) in colon tissues from mice under SPF (n = 4) or GF (n = 3) housing conditions. Scale bars, 50 μm.

(**b**) Quantification of Ki-67+ cells and AB+ goblet cells across intestinal segments (duodenum, jejunum, ileum, and colon) in SPF and GF mice. All data are presented as mean ± s.e.m. of at least 3 biological replicates. *P* values were calculated using unpaired (two-tailed) *t*-test.

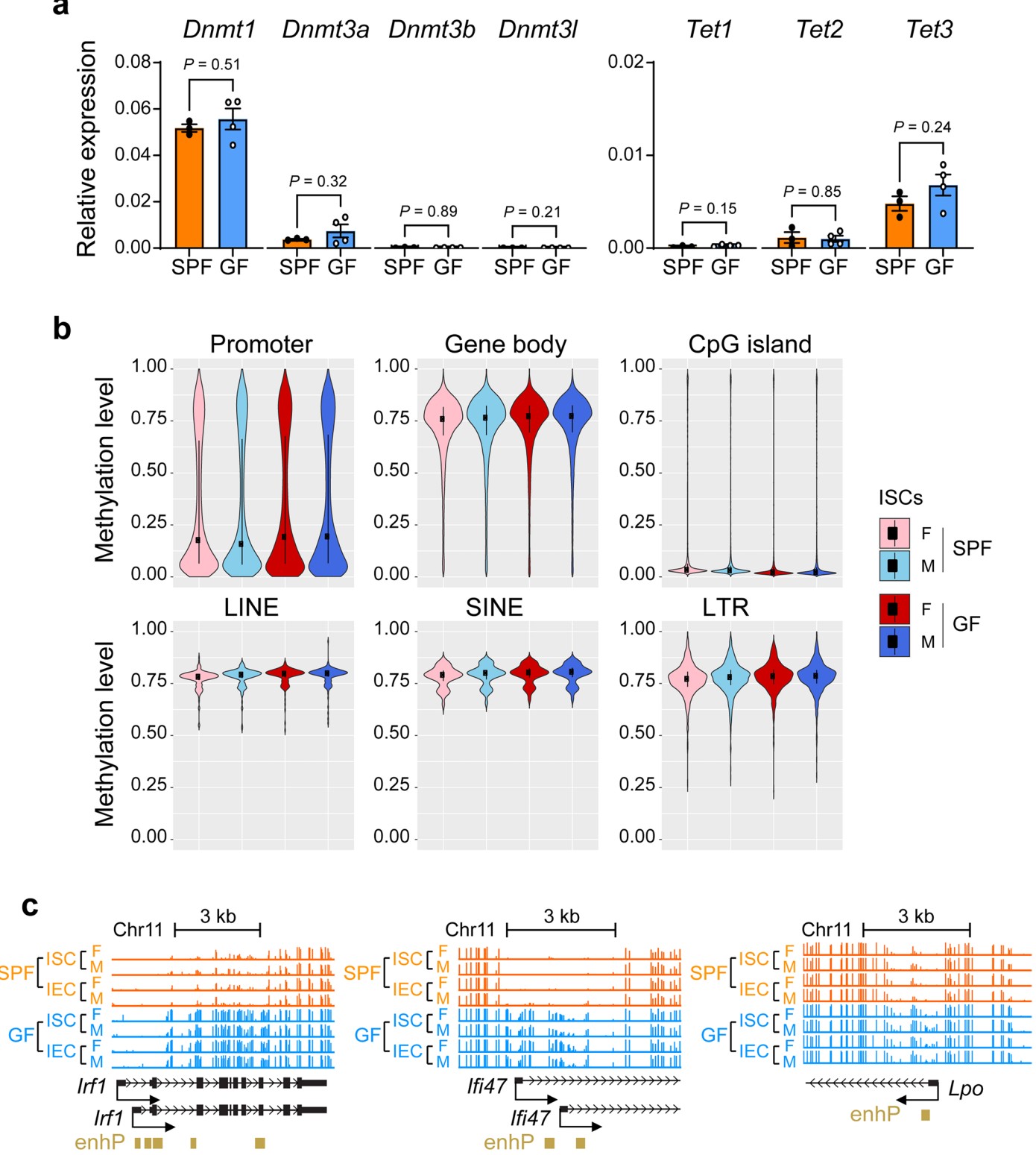

**Extended Data Fig. 2 | DNA methylation regulators, global methylation patterns, and microbiome-associated DMRs in colonic epithelial populations.** (**a**) Expression of DNA methylation regulators (*Dnmt* and *Tet* family genes) in colonic Lgr5⁺ ISCs of SPF (n = 3) and GF (n = 4) mice. *Dnmt1* and *Tet3* show the highest expression, with no significant differences between SPF and GF conditions. Data are presented as mean ± s.e.m. *P* values were calculated using unpaired (two-tailed) *t*-test. (**b**) Global DNA methylation levels in colonic

ISCs from SPF and GF mice across different gene categories and repetitive elements. Violin plots depict the distribution of methylation levels in ISCs from male and female mice, showing comparable global methylation patterns between conditions. (**c**) Genome browser views of additional microbiome-associated DMRs enriched at proximal enhancers in Lgr5⁺ ISCs and EpCAM⁺ IECs, highlighting loci associated with immune-related and host defense genes.

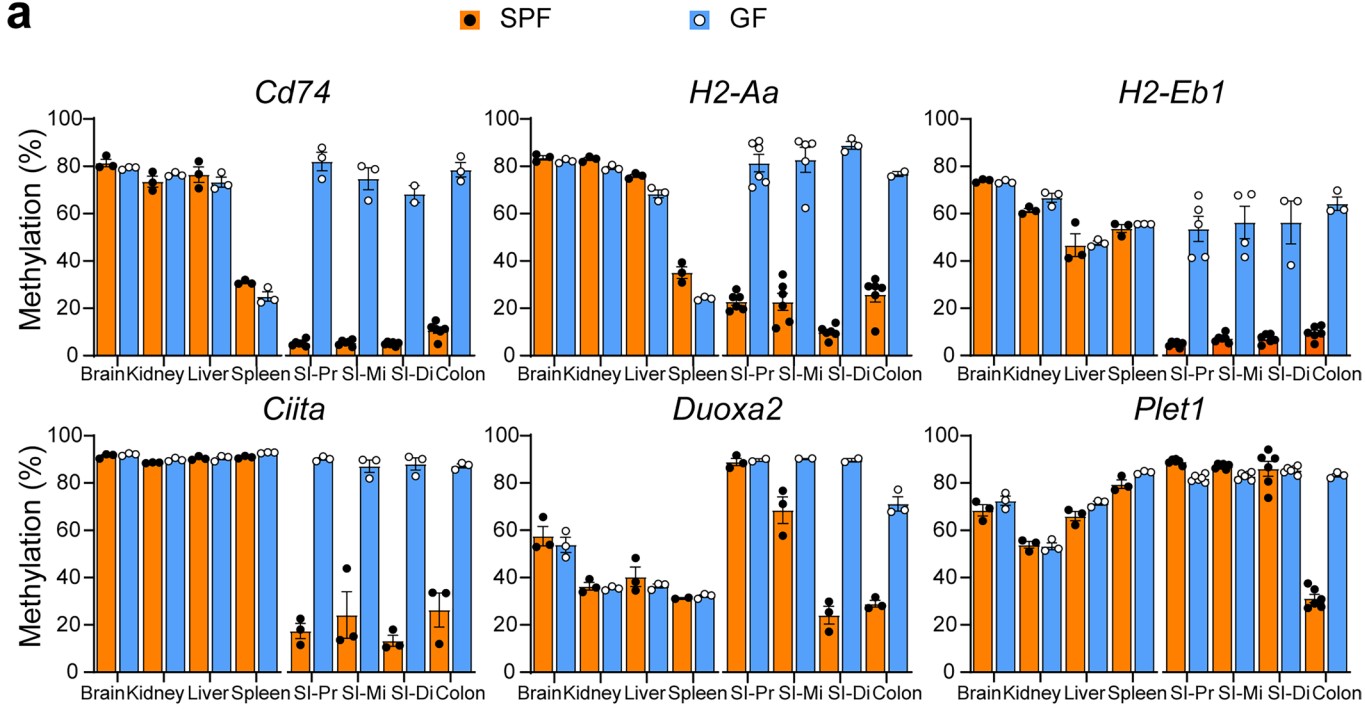

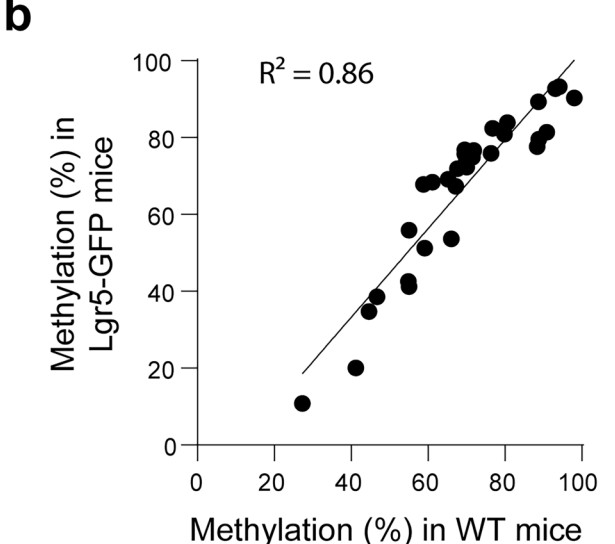

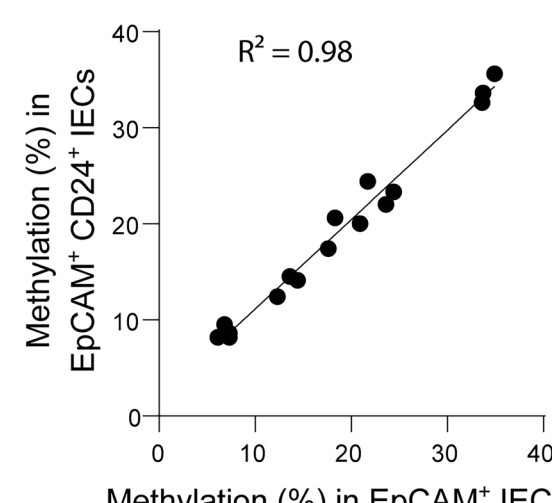

**Extended Data Fig. 3 | Microbiome-associated DMRs are intestine-specific.**
(**a**) Microbiome-associated methylation changes were restricted to the intestine and were not observed in other tissues analyzed. MHC-II genes displayed microbiome-dependent methylation changes across all intestinal segments, whereas other genes (for example, *Duoxa2* and *Plet1*) showed changes specifically in the distal small intestine and colon. Data are presented as mean ± s.e.m. of at least 3 biological replicates. (**b**) Methylation levels at DMRs were similar in IECs from wild-type and Lgr5-GFP mice ($R^2 = 0.86$). (**c**) EpCAM$^+$ IECs and a subpopulation of EpCAM$^+$CD24$^+$ IECs had nearly identical microbiome-induced methylation changes ($R^2 = 0.98$).

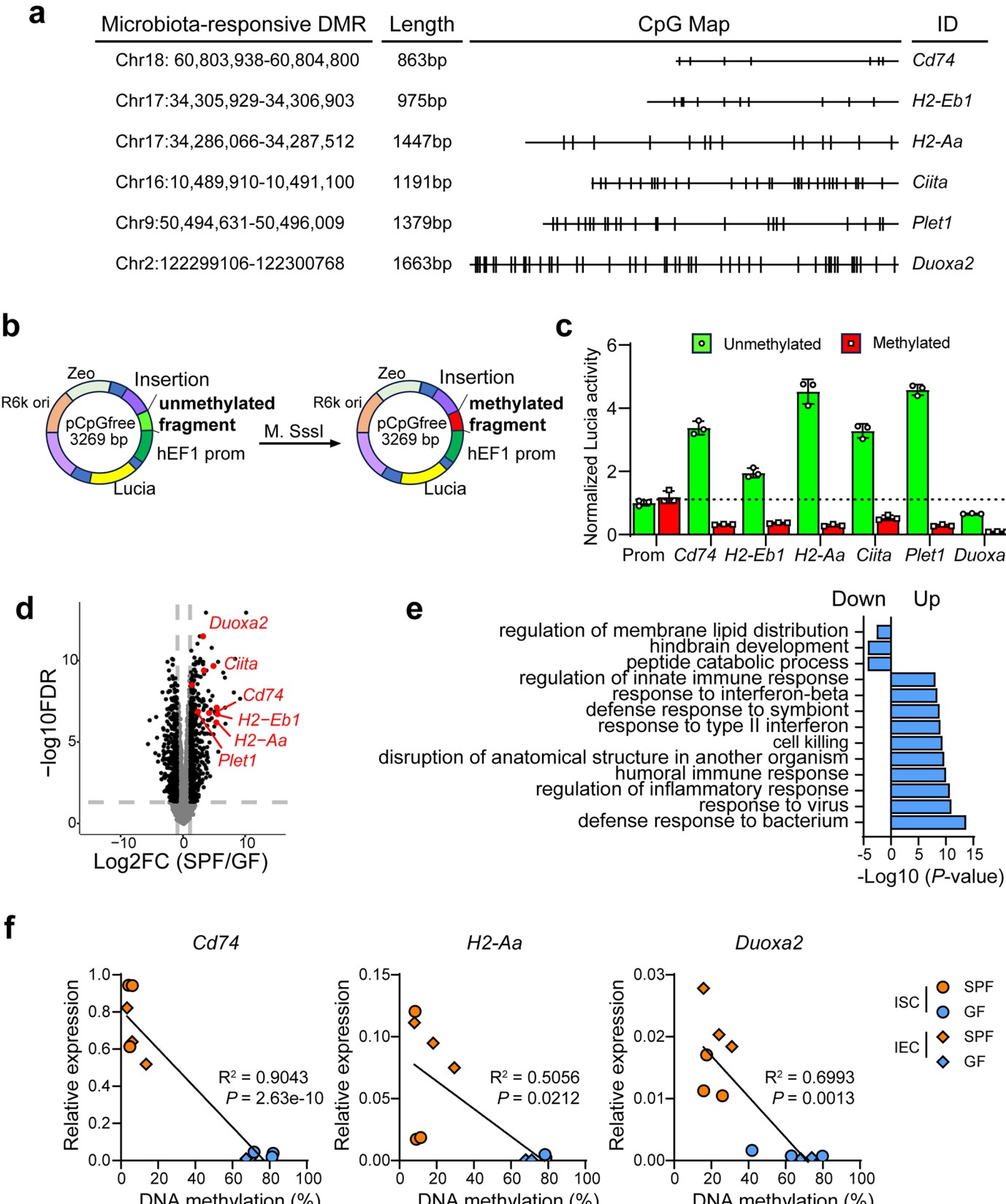

**Extended Data Fig. 4 | See next page for caption.**

**Extended Data Fig. 4 | Microbiota-induced enhancer hypomethylation of immune response genes sustains transcriptional activation during ISC differentiation.** (**a**) CpG maps of putative enhancers associated with immune-related genes. Vertical bars indicate CpG sites. *Duoxa2* is included as a negative control as it lacks ENCODE enhancer annotation. (**b**) Candidate enhancer fragments were inserted into a CpG-free vector containing a human EF1 promoter and the Lucia reporter gene. CpG sites within these fragments were methylated *in vitro* using M. SssI DNA methylase. (**c**) Reporter assays demonstrated that the enhancer activity of the identified candidate genes depended on methylation status. Data are presented as mean ± s.e.m. of 3 independent experiments.

(**d**) RNA-seq analysis revealed that transcriptional activation of hypomethylated immune genes ranked among the top hits in SPF ISCs. (**e**) GO analysis showed significant enrichment of immune response pathways among the upregulated genes in SPF mice. Enrichment significance was assessed using a hypergeometric test, and the *P* values were adjusted for multiple comparisons using the Benjamini-Hochberg method. (**f**) Quantitative DNA methylation and gene expression analyses demonstrating a negative correlation between methylation levels at microbiome-associated DMRs and the expression of nearby genes in both ISCs and IECs. Linear regression analysis was performed to assess the statistical significance.

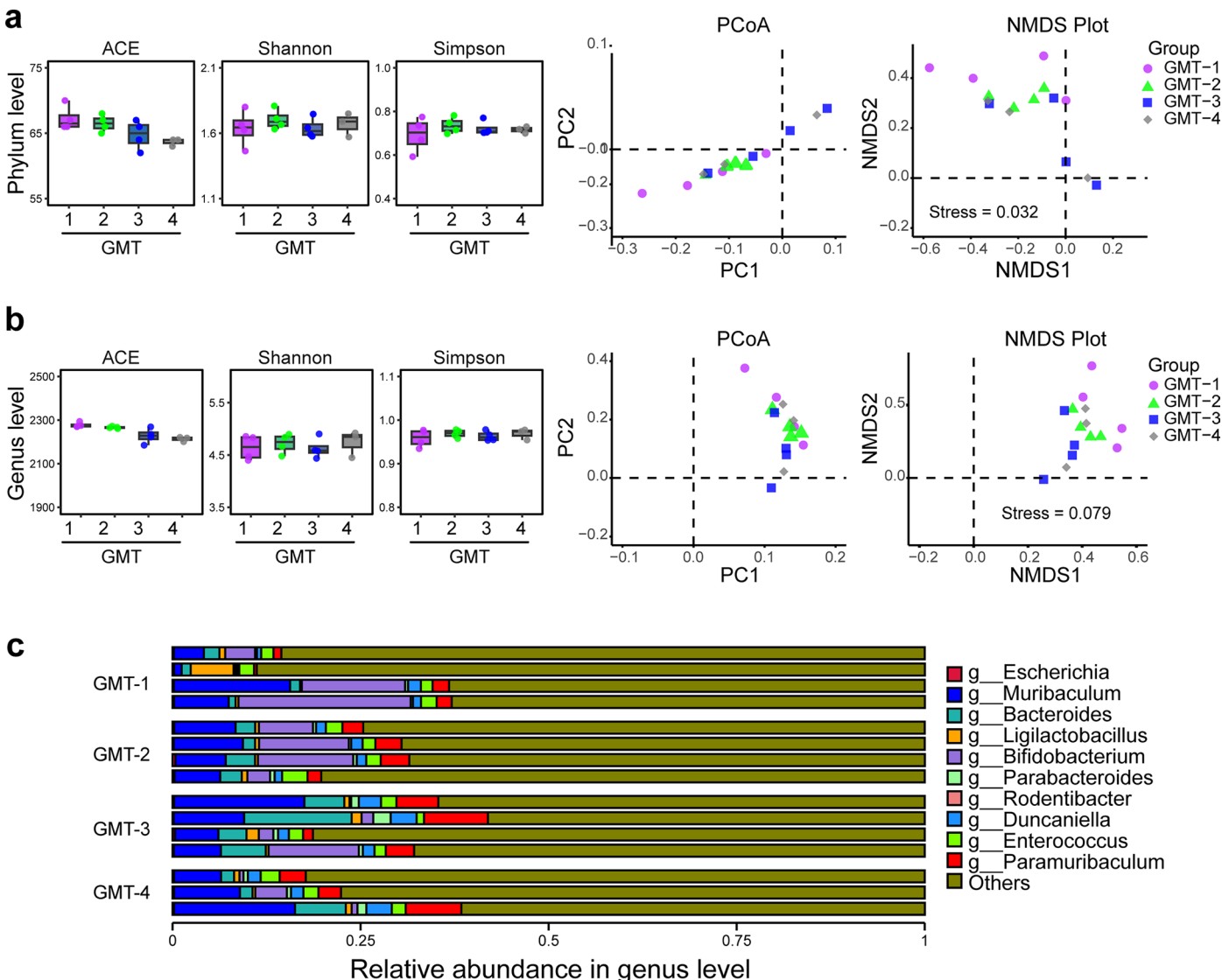

**Extended Data Fig. 5 | Microbiota diversity and composition across GMT groups.** (**a**) Alpha diversity (ACE, Shannon, Simpson) and beta diversity (PCoA and NMDS) at the phylum level. (**b**) Alpha and beta diversity at the genus level.

(**c**) Relative abundance of bacterial genera. In a-c, n = 3-4 per group. All box plots indicate the median (center line), upper and lower quartiles (boxes), and minimum to maximum values (whiskers); all individual data points are shown.

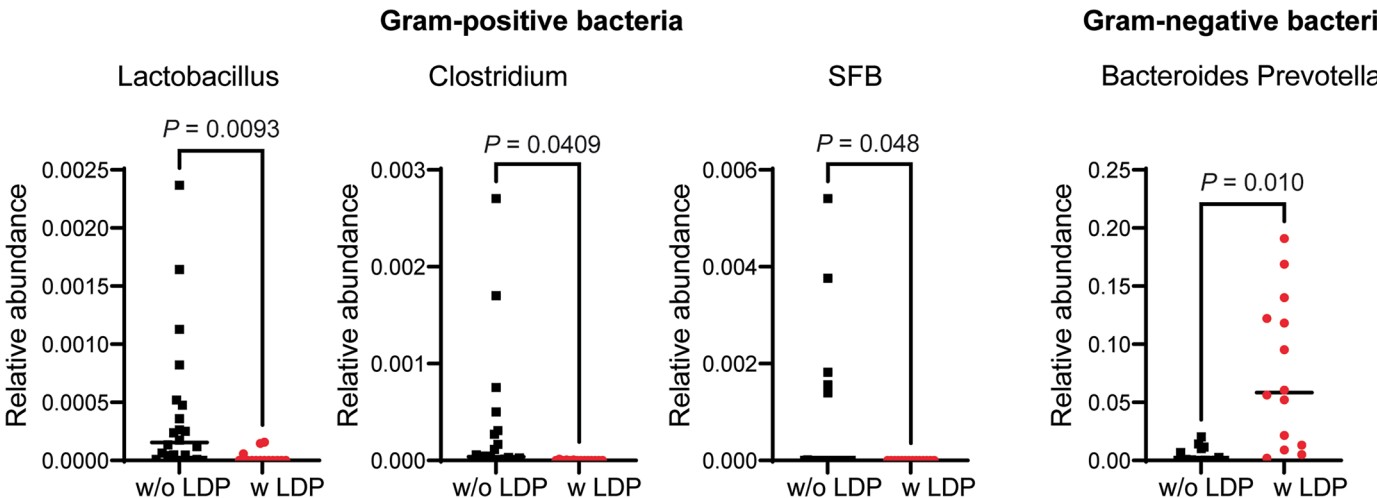

**Extended Data Fig. 6 | qPCR quantification of Gram-positive and Gram-negative bacteria in fecal samples from mice with or without low-dose penicillin (LDP) treatment.** Data are presented as mean ± s.e.m. and analyzed by unpaired (two-tailed) *t*-tests (w/o LDP, n = 22; w LDP, n = 14).

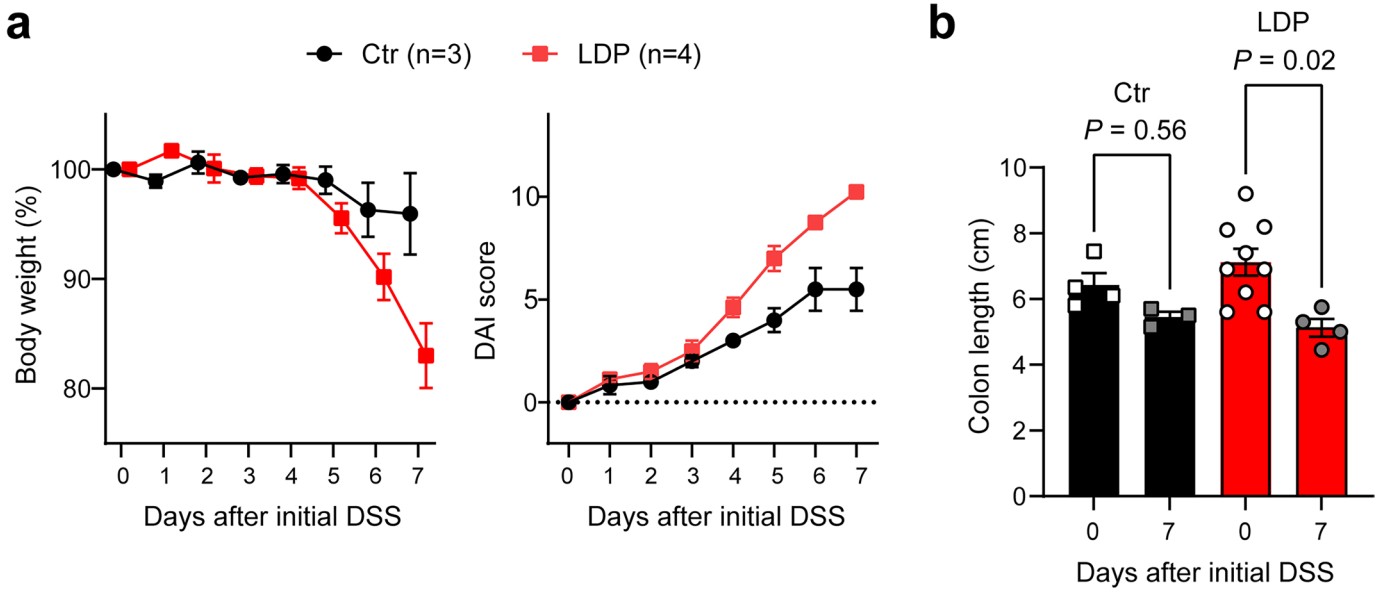

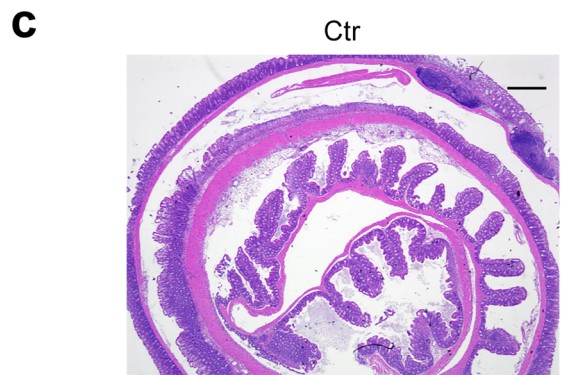

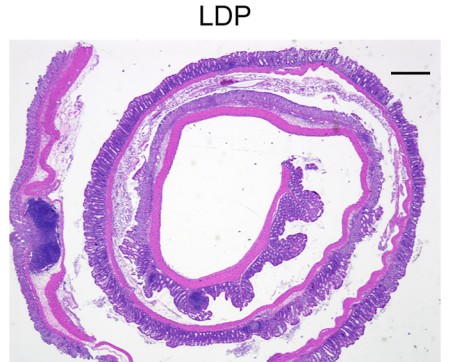

**Extended Data Fig. 7 | Early life exposure to LDP exacerbates DSS-induced colitis.** (**a**) LDP-pretreated mice lost more weight and had more severe colitis symptoms following the first cycle of DSS administration. (**b**) LDP-pretreated mice showed significantly reduced colon length 7 days after initial DSS treatment, whereas control mice exhibited no significant change. (**c**) Histology of DSS-induced colitis. Compared to controls, LDP-pretreated mice experienced severe loss of colonic crypts following 7 days of DSS administration. Scale bars, 500 μm. Ctr: control. In **a** and **b**, data are presented as mean ± s.e.m. of at least 3 biological replicates and analyzed by unpaired (two-tailed) *t*-tests.

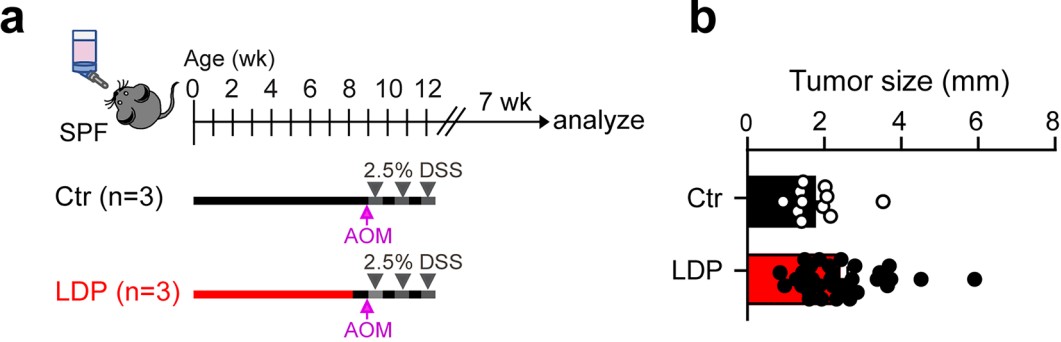

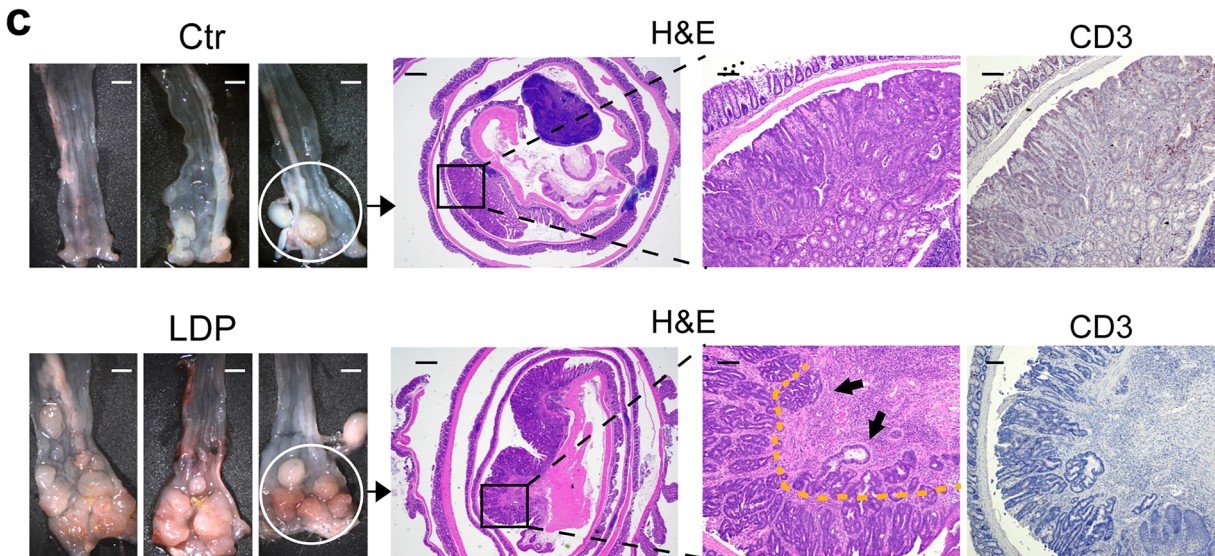

**Extended Data Fig. 8 | Early life exposure to LDP promotes colitis-associated colon cancer. (a)** Mice were exposed to LDP until 8 weeks of age, then given a single dose of AOM at 9 weeks, followed by three cycles of DSS. Tumors were analyzed at 19 weeks. **(b)** LDP-pretreated mice developed more and larger colon tumors than controls. Dots represent individual tumors from 3 mice per group.

**(c)** Representative images of distal colon tumors under a dissecting microscope, along with H&E and IHC staining. Tumors in LDP-pretreated mice were more aggressive with the loss of CD3⁺ immune cells. Arrows indicate invasive adenocarcinoma. Scale bars, 500 μm (left; dissecting microscope and low-magnification H&E images) and 100 μm (right; higher-magnification images).

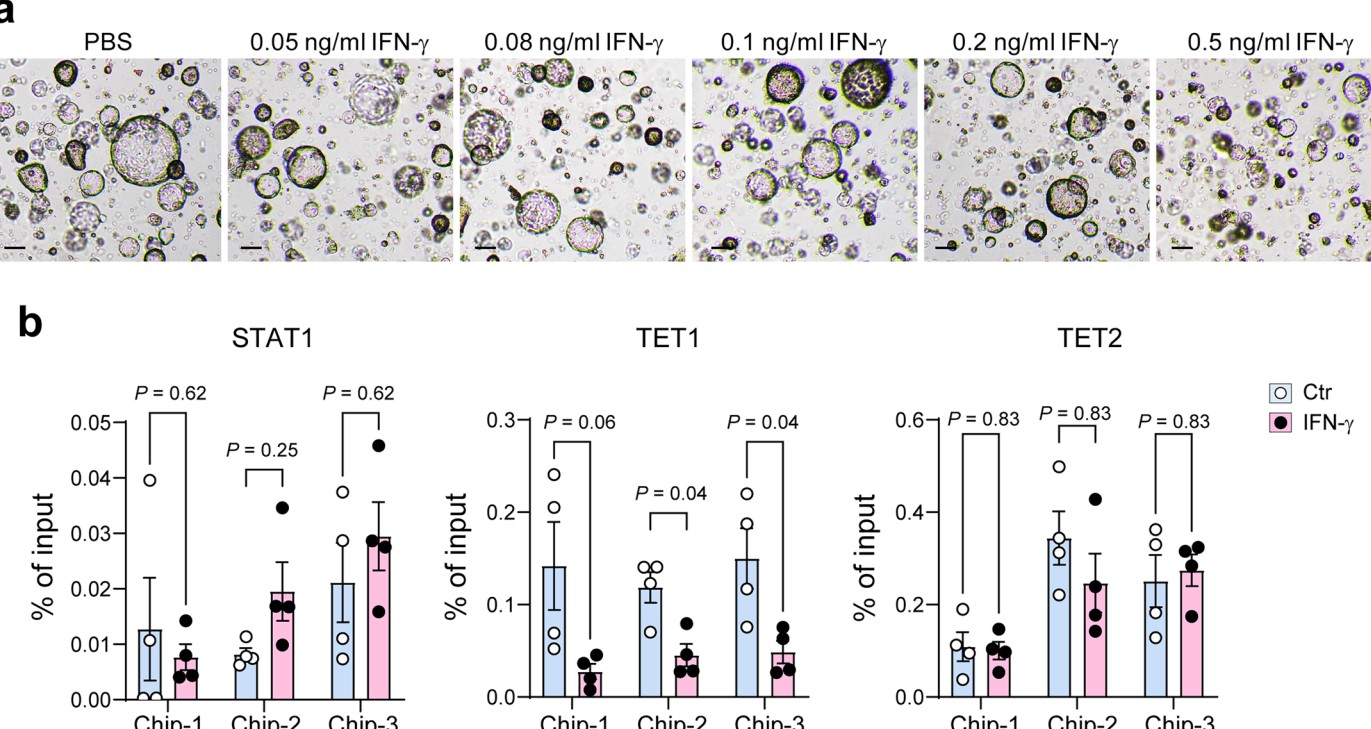

**Extended Data Fig. 9 | Effects of IFNγ on colonic organoids and lack of STAT1 and TET1/2 enrichment at the demethylated *Cd74* enhancer.** (**a**) GF colonic organoids treated with increasing concentration of IFNγ for 48 h. Low-dose IFNγ (0.1 ng/ml) had minimal effects on organoid proliferation and viability. Representative bright-field images show dose-dependent morphological changes. PBS, phosphate-buffered saline. Scale bars, 500 μm. (**b**) ChIP-qPCR analysis of GF colonic organoids treated with 0.1 ng/ml IFNγ to induce demethylation of the *Cd74* enhancer. Binding of STAT and TET family members was assessed at the enhancer regions showing demethylation (ChIP-2 and ChIP-3) and at a non-DMR control region (ChIP-1). STAT1, TET1, and TET2 showed no detectable enrichment at the demethylated enhancer regions relative to the control. Data are presented as mean ± s.e.m. of 4 independent experiments. *P* values were calculated using unpaired (two-tailed) *t*-test.

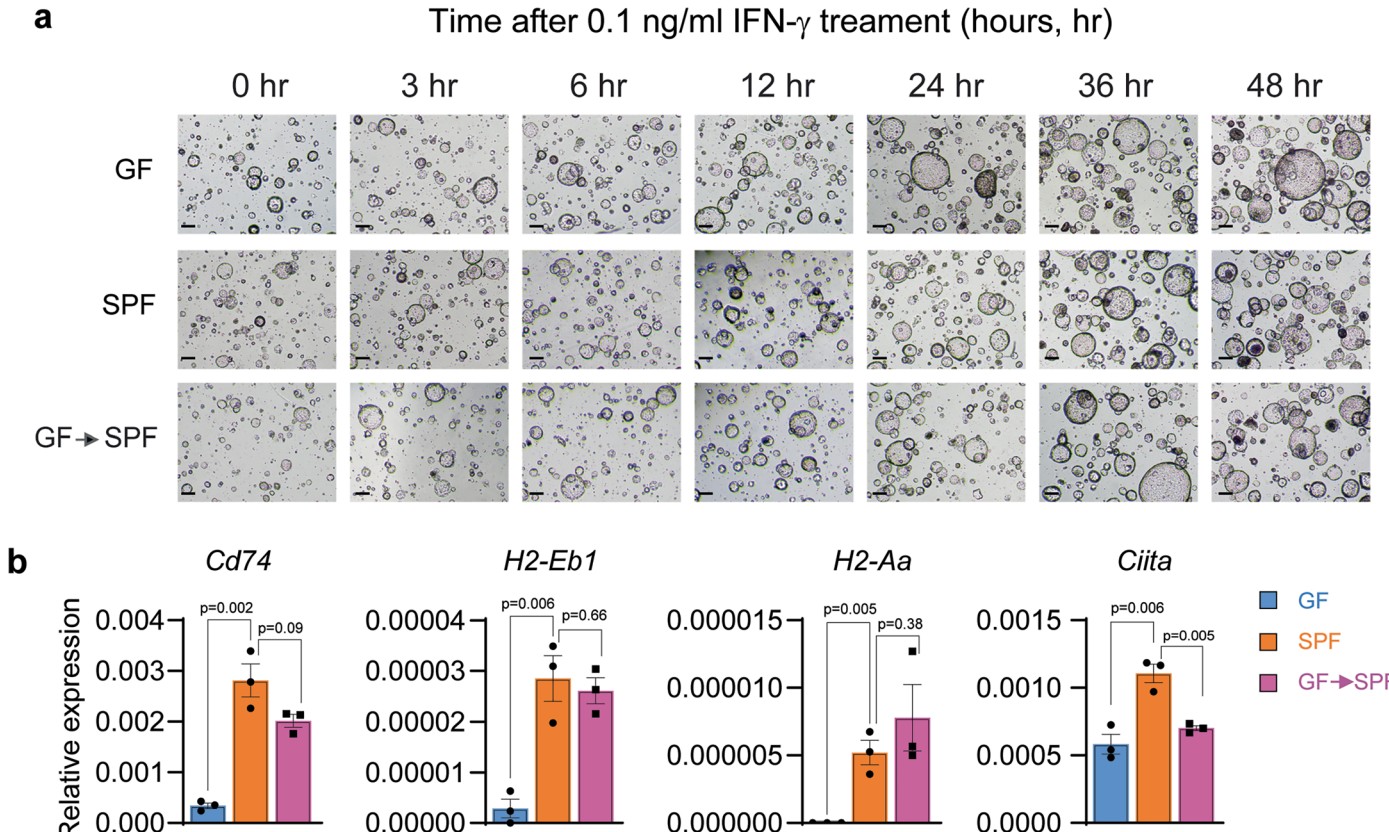

**a** Time after 0.1 ng/ml IFN-γ treament (hours, hr)

**Extended Data Fig. 10 | Microbiota-dependent IFNγ-induced MHC-II activation occurs independently of proliferation.** (**a**) Representative images show comparable proliferation and viability following IFNγ treatment in organoids derived from GF, SPF, or GF→SPF (converted at weaning) mice. Scale bars, 500 µm. (**b**) 6 h post-treatment, MHC-II gene activation was enhanced only in organoids from mice with prior microbial exposure, indicating that induction is microbiota- and methylation-dependent, and not due to changes in proliferation. Data are presented as mean ± s.e.m. of 3 independent experiments. P values were calculated using unpaired (two-tailed) t-test.

# Reporting Summary

## Statistics

For all statistical analyses, confirm that the following items are present in the figure legend, table legend, main text, or Methods section.

| n/a | Confirmed | |
|---|---|---|
| ☐ | ☒ | The exact sample size (*n*) for each experimental group/condition, given as a discrete number and unit of measurement |
| ☐ | ☒ | A statement on whether measurements were taken from distinct samples or whether the same sample was measured repeatedly |
| ☐ | ☒ | The statistical test(s) used AND whether they are one- or two-sided *Only common tests should be described solely by name; describe more complex techniques in the Methods section.* |
| ☒ | ☐ | A description of all covariates tested |
| ☐ | ☒ | A description of any assumptions or corrections, such as tests of normality and adjustment for multiple comparisons |
| ☐ | ☒ | A full description of the statistical parameters including central tendency (e.g. means) or other basic estimates (e.g. regression coefficient) AND variation (e.g. standard deviation) or associated estimates of uncertainty (e.g. confidence intervals) |
| ☐ | ☒ | For null hypothesis testing, the test statistic (e.g. *F*, *t*, *r*) with confidence intervals, effect sizes, degrees of freedom and *P* value noted *Give P values as exact values whenever suitable.* |
| ☒ | ☐ | For Bayesian analysis, information on the choice of priors and Markov chain Monte Carlo settings |
| ☒ | ☐ | For hierarchical and complex designs, identification of the appropriate level for tests and full reporting of outcomes |
| ☒ | ☐ | Estimates of effect sizes (e.g. Cohen's *d*, Pearson's *r*), indicating how they were calculated |

*Our web collection on statistics for biologists contains articles on many of the points above.*

## Software and code

Policy information about availability of computer code

| Data collection | Data were saved as Microsoft Excel (.xls) and Word (.docx), Portable Document Format (.pdf), Joint Photographic Experts Group (.jpg), and Tag Image File Format (.tif). Some raw data were collected using Flow Cytometry Standard (.fcs). Genomic sequencing data were stored in raw fastq files. |
|---|---|
| Data analysis | We utilized the following publically available softwares: BSMAP BASALkit (http://www.github.com/JiejunShi/BASAL) Metilene v0.2-862 RSEM DEseq2 PyroMark Q48 Auto StepOne Software V2.3 CellSensStandard FlowJo GraphPad Prism 9 Bowtie2 (v2.5.4) Kraken2 (v2.1.3) Bracken (v2.9) DIAMOND (v2.1.9) |

For manuscripts utilizing custom algorithms or software that are central to the research but not yet described in published literature, software must be made available to editors and reviewers. We strongly encourage code deposition in a community repository (e.g. GitHub). See the Nature Portfolio guidelines for submitting code & software for further information.

## Data

Policy information about availability of data
All manuscripts must include a data availability statement. This statement should provide the following information, where applicable:

- Accession codes, unique identifiers, or web links for publicly available datasets
- A description of any restrictions on data availability
- For clinical datasets or third party data, please ensure that the statement adheres to our policy

> Sequencing data are available at the GEO repository under the accession numbers "GSE275219", "GSE275418" and "GSE310995". Source data are provided in the paper.

## Research involving human participants, their data, or biological material

Policy information about studies with human participants or human data. See also policy information about sex, gender (identity/presentation), and sexual orientation and race, ethnicity and racism.

| | |
|---|---|
| Reporting on sex and gender | Our study does not involve human subjects. |
| Reporting on race, ethnicity, or other socially relevant groupings | Our study does not involve human subjects. |
| Population characteristics | Our study does not involve human subjects. |
| Recruitment | Our study does not involve human subjects. |
| Ethics oversight | Our study does not involve human subjects. |

Note that full information on the approval of the study protocol must also be provided in the manuscript.

# Field-specific reporting

Please select the one below that is the best fit for your research. If you are not sure, read the appropriate sections before making your selection.

☒ Life sciences   ☐ Behavioural & social sciences   ☐ Ecological, evolutionary & environmental sciences

For a reference copy of the document with all sections, see nature.com/documents/nr-reporting-summary-flat.pdf

# Life sciences study design

All studies must disclose on these points even when the disclosure is negative.

| | |
|---|---|
| Sample size | Sample size and statistical testes for each experiment are stated in the manuscript and figure legends. The number of animals chosen is based on our previous epigenome studies (PMID: 26420038). |
| Data exclusions | No data was excluded. |
| Replication | At least two independent experiments with three technical repeats were performed. |
| Randomization | Mice were randomly assigned to treatment groups. For all experiments, sample processing and data analysis were performed blinded to group assignment. |
| Blinding | Histology images were scored by a blinded pathologist. |

# Reporting for specific materials, systems and methods

We require information from authors about some types of materials, experimental systems and methods used in many studies. Here, indicate whether each material, system or method listed is relevant to your study. If you are not sure if a list item applies to your research, read the appropriate section before selecting a response.

## Materials & experimental systems

| n/a | Involved in the study |
|---|---|
| ☐ | ☒ Antibodies |
| ☐ | ☒ Eukaryotic cell lines |
| ☒ | ☐ Palaeontology and archaeology |
| ☐ | ☒ Animals and other organisms |
| ☒ | ☐ Clinical data |
| ☒ | ☐ Dual use research of concern |
| ☒ | ☐ Plants |

## Methods

| n/a | Involved in the study |
|---|---|
| ☒ | ☐ ChIP-seq |
| ☐ | ☒ Flow cytometry |
| ☒ | ☐ MRI-based neuroimaging |

# Antibodies

| | |
|---|---|
| Antibodies used | PE/Cyanine7 anti-mouse CD326 (EpCAM) (BioLegend, Clone G8.8, #118215, 1:200)<br>Anti-mouse CD16/CD32 (BioLegend, Clone S17011E, #156603, 1:100)<br>PE/Dazzle™ 594 anti-mouse CD45 (BioLegend, Clone 30-F11, #103145, 1:100)<br>BUV395 hamster anti-mouse TCRβ (BD, Clone H57-597, #569248, 1:100)<br>PerCP/Cyanine5.5 anti-mouse CD3 (BioLegend, Clone 17A2, #100217, 1:50)<br>APC/Cy7 anti-mouse CD4 (BioLegend, Clone RM4-5, #100525, 1:50)<br>Alexa Fluor® 700 anti-mouse CD8a (BioLegend, Clone 53-6.7, #100729, 1:200)<br>Brilliant Violet 785™ anti-mouse CD19 (BioLegend, Clone 6D5, #115543, 1:100)<br>Brilliant Violet 421™ anti-mouse CD11b (BioLegend, Clone M1/70, #101235, 1:50)<br>Brilliant Violet 650™ anti-mouse I-A/I-E (MHC II) (BioLegend, Clone M5/114.15.2, #107641, 1:100)<br>PE anti-mouse IFN-γ (BioLegend, Clone XMG1.2, #505807, 1:50)<br>PE anti-mouse CD326 (EpCAM) antibody (Biolegend, Clone G8.8, #118206, 1:200)<br>APC anti-mouse CD24 (BioLegend, Clone 30-F1, #138505, 1:100)<br>Ultra-LEAF™ purified anti-mouse IFN-γ antibody (BioLegend, Clone XMG1.2, #505847)<br>Ultra-LEAF™ purified Rat IgG1, κ Isotype Ctrl antibody (BioLegend, Clone RTK2071, #400457)<br>TET1 polyclonal antibody [N1], N-term (GeneTex, #GTX125888, 1:50)<br>TET2 Rabbit Monoclonal Antibody (Cell Signaling Technology, Clone D9K3E, #92529, 1:25)<br>Anti-TET3 polyclonal antibody (Sigma-Aldrich, #ABE290, 1:50)<br>Stat1 polyclonal antibody (Cell Signaling Technology, #9172, 1:25)<br>Stat3 rabbit monoclonal antibody (Cell Signaling Technology, Clone D3Z2G, #12640, 1:25)<br>Normal Rabbit IgG (Cell Signaling Technology, #2729S, 1:500)<br>Ki-67 recombinant rabbit monoclonal antibody (ThermoFisher Scientific, clone SP6, #MA5-14520, 1:100)<br>Anti-CD3 antibody (Abcam, Clone SP7, #ab16669, 1:100)<br>Horse anti-rabbit IgG antibody (H+L), Biotinylated (Vector Laboratories, BA-1100-1.5, 1:200) |
| Validation | https://www.biolegend.com/nl-be/quality/quality-assurance-certificates<br>https://www.genetex.com/Info/guarantee<br>https://www.cellsignal.com/about-us/cst-antibody-validation-principles<br>https://www.sigmaaldrich.com/US/en/life-science/quality-and-regulatory-management<br>https://www.thermofisher.com/us/en/home/life-science/antibodies/invitrogen-antibody-validation<br>https://www.abcam.com/en-us/stories/articles/biophysical-quality<br>https://vectorlabs.com/immunohistochemistry/ |

# Eukaryotic cell lines

Policy information about cell lines and Sex and Gender in Research

| | |
|---|---|
| Cell line source(s) | HEK-293T cells were purchased from ATCC. |
| Authentication | Authentication was assessed by short tandem repeats (STR) profiling. |
| Mycoplasma contamination | Cells were tested negative for mycoplasma contamination. |
| Commonly misidentified lines<br>(See ICLAC register) | No commonly misidentified cell lines were used in this study. |

# Animals and other research organisms

Policy information about studies involving animals; ARRIVE guidelines recommended for reporting animal research, and Sex and Gender in Research

| | |
|---|---|
| Laboratory animals | Lgr5-EGFP-IRES-CreERT2 mice (Jackson Laboratory, Strain # 008875) were backcrossed with C57BL/6J mice for over 20 generations to ensure a genetically identical background. |
| Wild animals | This study did not involve wild animals. |

| Reporting on sex | Both male and female mice were used across experiments. |
|---|---|
| Field-collected samples | Specific pathogen-free (SPF) and germ-free (GF) housing conditions were employed. Routine serological tests were conducted to exclude the presence of specific pathogens. The GF mice were monitored monthly using 16S sequencing. |
| Ethics oversight | All experiments were conducted with approval from the Institutional Animal Care and Use Committee (IACUC) at Baylor College of Medicine (protocol number AN-6775). |

Note that full information on the approval of the study protocol must also be provided in the manuscript.

# Plants

| Seed stocks | Not applicable |
|---|---|
| Novel plant genotypes | Not applicable |
| Authentication | Not applicable |

# Flow Cytometry

## Plots

Confirm that:

☒ The axis labels state the marker and fluorochrome used (e.g. CD4-FITC).

☒ The axis scales are clearly visible. Include numbers along axes only for bottom left plot of group (a 'group' is an analysis of identical markers).

☒ All plots are contour plots with outliers or pseudocolor plots.

☒ A numerical value for number of cells or percentage (with statistics) is provided.

## Methodology

| Sample preparation | After dissociation with EDTA and TrypLE, the colonic epithelial cells were in single cell suspension and were incubated with anti-mouse CD326 (EpCAM) antibody. The cells were then washed, filtered, and sorted based on their expression of EpCAM and GFP: stem cells were identified as EpCAM+/GFP+ and differentiated cells were identified as EpCAM+/GFP-.

Intraepithelial lymphocytes (IELs) were enriched using 40%/80% discontinuous Percoll gradient centrifugation. For surface marker staining, IELs were first treated with Zombie Aqua™ Fixable Viability Kit for live/dead staining. Following this, cells were blocked with anti-mouse CD16/CD32 to reduce non-specific binding. Surface markers were then stained using fluorochrome-conjugated antibodies. For intracellular staining, cells were fixed using the Foxp3/Transcription Factor Staining Buffer Set according to the manufacturer's instructions. After fixation, the cells were stained with cytokine-specific antibodies. |
|---|---|
| Instrument | BD Aria Fusion cell sorter and BD Symphony A5 flow cytometer |
| Software | FlowJo software |
| Cell population abundance | At least 100000 cells were acquired for each condition. |
| Gating strategy | Gating strategy to identify individual cell populations are available in Supplementary Fig. 2. |

☒ Tick this box to confirm that a figure exemplifying the gating strategy is provided in the Supplementary Information.

