## [Peer Review File · Nature Microbiology]

Weaning drives microbiome-mediated epigenetic regulation to shape immune memory in mice

Corresponding Author: Dr Lanlan Shen

Version 0:

Reviewer comments:

Reviewer #1

(Remarks to the Author)

Yang and colleagues report here that the intestinal microbiota shapes the epigenetic code of intestinal epithelial stem cells (ISC). Using genome-wide bisulfite sequencing, they show that methylation of genes coding for MHC class II molecules and associated proteins is increased in germfree animals, and thus expression of these genes is decreased. Colonizing germfree mice with microbiota at different ages, they show that the effect of the microbiota on methylation decreases with time. Using organoid cultures, they demonstrate that IFN γ is sufficient to induce these effects on Cd74, coding for the MHC II-associated invariant chain, and that the modifications in methylation are stable for at least 48h or 3 cell passages in culture. Finally, they report that the treatment of mice with penicillin from birth to adulthood leads to increased methylation of these genes, thus decreased expression, and increased susceptibility to DSS-induced colitis, in agreement with earlier data showing increased susceptibility of MHC II-deficient mice.

The study is potentially interesting in the context of the early life impact of microbiota on the immune system into adulthood, as the mechanisms involved in that phenomenon remain relatively unclear. Nevertheless, this study does not address this point, as claimed to be addressed in the introduction, but rather shows that the microbiota regulates expression of immune genes, probably through IFN γ . This latter point also requires more data, as only specifically addressed *in vitro*. Furthermore, immunological memory at the epigenetic level is only shown *in vitro* for 48h.

Specific comments:

1. Line 49: the authors should cite the original papers, and not reviews, when discussing the weaning reaction, in order to avoid confusion about the phenomenon that they really are addressing in the study. For example, in figure 1, the comparison of SPF and GF mice does not address the weaning reaction or the early life impact of the microbiota on the immune system.
2. Figures 1, 2, S3: it has been reported (Koo and Clevers, *Gastroenterology* 2014) that expression of GFP in the Lgr5-GFP reporter mice is mosaic due to random inactivation of the knock-in allele. Thus, LGR5+ ISCs are also present in the GFPneg cells, which is a problem when comparing ISC to differentiated epithelial cells in figure 2 and S3, particularly when defining common DMRs. CD24 staining would help to exclude GFPneg ISCs.
3. Figure 4 shows an apparently linear time-dependent effect of the microbiota on methylation. As presented, the data suggest instead that colonizing germfree mice during weaning, during early adulthood or as fully adult mice has decreasing effects. To make the point that it is the age at which the mice are colonized, rather than the duration of colonization that matters, the following experiment should be done: the "weaning" group, colonized for 12 weeks before assessment of methylation and gene expression, should be compared to adult mice colonized for 12 weeks before assessment.
4. Figure 5 shows that IFN γ alters the methylation and expression of Cd74 in organoid cultures. It is indeed expected that IFN γ is involved in the regulation of MHC II molecules and associated proteins, such as CIITA and CD74. But the key experiment has to be done *in vivo* in mice treated with antibodies neutralizing IFN γ at different age of the mice, and all MHC II-related molecules have to be monitored (MHC II, CD74, CIITA, ...). Furthermore, the authors mention "training immunological memory" in the title, but the only actual memory reported in this study is epigenetic stability for 48h *in vitro*.
5. Figure 6 is largely irrelevant in this study as it is expected that a loss in microbiota (here Gram+ bacteria) leads to a reduction in immune response, here via IFN γ , and thus a loss in MHC II expression and associated molecules.

Reviewer #2

(Remarks to the Author)

Yang et al report in their manuscript entitled “Weaning driven gut microbiome shapes intestinal stem cell epigenetics to train immunological memory” on microbiota induced epigenetic alterations in the colonic intestinal epithelium that are induced during an early life window and partly conferred by IFN-gamma. They among others alter epithelial CD74 and MHCII expression and susceptibility to DSS colitis. The topic is novel, relevant and highly interesting. The experiments are generally well-conducted and the data are clearly presented. However, some conclusions drawn by the authors are not supported by the data and the focus on CD74 and MHCII expression among the large number of epigenetically altered genes is not convincingly explained. A global view on genes influenced by epigenetic changes would have been very interesting. And this in particular of genes involved in immune responses and inflammation, central e.g. to the DSS colitis model used in Fig. 6.

Major points

1. In the abstract a causal relationship between vanishing maternal antibodies and the weaning reaction is implicated. This has not been shown. Instead, the alteration of the microbiota at weaning was suggested to cause the weaning reaction. In the introduction, this link is correctly described (I49).
2. Line 67 and 68 suggest that the loss of MHCII expression is associated with enhanced colitis susceptibility. However, this is a matter of debate (Ozkul et al., *Genome Med.*, 2020 versus Jin et al., *Sci Rep*, 2017).
3. L77: “...transitioned from SPF to germ-free conditions...” This transition has not been studied. Instead, GF bred mice were compared with SPF mice or conventionalized GF (exGF) mice.
4. The highest MHCII expression is found in the mid small intestine, much higher than in the colon (Stephens et al., *Cell Rep.*, 2021). It is not clear why the morphological changes that are observed in the colon but less so in the small intestine should prompt to focus on MHCII expression in the colon rather than the small intestine.
5. L168-169: Only CD74 is shown in Fig. S5 so the text should not refer to “MHCII-related genes” but only CD74. Alternatively, show the data to support the statement.
6. L105 and Fig. S3b suggest that only a fraction of DMRs is shared between ISCs and differentiated cells, however, Fig. 1CD, Fig. 2a and Fig. S3E suggest little difference between those two cell fractions. Please explain.
7. It is unclear why this paper focusses on MHCII and CD74 expression given that a vast number of genes (Fig. S3A and B) is affected by the epigenetic changes and differentially expressed (I107-113).
8. Doesn't the fact that the microbiota is much denser in the colon as compared to small intestine (that also shows differential MHCII and CD74 expression) suggest that the effect upstream of the epigenetic alterations is indirect? Is IFNg responsible for this indirect effect?
9. It is unclear how old the mice are in Fig. 1 (not indicated in figure or legend nor results text). These mice apparently show no difference in overall methylation but the adult mice in Fig. S2 do.
10. L181ff. I disagree with the interpretation. GMT during early life also means a much longer presence of the microbiota when all groups are analysed at the same age. The length of exposure (and not the age window) may therefore be important.
11. Fig. 5D: Again, I disagree with the interpretation. Previous IFNg treatment also means prolonged IFNg exposure which by itself could explain the stronger response. I believe one should do an experiment in which IFNg exposure comes first, but another second stimulus induces another target gene that is under IFNg-mediated epigenetic control.
12. Fig. 6B. The IELs are insufficiently characterized to call them IELs (see Fig. S7). Staining for TCRa/b or TCRg/d plus CD8aa would be needed to make that point. Please also consider TCRg/d nIELs as source of the IFNg.
13. L237/238: I agree with the comment that CD74 KO mice are more susceptible to DSS colitis, but the interpretation of the results reported in ref 43 on DSS exposed IEC-specific MHCII KO mice is difficult. No more weight loss and no shorter colon length reported, both typical markers of DSS colitis.
14. Fig. 1B: Please include FACS analysis of Lgr5-GFP allele negative mice as control.
15. Fig. 5C: The IFNg effect is relatively weak (and the addition of the DAC reagent does not add any information). Have other cytokines been tested? What might also have a similar effect?
16. Fig5D: Does not show that IFNg acts via an epigenetic change. The difference could also be explained by prolonged IFNg exposure.
17. Fig. 5D vs E: Isn't the 48h group in Fig. 5E very similar to the pretreated group in Fig.5D: But in 5E, the expression does not differ from the naïve (control) group. Please explain.
18. Fig. 6B: Please show complete 16S rRNA gene analysis. Lactobacilli, Clostridium (?), SFB and Bacteroides/Prevotella (also Fig. S7)) may not be representative for Gram+/- bacteria. Also, all taxa shown have very very low abundance in a substantial number of mice. A quantitative bacterial PCR would help to assess the effect of the LDP regimen.
19. Fig. 6D: p values are only shown for the comparison between the SPF and LDP group. Is none of the LDP GF comparisons significantly different?
20. Fig. 6E: the MHCII staining is hard to interpret. The gate is placed in a very artificial way.
21. Fig. 6F-H: I believe the authors imply that the effect of LDP on DSS susceptibility is MHCII dependent (based on the ref43). However, this is not formally shown. The enhanced susceptibility LPS could be due to other epigenetically modified genes or independent effects (e.g. of immune cell homing or IEC proliferation etc.). Please repeat the experiment with IEC MHCII KO mice to show that the difference is gone.
22. The title is misleading. I don't think DSS colitis can formally be interpreted as adaptive immunity model.

Minor points

1. L158: 15 week old mice are not young adult mice but adult mice.
2. The work from Dana Philpott (Tsang et al., *Gut microbes*, 2022) that describes expression differences between GF and SPF mice should be cited.
3. Fig. 6B. Please show quantitative data.

Reviewer #3

(Remarks to the Author)

The study entitled "Weaning-Driven Gut Microbiome Shapes Intestinal Stem Cell Epigenetics to Train Immunological Memory" by Yang et al, presents interesting findings on the potential role of the presence (or absence) of the gut microbiome in shaping intestinal stem cell (ISC) epigenetics during weaning. However, there are several areas where the study could be strengthened, both in terms of experimental design and data interpretation.

Comments:

- The use of germ-free (GF) mice is appropriate for this study, but it is unclear if microbial products were also excluded (or tested to confirm they were not present), which could lead to activation of responses, and could occur even in the absence of live bacteria (this links to a comment on the organoid studies). The descriptive nature of the findings is significant, but the lack of further mechanistic exploration into what specific components of the microbiota are critical limits the depth of this study. Furthermore, there is no clear identification of which microbiota components (microbes/MAMPs/metabolites etc.) are driving the observed effects. It would be key to address whether microbial products might have been tested alongside live bacterial exposures to distinguish their individual impacts.
- The study heavily relies on the presence or absence of microbiota, but it does not provide any detailed microbiota profiling data (only qPCR targeted a very limited number of 'taxa' and on a small number of samples). This reductionist approach limits the scope of the study and findings. Microbiota composition could significantly influence the epigenetic regulation of ISCs. Profiling at different stages of the experiment would provide insights into how specific microbial populations contribute to the observed effects. Indeed, the article focuses largely on epigenetic changes, but it lacks a strong connection to microbial mechanisms. While the epigenetic changes in response to weaning and microbiota are interesting, the lack of sequencing data limits the ability to link these changes to specific microbial species or community structures. A deeper microbiome analysis using shotgun metagenomics, and potentially metabolomics, would significantly add robustness to the conclusions.
- The authors claim that microbial exposure during early life impacts ISC epigenetics, but they do not explore how stable microbial colonization is achieved. What microbial species are colonizing, and how stable is the colonization over time (linked to point above about lack of detailed microbiome profiling)? Data on microbial stability or shifts would strengthen the paper's claims about long-term impacts.
- The authors suggest that juvenile gut microbiota transfer (GMT) is more effective in restoring epigenetic regulation, but they do not provide sufficient data on how the timing of microbial introduction affects colonization or stability. Further data on the longitudinal stability of microbiota following transfer would help elucidate whether the effects are transient or lasting.
- Moreover, while low-dose antibiotics were tested, the lack of detailed analysis on how these interventions affect later microbiota dynamics and subsequent epigenetic reprogramming limits the interpretation of the data. The authors focus on the 4-6-week window for epigenetic changes but fail to explore bacterial knockdown timelines or bacterial regrowth post-intervention.
- In the GF organoid experiments, the authors mention that LPS did not induce responses. However, they do not test other bacterial ligands and metabolites that might have elicited a different response, such as peptidoglycan, other types of LPS like Bacteroides-derived LPS, and SCFAs, which may have different effects.
- The use of the term "dysbiosis" is over-simplified in this context. The authors should define the term more rigorously, particularly when referencing such complex microbiota-host interactions.
- Several studies have demonstrated impacts of the microbiota, including during pre- and post-weaning within the murine gastrointestinal track with response to epithelial responses and these studies are currently not referenced. The authors should cite these works to provide context and acknowledge existing literature.
- As highlighted above - introducing specific bacterial strains or microbial consortia into GF mice could help identify key microbial players and confirm the protective or detrimental effects observed. Additionally, using an antibiotic cocktail to induce near GF conditions (would also represent a more 'normally-programmed' host than GF mice) and then reintroducing specific bacteria could provide more detailed insights into which microbes are crucial for ISC epigenetic regulation. This could be coupled with in-depth sequencing (and metabolomics) on microbial populations during the key stages of intervention, as this would reveal microbial dynamics and potential drivers of the observed epigenetic changes.

This study provides valuable data on the microbiome's role in shaping ISC epigenetics, particularly during the weaning period. However, the lack of microbiota profiling, mechanistic detail, and consideration of microbial products weakens the conclusions. Incorporating microbial profiling and addressing the concerns mentioned above would significantly strengthen the manuscript.

Decision Letter:

22nd October 2024

Dear Dr Shen,

Thank you for your patience while your manuscript "Weaning-Driven Gut Microbiome Shapes Intestinal Stem Cell Epigenetics to Train Immunological Memory" was under peer review at Nature Microbiology. It has now been seen by our referees, whose expertise and comments you will find at the end of this email. In the light of their advice, we have decided that we cannot offer to publish your manuscript in Nature Microbiology.

From the reports, you will see that while they find your work of some potential interest, the referees raise concerns regarding the experimental design, interpretation of data, over-exaggerated claims, and a general lack of microbiological insights. Unfortunately, these criticisms are sufficiently important as to preclude publication of your work in Nature Microbiology.

Although we cannot offer to publish your manuscript, I suggest that you consider Communications Biology as a suitable venue for this work. To transfer your manuscript, please use our manuscript transfer portal. You will not have to re-supply manuscript metadata and files, unless you wish to make modifications. For more information, please see our http://www.nature.com/authors/author_resources/transfer_manuscripts.html?WT.mc_id=EMI_NPG_1511_AUTHORTRANSF&WT.ec_id=AUTHOR manuscript transfer FAQ page.

I am sorry that we cannot be more positive on this occasion, but hope that you find the referees' comments helpful when

preparing your paper for resubmission elsewhere.

Yours sincerely,

Reviewer Expertise:

Referee #1: Microbiota, methylation, immunity

Referee #2: Intestinal stem cells and development

Referee #3: Early age microbiota

Reviewers Comments:

Reviewer #1 (Remarks to the Author):

Yang and colleagues report here that the intestinal microbiota shapes the epigenetic code of intestinal epithelial stem cells (ISC). Using genome-wide bisulfite sequencing, they show that methylation of genes coding for MHC class II molecules and associated proteins is increased in germfree animals, and thus expression of these genes is decreased. Colonizing germfree mice with microbiota at different ages, they show that the effect of the microbiota on methylation decreases with time. Using organoid cultures, they demonstrate that IFN γ is sufficient to induce these effects on Cd74, coding for the MHC II-associated invariant chain, and that the modifications in methylation are stable for at least 48h or 3 cell passages in culture. Finally, they report that the treatment of mice with penicillin from birth to adulthood leads to increased methylation of these genes, thus decreased expression, and increased susceptibility to DSS-induced colitis, in agreement with earlier data showing increased susceptibility of MHC II-deficient mice.

The study is potentially interesting in the context of the early life impact of microbiota on the immune system into adulthood, as the mechanisms involved in that phenomenon remain relatively unclear. Nevertheless, this study does not address this point, as claimed to be addressed in the introduction, but rather shows that the microbiota regulates expression of immune genes, probably through IFN γ . This latter point also requires more data, as only specifically addressed in vitro. Furthermore, immunological memory at the epigenetic level is only shown in vitro for 48h.

Specific comments:

1. Line 49: the authors should cite the original papers, and not reviews, when discussing the weaning reaction, in order to avoid confusion about the phenomenon that they really are addressing in the study. For example, in figure 1, the comparison of SPF and GF mice does not address the weaning reaction or the early life impact of the microbiota on the immune system.
2. Figures 1, 2, S3: it has been reported (Koo and Clevers, Gastroenterology 2014) that expression of GFP in the Lgr5-GFP reporter mice is mosaic due to random inactivation of the knock-in allele. Thus, LGR5+ ISCs are also present in the GFPneg cells, which is a problem when comparing ISC to differentiated epithelial cells in figure 2 and S3, particularly when defining common DMRs. CD24 staining would help to exclude GFPneg ISCs.
3. Figure 4 shows an apparently linear time-dependent effect of the microbiota on methylation. As presented, the data suggest instead that colonizing germfree mice during weaning, during early adulthood or as fully adult mice has decreasing effects. To make the point that it is the age at which the mice are colonized, rather than the duration of colonization that matters, the following experiment should be done: the "weaning" group, colonized for 12 weeks before assessment of methylation and gene expression, should be compared to adult mice colonized for 12 weeks before assessment.
4. Figure 5 shows that IFN γ alters the methylation and expression of Cd74 in organoid cultures. It is indeed expected that IFN γ is involved in the regulation of MHC II molecules and associated proteins, such as CIITA and CD74. But the key experiment has to be done in vivo in mice treated with antibodies neutralizing IFN γ at different age of the mice, and all MHC II-related molecules have to be monitored (MHC II, CD74, CIITA, ...). Furthermore, the authors mention "training immunological memory" in the title, but the only actual memory reported in this study is epigenetic stability for 48h in vitro.
5. Figure 6 is largely irrelevant in this study as it is expected that a loss in microbiota (here Gram+ bacteria) leads to a reduction in immune response, here via IFN γ , and thus a loss in MHC II expression and associated molecules.

Reviewer #2 (Remarks to the Author):

Yang et al report in their manuscript entitled "Weaning driven gut microbiome shapes intestinal stem cell epigenetics to train immunological memory" on microbiota induced epigenetic alterations in the colonic intestinal epithelium that are induced during an early life window and partly conferred by IFN- γ . They among others alter epithelial CD74 and MHCII expression and susceptibility to DSS colitis. The topic is novel, relevant and highly interesting. The experiments are generally well-conducted and the data are clearly presented. However, some conclusions drawn by the authors are not supported by the data and the focus on CD74 and MHCII expression among the large number of epigenetically altered genes is not convincingly explained. A global view on genes influenced by epigenetic changes would have been very interesting. And this in particular of genes involved in immune responses and inflammation, central e.g. to the DSS colitis model used in Fig. 6.

Major points

1. In the abstract a causal relationship between vanishing maternal antibodies and the weaning reaction is implicated. This has not been shown. Instead, the alteration of the microbiota at weaning was suggested to cause the weaning reaction. In the introduction, this link is correctly described (I49).
2. Line 67 and 68 suggest that the loss of MHCII expression is associated with enhanced colitis susceptibility. However, this is a matter of debate (Ozkul et al., *Genome Med.*, 2020 versus Jin et al., *Sci Rep*, 2017).
3. L77: "...transitioned from SPF to germ-free conditions..." This transition has not been studied. Instead, GF bred mice were compared with SPF mice or conventionalized GF (exGF) mice.
4. The highest MHCII expression is found in the mid small intestine, much higher than in the colon (Stephens et al., *Cell Rep.*, 2021). It is not clear why the morphological changes that are observed in the colon but less so in the small intestine should prompt to focus on MHCII expression in the colon rather than the small intestine.
5. L168-169: Only CD74 is shown in Fig. S5 so the text should not refer to "MHCII-related genes" but only CD74. Alternatively, show the data to support the statement.
6. L105 and Fig. S3b suggest that only a fraction of DMRs is shared between ISCs and differentiated cells, however, Fig. 1CD, Fig. 2a and Fig. S3E suggest little difference between those two cell fractions. Please explain.
7. It is unclear why this paper focusses on MHCII and CD74 expression given that a vast number of genes (Fig. S3A and B) is affected by the epigenetic changes and differentially expressed (I107-113).
8. Doesn't the fact that the microbiota is much denser in the colon as compared to small intestine (that also shows differential MHCII and CD74 expression) suggest that the effect upstream of the epigenetic alterations is indirect? Is IFN γ responsible for this indirect effect?
9. It is unclear how old the mice are in Fig. 1 (not indicated in figure or legend nor results text). These mice apparently show no difference in overall methylation but the adult mice in Fig. S2 do.
10. L181ff. I disagree with the interpretation. GMT during early life also means a much longer presence of the microbiota when all groups are analysed at the same age. The length of exposure (and not the age window) may therefore be important.
11. Fig. 5D: Again, I disagree with the interpretation. Previous IFN γ treatment also means prolonged IFN γ exposure which by itself could explain the stronger response. I believe one should do an experiment in which IFN γ exposure comes first, but another second stimulus induces another target gene that is under IFN γ -mediated epigenetic control.
12. Fig. 6B. The IELs are insufficiently characterized to call them IELs (see Fig. S7). Staining for TCR α /b or TCR γ /d plus CD8aa would be needed to make that point. Please also consider TCR γ /d nIELs as source of the IFN γ .
13. L237/238: I agree with the comment that CD74 KO mice are more susceptible to DSS colitis, but the interpretation of the results reported in ref 43 on DSS exposed IEC-specific MHCII KO mice is difficult. No more weight loss and no shorter colon length reported, both typical markers of DSS colitis.
14. Fig. 1B: Please include FACS analysis of Lgr5-GFP allele negative mice as control.
15. Fig. 5C: The IFN γ effect is relatively weak (and the addition of the DAC reagent does not add any information). Have other cytokines been tested? What might also have a similar effect?
16. Fig5D: Does not show that IFN γ acts via an epigenetic change. The difference could also be explained by prolonged IFN γ exposure.
17. Fig. 5D vs E: Isn't the 48h group in Fig. 5E very similar to the pretreated group in Fig.5D: But in 5E, the expression does not differ from the naïve (control) group. Please explain.
18. Fig. 6B: Please show complete 16S rRNA gene analysis. Lactobacilli, Clostridium (?), SFB and Bacteroides/Prevotella (also Fig. S7)) may not be representative for Gram+/- bacteria. Also, all taxa shown have very very low abundance in a substantial number of mice. A quantitative bacterial PCR would help to assess the effect of the LDP regimen.
19. Fig. 6D: p values are only shown for the comparison between the SPF and LDP group. Is none of the LDP GF comparisons significantly different?
20. Fig. 6E: the MHCII staining is hard to interpret. The gate is placed in a very artificial way.
21. Fig. 6F-H: I believe the authors imply that the effect of LDP on DSS susceptibility is MHCII dependent (based on the ref43). However, this is not formally shown. The enhanced susceptibility LPS could be due to other epigenetically modified genes or independent effects (e.g. of immune cell homing or IEC proliferation etc.). Please repeat the experiment with IEC MHCII KO mice to show that the difference is gone.
22. The title is misleading. I don't think DSS colitis can formally be interpreted as adaptive immunity model.

Minor points

1. L158: 15 week old mice are not young adult mice but adult mice.
2. The work from Dana Philpott (Tsang et al., *Gut microbes*, 2022) that describes expression differences between GF and SPF mice should be cited.
3. Fig. 6B. Please show quantitative data.

Reviewer #3 (Remarks to the Author):

The study entitled "Weaning-Driven Gut Microbiome Shapes Intestinal Stem Cell Epigenetics to Train Immunological Memory" by Yang et al, presents interesting findings on the potential role of the presence (or absence) of the gut microbiome in shaping intestinal stem cell (ISC) epigenetics during weaning. However, there are several areas where the study could be strengthened, both in terms of experimental design and data interpretation.

Comments:

- The use of germ-free (GF) mice is appropriate for this study, but it is unclear if microbial products were also excluded (or tested to confirm they were not present), which could lead to activation of responses, and could occur even in the absence of live bacteria (this links to a comment on the organoid studies). The descriptive nature of the findings is significant, but the lack of

further mechanistic exploration into what specific components of the microbiota are critical limits the depth of this study. Furthermore, there is no clear identification of which microbiota components (microbes/MAMPs/metabolites etc.) are driving the observed effects. It would be key to address whether microbial products might have been tested alongside live bacterial exposures to distinguish their individual impacts.

- The study heavily relies on the presence or absence of microbiota, but it does not provide any detailed microbiota profiling data (only qPCR targeted a very limited number of 'taxa' and on a small number of samples). This reductionist approach limits the scope of the study and findings. Microbiota composition could significantly influence the epigenetic regulation of ISCs. Profiling at different stages of the experiment would provide insights into how specific microbial populations contribute to the observed effects. Indeed, the article focuses largely on epigenetic changes, but it lacks a strong connection to microbial mechanisms. While the epigenetic changes in response to weaning and microbiota are interesting, the lack of sequencing data limits the ability to link these changes to specific microbial species or community structures. A deeper microbiome analysis using shotgun metagenomics, and potentially metabolomics, would significantly add robustness to the conclusions.
 - The authors claim that microbial exposure during early life impacts ISC epigenetics, but they do not explore how stable microbial colonization is achieved. What microbial species are colonizing, and how stable is the colonization over time (linked to point above about lack of detailed microbiome profiling)? Data on microbial stability or shifts would strengthen the paper's claims about long-term impacts.
 - The authors suggest that juvenile gut microbiota transfer (GMT) is more effective in restoring epigenetic regulation, but they do not provide sufficient data on how the timing of microbial introduction affects colonization or stability. Further data on the longitudinal stability of microbiota following transfer would help elucidate whether the effects are transient or lasting.
 - Moreover, while low-dose antibiotics were tested, the lack of detailed analysis on how these interventions affect later microbiota dynamics and subsequent epigenetic reprogramming limits the interpretation of the data. The authors focus on the 4-6-week window for epigenetic changes but fail to explore bacterial knockdown timelines or bacterial regrowth post-intervention.
 - In the GF organoid experiments, the authors mention that LPS did not induce responses. However, they do not test other bacterial ligands and metabolites that might have elicited a different response, such as peptidoglycan, other types of LPS like Bacteroides-derived LPS, and SCFAs, which may have different effects.
 - The use of the term "dysbiosis" is over-simplified in this context. The authors should define the term more rigorously, particularly when referencing such complex microbiota-host interactions.
 - Several studies have demonstrated impacts of the microbiota, including during pre- and post-weaning within the murine gastrointestinal track with response to epithelial responses and these studies are currently not referenced. The authors should cite these works to provide context and acknowledge existing literature.
 - As highlighted above - introducing specific bacterial strains or microbial consortia into GF mice could help identify key microbial players and confirm the protective or detrimental effects observed. Additionally, using an antibiotic cocktail to induce near GF conditions (would also represent a more 'normally-programmed' host than GF mice) and then reintroducing specific bacteria could provide more detailed insights into which microbes are crucial for ISC epigenetic regulation. This could be coupled with in-depth sequencing (and metabolomics) on microbial populations during the key stages of intervention, as this would reveal microbial dynamics and potential drivers of the observed epigenetic changes.
- This study provides valuable data on the microbiome's role in shaping ISC epigenetics, particularly during the weaning period. However, the lack of microbiota profiling, mechanistic detail, and consideration of microbial products weakens the conclusions. Incorporating microbial profiling and addressing the concerns mentioned above would significantly strengthen the manuscript.

Version 1:

Decision Letter:

20th November 2025

Dear Dr Shen,

Thank you for your letter asking us to reconsider our decision on your Article entitled "Weaning-driven gut microbiome reprograms intestinal stem cell epigenetics to establish epithelial-intrinsic immune memory". After careful consideration of your point-by-point response and in light of new experimental data added, we have decided that we would be willing to consider a revised version of your manuscript.

Please resubmit a revised version of your manuscript (please ensure that the text file has line numbers) Along with the point-by-point response to all of the concerns raised by the referees, in each case describing what changes have been made to the manuscript or, alternatively, if no action has been taken, providing a compelling argument for why that is the case. If we feel that a substantial attempt has been made to address the referees' comments, this response will be sent back to the referees - along with the revised manuscript - so that they can judge whether their concerns have been addressed satisfactorily or otherwise.

- ensure it complies with our format requirements for Letters as set out in our guide to authors at www.nature.com/nmicrobiol/authors/index.html

- state in a cover note the length of the text, methods and legends; the number of references and the number of display items.

Please ensure that all correspondence is marked with your Nature Microbiology reference number in the subject line.

Please use the following link to submit your revised manuscript:

Link Redacted

We hope to receive your revised paper within four weeks. If you cannot send it within this time, please let us know so that we can close your file. In this event, we will still be happy to reconsider your paper at a later date so long as nothing similar has been accepted for publication at Nature Microbiology or published elsewhere in the meantime. Should you miss the four-week deadline and your paper is eventually published, the received date will be that of the revised, not the original, version.

I would appreciate it if you could tell me if you think you will be able to submit a revised manuscript, and also the likely timescale.

I look forward to hearing from you soon.

Best wishes,

Version 2:

Reviewer comments:

Reviewer #1

(Remarks to the Author)

The authors have addressed my concerns and I thank them for their work.

Reviewer #2

(Remarks to the Author)

The authors have adequately and in great detail responded to the reviewers questions and performed substantial new experiments (e.g. microbiota metagenomics, IFN γ inhibition in vivo). The revised manuscript presents a comprehensive study on the effect of the weaning microbiota induced IFN γ on the epigenetic control of the inducible MHCII gene expression with long-term consequences on disease susceptibility. With the changes made, I think the manuscript should be accepted for publication.
Mathias Hornef

Reviewer #3

(Remarks to the Author)

Thank you to the authors for the substantial revisions and for incorporating detailed microbiome analyses. These additions go a long way toward addressing the concerns raised in the previous submission. However, the microbiome component of the study remains largely descriptive, and the new analyses, while informative, still rely primarily on correlations between shotgun metagenomic profiles and GO-term-based functional predictions. This does not provide a direct mechanistic link or in vivo evidence demonstrating microbial causality.

Given the expectations for Nature Microbiology, it would strengthen the manuscript considerably to include more robust microbiological work that directly connects specific microbial taxa or functions to host phenotypes or mechanisms. Although the revised manuscript is significantly improved overall and the host profiling is very detailed, the absence of mechanistic microbial validation may mean that another (still excellent) journal would be a more suitable fit.

Decision Letter:

15th January 2026

Dear Lanlan,

I hope you are having a good start to the year. Thank you so much for your patience while your manuscript was under consideration. We have heard back from the original referees and we think that the major concerns of the referees have been addressed and therefore, we would be happy to publish your paper, in principle, in Nature Microbiology. It's been a long journey and I am glad that it is ending on a happy note.

However, the official status of your manuscript, for now, will be 'under revision' because we noted that the figures are not provided separately and there are no Extended Data figures. Since the figures will undergo extensive checks we need them to be in separate files at this stage and we want you to submit 10 supplementary figures as Extended Data figures. Once that is done, I will officially send you an 'accepted in principle' decision which will be followed by an author checklist to ensure that the manuscript adheres to our journalistic style.

Below you will find some additional information (from our template email) that will help you prepare the final manuscript. The link to resubmit the revised version is also there.

If you have any questions or concerns, please feel free to write back.

All the best!

Atin

Thank you for your patience while your manuscript "Weaning-driven gut microbiome reprograms intestinal stem cell epigenetics to establish epithelial-intrinsic immune memory" was under peer-review at Nature Microbiology. It has now been seen by 3 referees, whose comments you will find at the end of this email. We are very interested in the possibility of publishing your study in Nature Microbiology, but would like to consider your response to these concerns in the form of a revised manuscript before we make a final decision on publication.

If you have not done so already please begin to revise your manuscript so that it conforms to our Article format instructions at <http://www.nature.com/nmicrobiol/info/final-submission/>

The usual length limit for a Nature Microbiology Article is six display items (figures or tables) and 3,000 words. We have some flexibility, and can allow a revised manuscript at 3,500 words, but please consider this a firm upper limit. There is a trade-off of ~250 words per display item, so if you need more space, you could move a Figure or Table to Supplementary Information.

Some reduction could be achieved by focusing any introductory material and moving it to the start of your opening 'bold' paragraph, whose function is to outline the background to your work, describe in a sentence your new observations, and explain your main conclusions. The discussion should also be limited. Methods should be described in a separate section following the discussion, we do not place a word limit on Methods.

Nature Microbiology titles should give a sense of the main new findings of a manuscript, and should not contain punctuation. Please keep in mind that we strongly discourage active verbs in titles, and that they should ideally fit within 90 characters each (including spaces).

Please include a data availability statement as a separate section after Methods but before references, under the heading "Data Availability". This section should inform readers about the availability of the data used to support the conclusions of your study. This information includes accession codes to public repositories (data banks for protein, DNA or RNA sequences, microarray, proteomics data etc...), references to source data published alongside the paper, unique identifiers such as URLs to data repository entries, or data set DOIs, and any other statement about data availability. At a minimum, you should include the following statement: "The data that support the findings of this study are available from the corresponding author upon request", mentioning any restrictions on availability. If DOIs are provided, we also strongly encourage including these in the Reference list (authors, title, publisher (repository name), identifier, year). For more guidance on how to write this section please see: <http://www.nature.com/authors/policies/data/data-availability-statements-data-citations.pdf>

To improve the accessibility of your paper to readers from other research areas, please pay particular attention to the wording of the paper's opening bold paragraph, which serves both as an introduction and as a brief, non-technical summary in about 150 words. If, however, you require one or two extra sentences to explain your work clearly, please include them even if the paragraph is over-length as a result. The opening paragraph should not contain references. Because scientists from other sub-disciplines will be interested in your results and their implications, it is important to explain essential but specialised terms concisely. We suggest you show your summary paragraph to colleagues in other fields to uncover any problematic concepts.

If your paper is accepted for publication, we will edit your display items electronically so they conform to our house style and will reproduce clearly in print. If necessary, we will re-size figures to fit single or double column width. If your figures contain several parts, the parts should form a neat rectangle when assembled. Choosing the right electronic format at this stage will speed up the processing of your paper and give the best possible results in print. We would like the figures to be supplied as vector files - EPS, PDF, AI or postscript (PS) file formats (not raster or bitmap files), preferably generated with vector-graphics software (Adobe Illustrator for example). Please try to ensure that all figures are non-flattened and fully editable. All images should be at least 300 dpi resolution (when figures are scaled to approximately the size that they are to be printed at) and in RGB colour format. Please do not submit Jpeg or flattened TIFF files. Please see also 'Guidelines for Electronic Submission of Figures' at the end of this letter for further detail.

Figure legends must provide a brief description of the figure and the symbols used, within 350 words, including definitions of any error bars employed in the figures.

EXTENDED DATA FIGURES

Please include a statement before the acknowledgements naming the author to whom correspondence and requests for materials should be addressed.

Finally, we require authors to include a statement of their individual contributions to the paper -- such as experimental work, project planning, data analysis, etc. -- immediately after the acknowledgements. The statement should be short, and refer to authors by their initials. For details please see the Authorship section of our joint Editorial policies at http://www.nature.com/authors/editorial_policies/authorship.html

* include a point-by-point response to any editorial suggestions and to our referees. Please include your response to the editorial suggestions in your cover letter, and please upload your response to the referees as a separate document.

* ensure it complies with our format requirements for Letters as set out in our guide to authors at www.nature.com/nmicrobiol/info/gta/

* state in a cover note the length of the text, methods and legends; the number of references; number and estimated final size of figures and tables

* resubmit electronically if possible using the link below to access your home page:

Link Redacted

*This url links to your confidential homepage and associated information about manuscripts you may have submitted or be reviewing for us. If you wish to forward this e-mail to co-authors, please delete this link to your homepage first.

Please ensure that all correspondence is marked with your Nature Microbiology reference number in the subject line.

Nature Microbiology is committed to improving transparency in authorship. As part of our efforts in this direction, we are now requesting that all authors identified as 'corresponding author' on published papers create and link their Open Researcher and Contributor Identifier (ORCID) with their account on the Manuscript Tracking System (MTS), prior to acceptance. This applies to primary research papers only. ORCID helps the scientific community achieve unambiguous attribution of all scholarly contributions. You can create and link your ORCID from the home page of the MTS by clicking on 'Modify my Springer Nature account'. For more information please visit www.springernature.com/orcid.

We hope to receive your revised paper within three weeks. If you cannot send it within this time, please let us know.

Yours sincerely,

Reviewers Comments:

Reviewer #1 (Remarks to the Author):

The authors have addressed my concerns and I thank them for their work.

Reviewer #2 (Remarks to the Author):

The authors have adequately and in great detail responded to the reviewers questions and performed substantial new experiments (e.g. microbiota metagenomics, IFN γ inhibition in vivo). The revised manuscript presents a comprehensive study on the effect of the weaning microbiota induced IFN γ on the epigenetic control of the inducible MHCII gene expression with long-term consequences on disease susceptibility. With the changes made, I think the manuscript should be accepted for publication.
Mathias Hornef

Reviewer #3 (Remarks to the Author):

Thank you to the authors for the substantial revisions and for incorporating detailed microbiome analyses. These additions go a long way toward addressing the concerns raised in the previous submission. However, the microbiome component of the study remains largely descriptive, and the new analyses, while informative, still rely primarily on correlations between shotgun metagenomic profiles and GO-term-based functional predictions. This does not provide a direct mechanistic link or in vivo evidence demonstrating microbial causality.

Given the expectations for Nature Microbiology, it would strengthen the manuscript considerably to include more robust microbiological work that directly connects specific microbial taxa or functions to host phenotypes or mechanisms. Although the revised manuscript is significantly improved overall and the host profiling is very detailed, the absence of mechanistic microbial validation may mean that another (still excellent) journal would be a more suitable fit.

Version 3:

Decision Letter:

Our ref: NMICROBIOL-24082649C

22nd January 2026

Dear Lanlan,

Thank you for submitting your revised manuscript "Weaning-driven gut microbiome reprograms intestinal stem cell epigenetics to establish epithelial-intrinsic immune memory" (NMICROBIOL-24082649C). We are happy to offer to publish it in principle in Nature Microbiology, pending minor revisions to comply with our editorial and formatting guidelines.

Thank you again for your interest in Nature Microbiology Please do not hesitate to contact me if you have any questions.

Best wishes,

Version 4:

Decision Letter:

12th February 2026

Dear Lanlan,

I am pleased to accept your Article "Weaning drives microbiome-mediated epigenetic regulation to shape immune memory in mice" for publication in Nature Microbiology. Thank you for choosing us to submit your work to, and many congratulations!

Over the next few weeks, your paper will be copyedited to ensure that it conforms to Nature Microbiology style. We look

particularly carefully at the titles of all papers to ensure that they are relatively brief and understandable.

Once your paper is typeset, you will receive an email with a link to choose the appropriate publishing options for your paper and our Author Services team will be in touch regarding any additional information that may be required. Once your paper has been scheduled for online publication, the Nature press office will be in touch to confirm the details. Some other information is at the end of this email.

Best wishes,

Authors may need to take specific actions to achieve compliance with funder and institutional open access mandates. If your research is supported by a funder that requires immediate open access (e.g. according to [Plan S principles](https://www.springernature.com/gp/open-science/plan-s-compliance) or the [NIH public access policy](https://www.springernature.com/gp/open-science/us-federal-agency-compliance)) then you should select the gold OA route, and we will direct you to the compliant route where possible. Because authors warrant under our subscription licensing terms that they haven't committed to licensing any version of their article under a licence inconsistent with the terms of our agreement – including the applicable embargo period – publication under the subscription model isn't suitable for authors whose funders require no embargo.

P.S. Click on the following link if you would like to recommend Nature Microbiology to your librarian
<http://www.nature.com/subscriptions/recommend.html#forms>

** Visit the Springer Nature Editorial and Publishing website at http://editorial-jobs.springernature.com?utm_source=ejP_NMicro_email&utm_medium=ejP_NMicro_email&utm_campaign=ejp_NMicro for more information about our career opportunities. If you have any questions please click [here](mailto:editorial.publishing.jobs@springernature.com).

POINT-BY-POINT RESPONSE TO THE REVIEWERS' COMMENTS

We thank all the reviewers for their thoughtful and constructive comments on our manuscript. We have carefully revised the work with additional clarifications, new experiments, and expanded data analyses. These revisions directly address the reviewers' concerns and substantially strengthen our central conclusion: the coordination of microbial and immune cues, driven by weaning, regulates epithelial MHC-II genes epigenetically for lifelong function.

Specifically, in this revision, we:

1. **Clarified the focus on long-lasting epigenetic effects** – We study how the gut microbiome stably influences the epigenetic state of the colonic epithelium by examining genome-wide DNA methylation and gene expression in adult intestinal stem cells (ISCs) and their differentiated cells. MHC-II genes stood out because their expression in epithelial cells is controlled by microbes, but the underlying mechanisms are still unclear.
2. **Expanded microbiome profiling** – We now include a detailed analysis of microbial communities, identifying specific bacterial species and molecular functions that drive epigenetic reprogramming in ISCs.
3. **Revealed a new aspect of epigenetic regulation** – Our findings show that microbiota can reprogram epigenetic states through enhancer demethylation, priming long-lived ISCs and their progeny to respond more effectively to later challenges. This “epigenetic memory” mirrors the transcriptional memory seen in immune cells.
4. **Demonstrated physiological relevance** – Using IFN- γ antibody to neutralize IFN- γ in vivo, we show that the weaning-associated IFN- γ burst is essential for initiating this epigenetic reprogramming, directly connecting this process to the critical timing of demethylation driven by weaning.
5. **Elucidated the mechanistic pathway** – We demonstrate that the IFN- γ -TET3 axis with microbiota-derived metabolites mediates DNA demethylation, thereby linking early-life microbial signals with stable ISC epigenetic remodeling.
6. **Demonstrated long-term functional consequences** – We provide evidence that MHC-II enhancer demethylation in ISCs creates a durable epigenetic memory, providing a critical protective role against colitis and tumorigenesis.

The new data not only strengthen our original conclusions but also improve the manuscript. We believe this work represents a conceptual advance in understanding how early-life events shape mucosal immunity through long-lasting epigenetic regulation.

We have provided a detailed, point-by-point response to each reviewer's comments below.

Reviewer #1 (Remarks to the Author):

The study is potentially interesting in the context of the early life impact of microbiota on the immune system into adulthood, as the mechanisms involved in that phenomenon remain relatively unclear. Nevertheless, this study does not address this point, as claimed to be addressed in the introduction, but rather shows that the microbiota regulates expression of immune genes, probably through IFN γ . This latter point also requires more data, as only specifically addressed in vitro. Furthermore, immunological memory at the epigenetic level is only shown in vitro for 48h.

We appreciate the reviewer's thoughtful comments and the opportunity to clarify our findings. Our results demonstrate that the weaning period is critical for establishing epigenetic regulation of ISCs that supports epithelial-intrinsic immune responses. We show that weaning-associated microbial and IFN- γ signals stably reprogram enhancer methylation of MHC-II genes in ISCs,

resulting in an epigenetic memory that persists into adulthood. As intestinal epithelial cells form the first line of innate immune defense, these findings establish a direct link between early-life microbial exposures and long-term epithelial immune function. Accordingly, we have revised the manuscript title to: “Weaning-driven gut microbiome reprograms intestinal stem cell epigenetics to establish epithelial-intrinsic immune memory”.

To further address the reviewer’s concerns regarding *in vivo* IFN- γ effects and transcriptional memory, we performed additional experiments. Neutralizing IFN- γ *in vivo* revealed that transient weaning-associated IFN- γ production is required for establishing long-lasting, microbiome-dependent epigenetic regulation of epithelial MHC-II signaling (Figure 4). IFN- γ primed organoids challenged with unrelated stimuli exhibited methylation-dependent enhancement of MHC-II transcription. Similarly, organoids from 15-week-old GF, SPF, and GF mice that received gut microbiota transfer (GMT) at weaning showed rapid MHC-II gene activation within 6 h of IFN- γ restimulation (Figure 7G-I). Together, these findings further support that microbiota- and IFN- γ -driven enhancer demethylation establishes durable epithelial trained immunity in long-lived ISCs and their progeny, priming them for rapid responses to diverse secondary challenges.

Specific comments:

1. Line 49: the authors should cite the original papers, and not reviews, when discussing the weaning reaction, in order to avoid confusion about the phenomenon that they really are addressing in the study. For example, in figure 1, the comparison of SPF and GF mice does not address the weaning reaction or the early life impact of the microbiota on the immune system.

We have cited the original paper (Al Nabhani, Z. et al. *Immunity*, 2019) regarding the weaning reaction. In response to the reviewers’ feedback, Figure 1 has been revised to specifically highlight microbiome-associated methylation changes that persist during epithelial differentiation, likely representing an epigenetic memory of early microbial exposure.

2. Figures 1, 2, S3: it has been reported (Koo and Clevers, *Gastroenterology* 2014) that expression of GFP in the *Lgr5*-GFP reporter mice is mosaic due to random inactivation of the knock-in allele. Thus, *LGR5*⁺ ISCs are also present in the GFP^{neg} cells, which is a problem when comparing ISC to differentiated epithelial cells in figure 2 and S3, particularly when defining common DMRs. CD24 staining would help to exclude GFP^{neg} ISCs.

We agree that *Lgr5*-GFP mice are mosaic and that GFP⁻ fractions may contain a small fraction (<5%) of stem cells. However, to determine whether microbiota-dependent DNA methylation changes in ISCs persist during differentiation, purified ISCs from *Lgr5*-GFP mice were essential for this analysis. The vast majority of GFP⁻ cells are differentiated EpCAM⁺ intestinal epithelial cells (IECs), and we have revised the manuscript to refer to this population accordingly.

Consistent with this, we demonstrate microbiota-dependent persistent methylation changes in ISCs and IECs, including immune and host-defense genes, that, once established after weaning, persist into adulthood (Figure 3C). Similar methylation profiles observed in wild-type (WT) mice and in EpCAM⁺CD24⁺ epithelial subpopulations further highlight the stable, genotype- and cell type-independent epigenetic regulation consistent with the long-term epigenetic memory (Figure S4B,C).

3. Figure 4 shows an apparently linear time-dependent effect of the microbiota on methylation. As presented, the data suggest instead that colonizing germfree mice during weaning, during early adulthood or as fully adult mice has decreasing effects. To make the point that it is the age at which the mice are colonized, rather than the duration of colonization that matters, the

following experiment should be done: the “weaning” group, colonized for 12 weeks before assessment of methylation and gene expression, should be compared to adult mice colonized for 12 weeks before assessment.

We apologize for not making this clear in the original manuscript. The experiment addressing this point was previously shown in the Supplemental Figures. In the revised manuscript, Figures 3D-F present the adult transplant data alongside the post-weaning and adolescent groups, with samples collected either at the same age (15 weeks) or after an extended period (20 weeks). After 20 weeks, we observed the same results for both DNA methylation and gene expression. Additionally, we performed metagenomic profiling using shotgun whole-genome sequencing of cecal samples, which confirmed similar levels of alpha and beta diversity at both phylum and genus levels (Figure S6A,B). Interestingly, although the relative abundance of the major bacterial phyla was similar across groups (Figure 3E), the relative abundance of *Bifidobacteria* at the genus level in the adult transplant groups (Figure S6C) was reduced. This suggests that specific early-colonizing Gram-positive bacteria that induce IFN- γ may direct the epigenetic reprogramming. We provide further support for this in our studies using IFN- γ inhibition and low-dose penicillin treatments.

4. Figure 5 shows that IFN γ alters the methylation and expression of Cd74 in organoid cultures. It is indeed expected that IFN γ is involved in the regulation of MHC II molecules and associated proteins, such as CIITA and CD74. But the key experiment has to be done in vivo in mice treated with antibodies neutralizing IFN γ at different age of the mice, and all MHC II-related molecules have to be monitored (MHC II, CD74, CIITA, ...). Furthermore, the authors mention “training immunological memory” in the title, but the only actual memory reported in this study is epigenetic stability for 48h in vitro.

As mentioned, we added in vivo experiments showing that neutralizing IFN- γ at weaning disrupts microbe-induced epigenetic reprogramming—both demethylation and transcriptional activation—of MHC-II genes (*Cd74*, *H2-Aa*, *H2-Eb1*, and *Ciita*), whereas neutralization in adult mice had no significant effect (Figure 4).

We replaced “immunological memory” with “epithelial-intrinsic immune memory” because organoids previously exposed to IFN- γ or gut microbiota exhibit features of trained immunity, acquiring faster and stronger MHC-II gene activation upon restimulation, independent of adaptive immunity. Following standard trained immunity protocols, organoids were rested for 7–20 days before secondary stimulation with IFN- γ or TNF- α , and transcriptional responses were measured over 48 h, as is typical for such assays (Figure 7D-I). These data indicate that MHC-II transcriptional memory is mediated by enhancer demethylation.

5. Figure 6 is largely irrelevant in this study as it is expected that a loss in microbiota (here Gram+ bacteria) leads to a reduction in immune response, here via IFN γ , and thus a loss in MHC II expression and associated molecules.

Although the link between Gram⁺ bacteria, IFN- γ , and MHC-II expression is known, the underlying regulatory mechanism has not been described. The low-dose penicillin model served to validate this connection at the epigenetic level and to provide mechanistic insights at both the microbiome species and functional levels (Figure 5 and Figure 6).

Reviewer #2 (Remarks to the Author):

Yang et al report in their manuscript entitled “Weaning driven gut microbiome shapes intestinal stem cell epigenetics to train immunological memory” on microbiota induced epigenetic

alterations in the colonic intestinal epithelium that are induced during an early life window and partly conferred by IFN-gamma. They among others alter epithelial CD74 and MHCII expression and susceptibility to DSS colitis. The topic is novel, relevant and highly interesting. The experiments are generally well-conducted and the data are clearly presented.

However, some conclusions drawn by the authors are not supported by the data and the focus on CD74 and MHCII expression among the large number of epigenetically altered genes is not convincingly explained. A global view on genes influenced by epigenetic changes would have been very interesting. And this in particular of genes involved in immune responses and inflammation, central e.g. to the DSS colitis model used in Fig. 6.

We thank the reviewer for the positive assessment and constructive comments. Because of their potential relevance to chronic diseases, our study was designed to comprehensively analyze microbiome-dependent DNA methylation changes and identify those stably maintained during differentiation and into adulthood. Within this framework, we focused on *Cd74* and MHC-II genes because they are functionally relevant, microbially regulated targets. Thus, we have added clarifications, conducted new experiments, and provided detailed lists of differentially methylated regions and genes in the revised manuscript.

Major points

1. In the abstract a causal relationship between vanishing maternal antibodies and the weaning reaction is implicated. This has not been shown. Instead, the alteration of the microbiota at weaning was suggested to cause the weaning reaction. In the introduction, this link is correctly described (I49).

We agree and have changed this in the Abstract.

2. Line 67 and 68 suggest that the loss of MHCII expression is associated with enhanced colitis susceptibility. However, this is a matter of debate (Ozkul et al., *Genome Med.*, 2020 versus Jin et al., *Sci Rep*, 2017).

We appreciate the reviewer's comment. Conflicting results regarding the mouse colitis model were noted in our original manuscript, and to ensure robustness, we repeated the experiment. Consistent with Ozkul et al, low-dose penicillin exposure exacerbates DSS-induced colitis, and in an additional AOM/DSS model, we observed enhanced colon cancer development (Figure 5, Figure S8 and S9).

3. L77: "...transitioned from SPF to germ-free conditions..." This transition has not been studied. Instead, GF bred mice were compared with SPF mice or conventionalized GF (exGF) mice.

We apologize for the confusion. GF *Lgr5*-GFP mice were generated from SPF *Lgr5*-GFP mice by C-section and cross-fostering with a GF dam. These mice were then colonized to study microbiota effects on stem cells; a schematic diagram of this approach has been added in Figure 1.

4. The highest MHCII expression is found in the mid small intestine, much higher than in the colon (Stephens et al., *Cell Rep.*, 2021). It is not clear why the morphological changes that are observed in the colon but less so in the small intestine should prompt to focus on MHCII expression in the colon rather than the small intestine.

To clarify, our study design first ensured that GF *Lgr5*-GFP mice faithfully recapitulated previously reported GF phenotypes by comparing them to published results in WT mice. As

expected, we observed prominent GF-associated morphological and histological changes in the colon, including reduced proliferation and a decrease in goblet cells. These findings guided our focus on the colon, where microbiota-dependent effects were most pronounced.

Because DNA methylation patterns vary with both cell type and differentiation, there are limitations in using bulk IECs to study microbial influences. To address this, we used purified Lgr5-GFP⁺ ISCs, which allowed us to avoid cell-type heterogeneity and directly assess microbiota-dependent, locus-specific methylation changes.

Importantly, in characterizing methylation states across intestinal segments, we found that microbiota-dependent changes at MHC-II loci were consistent in both the small and large intestines (Figure S4A). The absolute expression levels of MHC-II, however, varied by segment, which we attribute primarily to promoter-driven baseline activity. In contrast, our data suggest that enhancer methylation predominantly regulates inducible rather than basal expression. This conclusion is supported by our organoid experiments (Figure 7), which show that enhancer methylation modulates cytokine responsiveness, providing faster and stronger transcriptional induction.

In summary, our focus on the colon was motivated by (1) its more pronounced microbiota-dependent morphological and histological changes, and (2) our interest in determining how stable ISC-to-IEC methylation changes, particularly in immune-related and host-defense genes, contribute to colitis and colorectal cancer.

5. L168-169: Only CD74 is shown in Fig. S5 so the text should not refer to “MHCII-related genes” but only CD74. Alternatively, show the data to support the statement.

We would like to clarify that we measured DNA methylation and gene expression changes from the post-weaning period into adulthood and observed persistent hypomethylation in MHC-II genes (*Cd74*, *H2-Eb1*, *H2-Aa*, and *Ciita*) after weaning. Notably, demethylation of *Cd74* occurred most rapidly at 4 weeks of age. To further support this finding, we included methylation and expression analyses before weaning (at 2 weeks of age). The new data are consistent: microbiome-induced hypomethylation arises after weaning, with these changes preceding or coinciding with increased transcription (Figure 3C).

6. L105 and Fig. S3b suggest that only a fraction of DMRs is shared between ISCs and differentiated cells, however, Fig. 1CD, Fig. 2a and Fig. S3E suggest little difference between those two cell fractions. Please explain.

Only a fraction of DMRs is shared between ISCs and IECs, likely reflecting cell-type- and differentiation-dependent methylation differences (Figure 1C). Our analyses focused on the shared DMRs, which represent microbial-dependent changes established in long-lived ISCs and maintained through differentiation (Figure 1D). GO analysis of the shared DMR-genes revealed enrichment for immune-related pathways, including response to type II interferon and antigen processing/presentation (Figure 1E). These results guided our detailed characterization as shown in the genome browser views (Figure 1F). Thus, we prioritized the microbial effects most likely to have a lasting functional impact.

7. It is unclear why this paper focusses on MHCII and CD74 expression given that a vast number of genes (Fig. S3A and B) is affected by the epigenetic changes and differentially expressed (1107-113).

As noted above, we focus on MHC-II genes because they are functionally relevant targets of the microbiota, yet their underlying regulatory mechanisms remain poorly understood. Nonetheless, our complete genome-wide datasets, deposited in GEO, along with supplemental tables listing all DMRs, provide a valuable resource for the scientific community studying host-microbiome interactions.

8. Doesn't the fact that the microbiota is much denser in the colon as compared to small intestine (that also shows differential MHCII and CD74 expression) suggest that the effect upstream of the epigenetic alterations is indirect? Is IFN γ responsible for this indirect effect?

Our results show that the epigenetic effects are directly mediated by the gut microbiota, as demonstrated by microbiome transplant and antibiotic experiments. Microbial colonization during weaning triggers a cytokine response, inducing IFN- γ , which drives TET-mediated locus-specific demethylation. Consequently, long-lasting epigenetic regulation is driven by microbiome-immune-ISC cross-talk during postnatal development, rather than being an indirect effect of the higher microbial density in the colon.

9. It is unclear how old the mice are in Fig. 1 (not indicated in figure or legend nor results text). These mice apparently show no difference in overall methylation but the adult mice in Fig. S2 do.

We apologize for this oversight and have now added the age of the mice for all relevant figures in both the figure legends and the Results section.

10. L181ff. I disagree with the interpretation. GMT during early life also means a much longer presence of the microbiota when all groups are analysed at the same age. The length of exposure (and not the age window) may therefore be important.

As noted in our previous response to Reviewer 1 (point 3), the adult transplant experiment is now presented in Figure 3D-F alongside the post-weaning and adolescent groups, with samples collected at 15 weeks and 20 weeks. The extended time did not alter DNA methylation or gene expression, and microbiome profiling showed comparable composition across groups based on both alpha and beta diversity.

11. Fig. 5D: Again, I disagree with the interpretation. Previous IFN γ treatment also means prolonged IFN γ exposure which by itself could explain the stronger response. I believe one should do an experiment in which IFN γ exposure comes first, but another second stimulus induces another target gene that is under IFN γ -mediated epigenetic control.

This point is well taken. We added experiments with colon organoids from 6-week-old GF mice, pretreated with IFN- γ and subsequently challenged with TNF- α , which induced rapid and enhanced *Cd74* expression. Organoids from 15-week-old GF, SPF, and GF \rightarrow SPF mice (converted at weaning) confirmed microbial effects on methylation-dependent, memory-like responses across all MHC-II genes. These results are now shown in Figure 7.

12. Fig. 6B. The IELs are insufficiently characterized to call them IELs (see Fig. S7). Staining for TCR α /b or TCR γ /d plus CD8aa would be needed to make that point. Please also consider TCR γ /d nIELs as source of the IFN γ .

We stained IFN- γ expressing CD4 $^+$ or CD8 $^+$ T cells for both TCR α / β $^+$ (CD45 $^+$ TCR β $^+$ CD3 ϵ $^+$) and non- α / β (CD45 $^+$ TCR β $^+$ CD3 ϵ $^+$) populations following the STAR protocol (Kim E, et al, STAR

Protoc, 2022). While we did not observe differences in the CD8⁺ population after low-dose penicillin treatment, we identified a significant difference in IFN- γ -producing CD4⁺ IELs. These analyses have been incorporated into our updated gating strategy (Figure S14).

13. L237/238: I agree with the comment that CD74 KO mice are more susceptible to DSS colitis, but the interpretation of the results reported in ref 43 on DSS exposed IEC-specific MHCII KO mice is difficult. No more weight loss and no shorter colon length reported, both typical markers of DSS colitis.

We thank the reviewer for the comment. The IEC-specific MHC-II knockout was generated using an *H2-Ab1* floxed allele with constitutive Villin-Cre expression from embryonic day E12.5, and indeed, there are conflicting results regarding DSS colitis in these mice (Jamwal DP, et al, Gastroenterology, 2020; Stephens WZ, et al, Cell Rep, 2021). Our study demonstrates that enhancer-mediated epigenetic regulation occurs after weaning and affects primarily inducible, rather than basal MHC-II expression, particularly in the colon. We cited these papers to provide a functional context. We believe that our mechanistic insights—such as the timing of microbiota-dependent epigenetic reprogramming and the role of memory responses—may help explain these previously conflicting observations.

14. Fig. 1B: Please include FACS analysis of Lgr5-GFP allele negative mice as control.

In all FACS sorting and downstream analyses, we used wild-type (without Lgr5-GFP knock-in) mice as negative controls for gating. This is shown in our gating strategy in Figure S14A, and we have stated this explicitly in the figure legend to ensure it is clear to readers.

15. Fig. 5C: The IFN γ effect is relatively weak (and the addition of the DAC reagent does not add any information). Have other cytokines been tested? What might also have a similar effect?

The relatively modest IFN- γ effect observed at 48 h in organoid cultures reflects its toxicity to ISCs (Figure 7C and Figure S10). To achieve complete demethylation at the stem cell level, we therefore applied a very low dose of IFN- γ over three passages (Figure 7E). We also tested TNF- α , sodium butyrate, and LPS; however, none altered methylation of MHC-II genes. In contrast, the hypomethylating agent DAC alone effectively demethylated the *Cd74* enhancer but did not induce gene expression. Interestingly, DAC and IFN- γ together synergistically upregulated transcription (Figure 7C). These observations promoted us to hypothesize that enhancer demethylation conveys an epigenetic form of transcriptional memory: it leaves basal expression unchanged but primes epithelial cells for faster and stronger responses to future stimuli. Indeed, we further demonstrated that microbiota-induced enhancer demethylation enables transcriptional memory in long-lived ISCs (Figure 7D-I).

16. Fig5D: Does not show that IFN γ acts via an epigenetic change. The difference could also be explained by prolonged IFN γ exposure.

17. Fig. 5D vs E: Isn't the 48h group in Fig. 5E very similar to the pretreated group in Fig.5D: But in 5E, the expression does not differ from the naïve (control) group. Please explain.

For both points, we agree with the reviewer that IFN- γ treatment for 48 hours results in heterogeneous demethylation that does not fully reflect ISC-level changes. Accordingly, we removed this group and now present data after three passages, which allows sufficient time for complete demethylation in ISCs. Our revised data clearly show that IFN- γ -primed organoids display a memory response to subsequent related or unrelated challenges (IFN- γ or TNF- α) (Figure 7D-F).

In addition to the experiments described above, we performed ChIP-qPCR analysis showing that IFN- γ recruits STAT1 and demethylase TET3 to the enhancer regions of *Cd74*, directly demonstrating the mechanistic link between IFN- γ signaling and epigenetic regulation (Figure 7B).

18. Fig. 6B: Please show complete 16S rRNA gene analysis. Lactobacilli, Clostridium (?), SFB and Bacteroides/Prevotella (also Fig. S7)) may not be representative for Gram+/- bacteria. Also, all taxa shown have very very low abundance in a substantial number of mice. A quantitative bacterial PCR would help to assess the effect of the LDP regimen.

We agree and now include longitudinal metagenomic analysis of fecal samples with or without low-dose penicillin (LDP) treatment to characterize the microbiomes and identify specific bacteria relevant to the observed epigenetic effects (Figure 6).

19. Fig. 6D: p values are only shown for the comparison between the SPF and LDP group. Is none of the LDP GF comparisons significantly different?

To avoid confusion, we now show only the comparison between mice with or without (w/o) LDP treatment (Figure 5B-F), removing GF data and SPF labels since all LDP-treated mice were housed under SPF conditions.

20. Fig. 6E: the MHCII staining is hard to interpret. The gate is placed in a very artificial way.

We reanalyzed the data using mean fluorescence intensity (MFI) to quantify MHC-II levels, which confirmed the original finding of reduced MHC-II expression in IECs under LDP treatment. Representative flow plots and the corresponding quantitative data are shown in Figure 5F.

21. Fig. 6F-H: I believe the authors imply that the effect of LDP on DSS susceptibility is MHCII dependent (based on the ref43). However, this is not formally shown. The enhanced susceptibility LPS could be due to other epigenetically modified genes or independent effects (e.g. of immune cell homing or IEC proliferation etc.). Please repeat the experiment with IEC MHCII KO mice to show that the difference is gone.

We thank the reviewer for the suggestion. However, using IEC-specific MHC-II knockout mice for LDP/DSS treatment would not directly address the role of DNA methylation. If the gene regulatory sequence is deleted, transcriptional changes driven by epigenetic modifications could not occur; therefore, the absence of differences could not be attributed to methylation.

22. The title is misleading. I don't think DSS colitis can formally be interpreted as adaptive immunity model.

We have revised the title to: "Weaning-driven gut microbiome reprograms intestinal stem cell epigenetics to establish epithelial-intrinsic immune memory". The title highlights our key finding that weaning-driven gut microbiota induce enhancer demethylation, priming long-lived ISCs to orchestrate immune-like transcriptional responses that link early-life microbial exposures to long-term mucosal immune function.

Minor points

1. L158: 15 week old mice are not young adult mice but adult mice.

We have made the changes as suggested.

2. The work from Dana Philpott (Tsang et al., Gut microbes, 2022) that describes expression differences between GF and SPF mice should be cited.

We have cited this work in the Introduction.

3. Fig. 6B. Please show quantitative data.

We now show quantitative data in Figure S7.

Reviewer #3 (Remarks to the Author):

The study entitled “Weaning-Driven Gut Microbiome Shapes Intestinal Stem Cell Epigenetics to Train Immunological Memory” by Yang et al, presents interesting findings on the potential role of the presence (or absence) of the gut microbiome in shaping intestinal stem cell (ISC) epigenetics during weaning. However, there are several areas where the study could be strengthened, both in terms of experimental design and data interpretation.

We thank the reviewer for recognizing the novelty of our findings on the role of the gut microbiome in shaping ISC epigenetics during weaning, and we appreciate the suggestions regarding experimental design and interpretation. In this revision, we have clarified our focus on MHC-II genes, expanded microbiome profiling, elucidated the IFN- γ -TET3-metabolite pathway, demonstrated the epigenetic regulatory role of the weaning-associated IFN- γ burst, and established the long-term functional consequences of this epigenetic memory. We believe these revisions address the reviewer’s concerns, and we respond to the specific comments in detail below.

Comments:

- The use of germ-free (GF) mice is appropriate for this study, but it is unclear if microbial products were also excluded (or tested to confirm they were not present), which could lead to activation of responses, and could occur even in the absence of live bacteria (this links to a comment on the organoid studies). The descriptive nature of the findings is significant, but the lack of further mechanistic exploration into what specific components of the microbiota are critical limits the depth of this study. Furthermore, there is no clear identification of which microbiota components (microbes/MAMPs/metabolites etc.) are driving the observed effects. It would be key to address whether microbial products might have been tested alongside live bacterial exposures to distinguish their individual impacts.

We agree that microbial products or components, even in the absence of live bacteria, can influence host responses. During weaning, microbiota colonization and diversification trigger an IFN- γ peak which, together with microbiota-derived metabolites, drives TET3-mediated enhancer demethylation in ISCs. This establishes a durable, immune-cell-like memory, enabling robust MHC-II induction upon subsequent challenges. Thus, our findings reveal how microbial signals give rise to stable epigenetic and transcriptional memory in innate epithelial cells.

- The study heavily relies on the presence or absence of microbiota, but it does not provide any detailed microbiota profiling data (only qPCR targeted a very limited number of ‘taxa’ and on a small number of samples). This reductionist approach limits the scope of the study and findings. Microbiota composition could significantly influence the epigenetic regulation of ISCs. Profiling at different stages of the experiment would provide insights into how specific microbial

populations contribute to the observed effects. Indeed, the article focuses largely on epigenetic changes, but it lacks a strong connection to microbial mechanisms.

While the epigenetic changes in response to weaning and microbiota are interesting, the lack of sequencing data limits the ability to link these changes to specific microbial species or community structures. A deeper microbiome analysis using shotgun metagenomics, and potentially metabolomics, would significantly add robustness to the conclusions.

We agree with the comments; thus, we have added microbiome analysis using shotgun metagenomics, which provides comprehensive profiling of microbial communities across different experimental stages. These data allowed us to identify specific bacterial taxa and their functional potential, which correlate with the observed epigenetic changes in ISCs. Specific results related to microbial colonization and antibiotic effects over time are explained after the next two comments.

- The authors claim that microbial exposure during early life impacts ISC epigenetics, but they do not explore how stable microbial colonization is achieved. What microbial species are colonizing, and how stable is the colonization over time (linked to point above about lack of detailed microbiome profiling)? Data on microbial stability or shifts would strengthen the paper's claims about long-term impacts.
- Moreover, while low-dose antibiotics were tested, the lack of detailed analysis on how these interventions affect later microbiota dynamics and subsequent epigenetic reprogramming limits the interpretation of the data. The authors focus on the 4-6-week window for epigenetic changes but fail to explore bacterial knockdown timelines or bacterial regrowth postintervention.

To determine the impact of the gut microbiome on epigenetic regulation, we analyzed fecal microbial composition longitudinally at 2, 4, 6, 8, 10, and 12 weeks of age in SPF mice with or without early-life low-dose penicillin (LDP) exposure. Mice exposed to LDP exhibited epigenetic dysregulation of MHC-II genes similar to that observed in GF mice, allowing us to link specific bacterial taxa and functional features to epigenetic alterations. Notably, several bacterial genera that normally expand during the post-weaning period (after 4 weeks) were persistently reduced in LDP-exposed mice into adulthood (10–12 weeks). These changes included a loss of SCFA-producing species and a decline in microbial functions that support TET-mediated demethylation, including α -KG production, iron availability, and DNMT inhibition. Our results are summarized in Figure 6.

- The authors suggest that juvenile gut microbiota transfer (GMT) is more effective in restoring epigenetic regulation, but they do not provide sufficient data on how the timing of microbial introduction affects colonization or stability. Further data on the longitudinal stability of microbiota following transfer would help elucidate whether the effects are transient or lasting.

We have addressed this concern in our response to Reviewer 1 (point 3) through microbiome profiling across groups and extended analysis of experiments in adult transplants.

- In the GF organoid experiments, the authors mention that LPS did not induce responses. However, they do not test other bacterial ligands and metabolites that might have elicited a different response, such as peptidoglycan, other types of LPS like Bacteroides-derived LPS, and SCFAs, which may have different effects.

In addition to IFN- γ , we evaluated the effects of LPS, TNF- α , and sodium butyrate on the methylation of enhancers of MHC-II genes in GF organoids. Individually, none of these factors had an effect, indicating that IFN- γ is the primary regulator in this context.

- The use of the term “dysbiosis” is over-simplified in this context. The authors should define the term more rigorously, particularly when referencing such complex microbiota-host interactions.

We agree and have replaced the term ‘dysbiosis’ in the context of LDP modeling with a precise description of changes in specific Gram-positive bacterial populations.

- Several studies have demonstrated impacts of the microbiota, including during pre- and postweaning within the murine gastrointestinal track with response to epithelial responses and these studies are currently not referenced. The authors should cite these works to provide context and acknowledge existing literature.

We appreciate the suggestion and have cited relevant prior studies in the Introduction and Discussion.

- As highlighted above - introducing specific bacterial strains or microbial consortia into GF mice could help identify key microbial players and confirm the protective or detrimental effects observed. Additionally, using an antibiotic cocktail to induce near GF conditions (would also represent a more ‘normally-programmed’ host than GF mice) and then reintroducing specific bacteria could provide more detailed insights into which microbes are crucial for ISC epigenetic regulation. This could be coupled with in-depth sequencing (and metabolomics) on microbial populations during the key stages of intervention, as this would reveal microbial dynamics and potential drivers of the observed epigenetic changes.

This study provides valuable data on the microbiome's role in shaping ISC epigenetics, particularly during the weaning period. However, the lack of microbiota profiling, mechanistic detail, and consideration of microbial products weakens the conclusions. Incorporating microbial profiling and addressing the concerns mentioned above would significantly strengthen the manuscript.

In summary, we have incorporated detailed microbiome profiling at both the taxonomic and functional levels into our experimental design and data analyses (Figure 3E, Figure S6, Figure 5B,C, and Figure 6). These additional analyses not only corroborate our original findings but also provide new insights into the role of specific bacterial taxa and microbial functions in influencing ISC epigenetics, particularly during the critical weaning period. We hope these enhanced microbiome analyses, conducted in response to the reviewer's suggestion, fully address the concerns.

POINT-BY-POINT RESPONSE TO THE REVIEWERS' COMMENTS

We thank all the reviewers for their thoughtful and constructive comments on our manuscript. We have carefully revised the work with additional clarifications, new experiments, and expanded data analyses. These revisions directly address the reviewers' concerns and substantially strengthen our central conclusion: the coordination of microbial and immune cues, driven by weaning, regulates epithelial MHC-II genes epigenetically for lifelong function.

Specifically, in this revision, we:

1. **Clarified the focus on long-lasting epigenetic effects** – We study how the gut microbiome stably influences the epigenetic state of the colonic epithelium by examining genome-wide DNA methylation and gene expression in adult intestinal stem cells (ISCs) and their differentiated cells. MHC-II genes stood out because their expression in epithelial cells is controlled by microbes, but the underlying mechanisms are still unclear.
2. **Expanded microbiome profiling** – We now include a detailed analysis of microbial communities, identifying specific bacterial species and molecular functions that drive epigenetic reprogramming in ISCs.
3. **Revealed a new aspect of epigenetic regulation** – Our findings show that microbiota can reprogram epigenetic states through enhancer demethylation, priming long-lived ISCs and their progeny to respond more effectively to later challenges. This “epigenetic memory” mirrors the transcriptional memory seen in immune cells.
4. **Demonstrated physiological relevance** – Using antibody to neutralize IFN- γ in vivo, we show that the weaning-associated IFN- γ burst is essential for initiating this epigenetic reprogramming, directly connecting this process to the critical timing of demethylation driven by weaning.
5. **Elucidated the mechanistic pathway** – We demonstrate that the IFN- γ -TET3 axis with microbiota-derived metabolites mediates DNA demethylation, thereby linking early-life microbial signals with stable ISC epigenetic remodeling.
6. **Demonstrated long-term functional consequences** – We provide evidence that MHC-II enhancer demethylation in ISCs creates a durable epigenetic memory, providing a critical protective role against colitis and tumorigenesis.

The new data not only strengthen our original conclusions but also improve the manuscript. We believe this work represents a conceptual advance in understanding how early-life events shape mucosal immunity through long-lasting epigenetic regulation.

We have provided a detailed, point-by-point response to each reviewer's comments below.

Reviewer #1 (Remarks to the Author):

The study is potentially interesting in the context of the early life impact of microbiota on the immune system into adulthood, as the mechanisms involved in that phenomenon remain relatively unclear. Nevertheless, this study does not address this point, as claimed to be addressed in the introduction, but rather shows that the microbiota regulates expression of immune genes, probably through IFN γ . This latter point also requires more data, as only specifically addressed in vitro. Furthermore, immunological memory at the epigenetic level is only shown in vitro for 48h.

We appreciate the reviewer's thoughtful comments and the opportunity to clarify our findings. Our results demonstrate that the weaning period is critical for establishing epigenetic regulation of ISCs that supports epithelial-intrinsic immune responses. We show that weaning-associated microbial and IFN- γ signals stably reprogram enhancer methylation of MHC-II genes in ISCs,

resulting in an epigenetic memory that persists into adulthood. As intestinal epithelial cells form the first line of innate immune defense, these findings establish a direct link between early-life microbial exposures and long-term epithelial immune function. Accordingly, we have revised the manuscript title to: “Weaning-driven gut microbiome reprograms intestinal stem cell epigenetics to establish epithelial-intrinsic immune memory”.

To further address the reviewer’s concerns regarding *in vivo* IFN- γ effects and transcriptional memory, we performed additional experiments. Neutralizing IFN- γ *in vivo* revealed that transient weaning-associated IFN- γ production is required for establishing long-lasting, microbiome-dependent epigenetic regulation of epithelial MHC-II signaling (Figure 4). IFN- γ primed organoids challenged with unrelated stimuli exhibited methylation-dependent enhancement of MHC-II transcription. Similarly, organoids from 15-week-old GF, SPF, and GF mice that received gut microbiota transfer (GMT) at weaning showed rapid MHC-II gene activation within 6 h of IFN- γ restimulation (Figure 7G-I). Together, these findings further support that microbiota- and IFN- γ -driven enhancer demethylation establishes durable epithelial trained immunity in long-lived ISCs and their progeny, priming them for rapid responses to diverse secondary challenges.

Specific comments:

1. Line 49: the authors should cite the original papers, and not reviews, when discussing the weaning reaction, in order to avoid confusion about the phenomenon that they really are addressing in the study. For example, in figure 1, the comparison of SPF and GF mice does not address the weaning reaction or the early life impact of the microbiota on the immune system.

We have cited the original paper (Al Nabhani, Z. et al. *Immunity*, 2019) regarding the weaning reaction. In response to the reviewers’ feedback, Figure 1 has been revised to specifically highlight microbiome-associated methylation changes that persist during epithelial differentiation, likely representing an epigenetic memory of early microbial exposure.

2. Figures 1, 2, S3: it has been reported (Koo and Clevers, *Gastroenterology* 2014) that expression of GFP in the *Lgr5*-GFP reporter mice is mosaic due to random inactivation of the knock-in allele. Thus, *LGR5*⁺ ISCs are also present in the GFP^{neg} cells, which is a problem when comparing ISC to differentiated epithelial cells in figure 2 and S3, particularly when defining common DMRs. CD24 staining would help to exclude GFP^{neg} ISCs.

We agree that *Lgr5*-GFP mice are mosaic and that GFP⁻ fractions may contain a small fraction (<5%) of stem cells. However, to determine whether microbiota-dependent DNA methylation changes in ISCs persist during differentiation, purified ISCs from *Lgr5*-GFP mice were essential for this analysis. The vast majority of GFP⁻ cells are differentiated EpCAM⁺ intestinal epithelial cells (IECs), and we have revised the manuscript to refer to this population accordingly.

Consistent with this, we demonstrate microbiota-dependent persistent methylation changes in ISCs and IECs, including immune and host-defense genes, that, once established after weaning, persist into adulthood (Figure 3C). Similar methylation profiles observed in wild-type (WT) mice and in EpCAM⁺CD24⁺ epithelial subpopulations further highlight the stable, genotype- and cell type-independent epigenetic regulation consistent with the long-term epigenetic memory (Figure S4B,C).

3. Figure 4 shows an apparently linear time-dependent effect of the microbiota on methylation. As presented, the data suggest instead that colonizing germfree mice during weaning, during early adulthood or as fully adult mice has decreasing effects. To make the point that it is the age at which the mice are colonized, rather than the duration of colonization that matters, the

following experiment should be done: the “weaning” group, colonized for 12 weeks before assessment of methylation and gene expression, should be compared to adult mice colonized for 12 weeks before assessment.

We apologize for not making this clear in the original manuscript. The experiment addressing this point was previously shown in the Supplemental Figures. In the revised manuscript, Figures 3D-F present the adult transplant data alongside the post-weaning and adolescent groups, with samples collected either at the same age (15 weeks) or after an extended period (20 weeks). After 20 weeks, we observed the same results for both DNA methylation and gene expression. Additionally, we performed metagenomic profiling using shotgun whole-genome sequencing of cecal samples, which confirmed similar levels of alpha and beta diversity at both phylum and genus levels (Figure S6A,B). Interestingly, although the relative abundance of the major bacterial phyla was similar across groups (Figure 3E), the relative abundance of *Bifidobacteria* at the genus level in the adult transplant groups (Figure S6C) was reduced. This suggests that specific early-colonizing Gram-positive bacteria that induce IFN- γ may direct the epigenetic reprogramming. We provide further support for this in our studies using IFN- γ inhibition and low-dose penicillin treatments.

4. Figure 5 shows that IFN γ alters the methylation and expression of Cd74 in organoid cultures. It is indeed expected that IFN γ is involved in the regulation of MHC II molecules and associated proteins, such as CIITA and CD74. But the key experiment has to be done in vivo in mice treated with antibodies neutralizing IFN γ at different age of the mice, and all MHC II-related molecules have to be monitored (MHC II, CD74, CIITA, ...). Furthermore, the authors mention “training immunological memory” in the title, but the only actual memory reported in this study is epigenetic stability for 48h in vitro.

As mentioned, we added in vivo experiments showing that neutralizing IFN- γ at weaning disrupts microbe-induced epigenetic reprogramming—both demethylation and transcriptional activation—of MHC-II genes (*Cd74*, *H2-Aa*, *H2-Eb1*, and *Ciita*), whereas neutralization in adult mice had no significant effect (Figure 4).

We replaced “immunological memory” with “epithelial-intrinsic immune memory” because organoids previously exposed to IFN- γ or gut microbiota exhibit features of trained immunity, acquiring faster and stronger MHC-II gene activation upon restimulation, independent of adaptive immunity. Following standard trained immunity protocols, organoids were rested for 7–20 days before secondary stimulation with IFN- γ or TNF- α , and transcriptional responses were measured over 48 h, as is typical for such assays (Figure 7D-I). These data indicate that MHC-II transcriptional memory is mediated by enhancer demethylation.

5. Figure 6 is largely irrelevant in this study as it is expected that a loss in microbiota (here Gram+ bacteria) leads to a reduction in immune response, here via IFN γ , and thus a loss in MHC II expression and associated molecules.

Although the link between Gram+ bacteria, IFN- γ , and MHC-II expression is known, the underlying regulatory mechanism has not been described. The low-dose penicillin model served to validate this connection at the epigenetic level and to provide mechanistic insights at both the microbiome species and functional levels (Figure 5 and Figure 6).

Reviewer #2 (Remarks to the Author):

Yang et al report in their manuscript entitled “Weaning driven gut microbiome shapes intestinal stem cell epigenetics to train immunological memory” on microbiota induced epigenetic

alterations in the colonic intestinal epithelium that are induced during an early life window and partly conferred by IFN-gamma. They among others alter epithelial CD74 and MHCII expression and susceptibility to DSS colitis. The topic is novel, relevant and highly interesting. The experiments are generally well-conducted and the data are clearly presented. However, some conclusions drawn by the authors are not supported by the data and the focus on CD74 and MHCII expression among the large number of epigenetically altered genes is not convincingly explained. A global view on genes influenced by epigenetic changes would have been very interesting. And this in particular of genes involved in immune responses and inflammation, central e.g. to the DSS colitis model used in Fig. 6.

We thank the reviewer for the positive assessment and constructive comments. Because of their potential relevance to chronic diseases, our study was designed to comprehensively analyze microbiome-dependent DNA methylation changes and identify those stably maintained during differentiation and into adulthood. Within this framework, we focused on *Cd74* and MHC-II genes because they are functionally relevant, microbially regulated targets. Thus, we have added clarifications, conducted new experiments, and provided detailed lists of differentially methylated regions and genes in the revised manuscript.

Major points

1. In the abstract a causal relationship between vanishing maternal antibodies and the weaning reaction is implicated. This has not been shown. Instead, the alteration of the microbiota at weaning was suggested to cause the weaning reaction. In the introduction, this link is correctly described (I49).

We agree and have changed this in the Abstract.

2. Line 67 and 68 suggest that the loss of MHCII expression is associated with enhanced colitis susceptibility. However, this is a matter of debate (Ozkul et al., *Genome Med.*, 2020 versus Jin et al., *Sci Rep*, 2017).

We appreciate the reviewer's comment. Conflicting results regarding the mouse colitis model were noted in our original manuscript, and to ensure robustness, we repeated the experiment. Consistent with Ozkul et al, low-dose penicillin exposure exacerbates DSS-induced colitis, and in an additional AOM/DSS model, we observed enhanced colon cancer development (Figure 5, Figure S8 and S9).

3. L77: "...transitioned from SPF to germ-free conditions..." This transition has not been studied. Instead, GF bred mice were compared with SPF mice or conventionalized GF (exGF) mice.

We apologize for the confusion. GF *Lgr5*-GFP mice were generated from SPF *Lgr5*-GFP mice by C-section and cross-fostering with a GF dam. These mice were then colonized to study microbiota effects on stem cells; a schematic diagram of this approach has been added in Figure 1.

4. The highest MHCII expression is found in the mid small intestine, much higher than in the colon (Stephens et al., *Cell Rep.*, 2021). It is not clear why the morphological changes that are observed in the colon but less so in the small intestine should prompt to focus on MHCII expression in the colon rather than the small intestine.

To clarify, our study design first ensured that GF *Lgr5*-GFP mice faithfully recapitulated previously reported GF phenotypes by comparing them to published results in WT mice. As

expected, we observed prominent GF-associated morphological and histological changes in the colon, including reduced proliferation and a decrease in goblet cells. These findings guided our focus on the colon, where microbiota-dependent effects were most pronounced.

Because DNA methylation patterns vary with both cell type and differentiation, there are limitations in using bulk IECs to study microbial influences. To address this, we used purified Lgr5-GFP⁺ ISCs, which allowed us to avoid cell-type heterogeneity and directly assess microbiota-dependent, locus-specific methylation changes.

Importantly, in characterizing methylation states across intestinal segments, we found that microbiota-dependent changes at MHC-II loci were consistent in both the small and large intestines (Figure S4A). The absolute expression levels of MHC-II, however, varied by segment, which we attribute primarily to promoter-driven baseline activity. In contrast, our data suggest that enhancer methylation predominantly regulates inducible rather than basal expression. This conclusion is supported by our organoid experiments (Figure 7), which show that enhancer methylation modulates cytokine responsiveness, providing faster and stronger transcriptional induction.

In summary, our focus on the colon was motivated by (1) its more pronounced microbiota-dependent morphological and histological changes, and (2) our interest in determining how stable ISC-to-IEC methylation changes, particularly in immune-related and host-defense genes, contribute to colitis and colorectal cancer.

5. L168-169: Only CD74 is shown in Fig. S5 so the text should not refer to “MHCII-related genes” but only CD74. Alternatively, show the data to support the statement.

We would like to clarify that we measured DNA methylation and gene expression changes from the post-weaning period into adulthood and observed persistent hypomethylation in MHC-II genes (*Cd74*, *H2-Eb1*, *H2-Aa*, and *Ciita*) after weaning. Notably, demethylation of *Cd74* occurred most rapidly at 4 weeks of age. To further support this finding, we included methylation and expression analyses before weaning (at 2 weeks of age). The new data are consistent: microbiome-induced hypomethylation arises after weaning, with these changes preceding or coinciding with increased transcription (Figure 3C).

6. L105 and Fig. S3b suggest that only a fraction of DMRs is shared between ISCs and differentiated cells, however, Fig. 1CD, Fig. 2a and Fig. S3E suggest little difference between those two cell fractions. Please explain.

Only a fraction of DMRs is shared between ISCs and IECs, likely reflecting cell-type- and differentiation-dependent methylation differences (Figure 1C). Our analyses focused on the shared DMRs, which represent microbial-dependent changes established in long-lived ISCs and maintained through differentiation (Figure 1D). GO analysis of the shared DMR-genes revealed enrichment for immune-related pathways, including response to type II interferon and antigen processing/presentation (Figure 1E). These results guided our detailed characterization as shown in the genome browser views (Figure 1F). Thus, we prioritized the microbial effects most likely to have a lasting functional impact.

7. It is unclear why this paper focusses on MHCII and CD74 expression given that a vast number of genes (Fig. S3A and B) is affected by the epigenetic changes and differentially expressed (1107-113).

As noted above, we focus on MHC-II genes because they are functionally relevant targets of the microbiota, yet their underlying regulatory mechanisms remain poorly understood. Nonetheless, our complete genome-wide datasets, deposited in GEO, along with supplemental tables listing all DMRs, provide a valuable resource for the scientific community studying host-microbiome interactions.

8. Doesn't the fact that the microbiota is much denser in the colon as compared to small intestine (that also shows differential MHCII and CD74 expression) suggest that the effect upstream of the epigenetic alterations is indirect? Is IFN γ responsible for this indirect effect?

Our results show that the epigenetic effects are directly mediated by the gut microbiota, as demonstrated by microbiome transplant and antibiotic experiments. Microbial colonization during weaning triggers a cytokine response, inducing IFN- γ , which drives TET-mediated locus-specific demethylation. Consequently, long-lasting epigenetic regulation is driven by microbiome-immune-ISC cross-talk during postnatal development, rather than being an indirect effect of the higher microbial density in the colon.

9. It is unclear how old the mice are in Fig. 1 (not indicated in figure or legend nor results text). These mice apparently show no difference in overall methylation but the adult mice in Fig. S2 do.

We apologize for this oversight and have now added the age of the mice for all relevant figures in both the figure legends and the Results section.

10. L181ff. I disagree with the interpretation. GMT during early life also means a much longer presence of the microbiota when all groups are analysed at the same age. The length of exposure (and not the age window) may therefore be important.

As noted in our previous response to Reviewer 1 (point 3), the adult transplant experiment is now presented in Figure 3D-F alongside the post-weaning and adolescent groups, with samples collected at 15 weeks and 20 weeks. The extended time did not alter DNA methylation or gene expression, and microbiome profiling showed comparable composition across groups based on both alpha and beta diversity.

11. Fig. 5D: Again, I disagree with the interpretation. Previous IFN γ treatment also means prolonged IFN γ exposure which by itself could explain the stronger response. I believe one should do an experiment in which IFN γ exposure comes first, but another second stimulus induces another target gene that is under IFN γ -mediated epigenetic control.

This point is well taken. We added experiments with colon organoids from 6-week-old GF mice, pretreated with IFN- γ and subsequently challenged with TNF- α , which induced rapid and enhanced *Cd74* expression. Organoids from 15-week-old GF, SPF, and GF \rightarrow SPF mice (converted at weaning) confirmed microbial effects on methylation-dependent, memory-like responses across all MHC-II genes. These results are now shown in Figure 7.

12. Fig. 6B. The IELs are insufficiently characterized to call them IELs (see Fig. S7). Staining for TCR α /b or TCR γ /d plus CD8aa would be needed to make that point. Please also consider TCR γ /d nIELs as source of the IFN γ .

We stained IFN- γ expressing CD4 $^+$ or CD8 $^+$ T cells for both TCR α / β $^+$ (CD45 $^+$ TCR β $^+$ CD3 ϵ $^+$) and non- α / β (CD45 $^+$ TCR β $^+$ CD3 ϵ $^+$) populations following the STAR protocol (Kim E, et al, STAR

Protoc, 2022). While we did not observe differences in the CD8⁺ population after low-dose penicillin treatment, we identified a significant difference in IFN- γ -producing CD4⁺ IELs. These analyses have been incorporated into our updated gating strategy (Figure S14).

13. L237/238: I agree with the comment that CD74 KO mice are more susceptible to DSS colitis, but the interpretation of the results reported in ref 43 on DSS exposed IEC-specific MHCII KO mice is difficult. No more weight loss and no shorter colon length reported, both typical markers of DSS colitis.

We thank the reviewer for the comment. The IEC-specific MHC-II knockout was generated using an *H2-Ab1* floxed allele with constitutive Villin-Cre expression from embryonic day E12.5, and indeed, there are conflicting results regarding DSS colitis in these mice (Jamwal DP, et al, Gastroenterology, 2020; Stephens WZ, et al, Cell Rep, 2021). Our study demonstrates that enhancer-mediated epigenetic regulation occurs after weaning and affects primarily inducible, rather than basal MHC-II expression, particularly in the colon. We cited these papers to provide a functional context. We believe that our mechanistic insights—such as the timing of microbiota-dependent epigenetic reprogramming and the role of memory responses—may help explain these previously conflicting observations.

14. Fig. 1B: Please include FACS analysis of Lgr5-GFP allele negative mice as control.

In all FACS sorting and downstream analyses, we used wild-type (without Lgr5-GFP knock-in) mice as negative controls for gating. This is shown in our gating strategy in Figure S14A, and we have stated this explicitly in the figure legend (Figure 1B) to ensure it is clear to readers.

15. Fig. 5C: The IFN γ effect is relatively weak (and the addition of the DAC reagent does not add any information). Have other cytokines been tested? What might also have a similar effect?

The relatively modest IFN- γ effect observed at 48 h in organoid cultures reflects its toxicity to ISCs (Figure 7C and Figure S10). To achieve complete demethylation at the stem cell level, we therefore applied a very low dose of IFN- γ over three passages (Figure 7E). We also tested TNF- α , sodium butyrate, and LPS; however, none altered methylation of MHC-II genes. In contrast, the hypomethylating agent DAC alone effectively demethylated the *Cd74* enhancer but did not induce gene expression. Interestingly, DAC and IFN- γ together synergistically upregulated transcription (Figure 7C). These observations promoted us to hypothesize that enhancer demethylation conveys an epigenetic form of transcriptional memory: it leaves basal expression unchanged but primes epithelial cells for faster and stronger responses to future stimuli. Indeed, we further demonstrated that microbiota-induced enhancer demethylation enables transcriptional memory in long-lived ISCs (Figure 7D-I).

16. Fig5D: Does not show that IFN γ acts via an epigenetic change. The difference could also be explained by prolonged IFN γ exposure.

17. Fig. 5D vs E: Isn't the 48h group in Fig. 5E very similar to the pretreated group in Fig.5D: But in 5E, the expression does not differ from the naïve (control) group. Please explain.

For both points, we agree with the reviewer that IFN- γ treatment for 48 hours results in heterogeneous demethylation that does not fully reflect ISC-level changes. Accordingly, we removed this group and now present data after three passages, which allows sufficient time for complete demethylation in ISCs. Our revised data clearly show that IFN- γ -primed organoids display a memory response to subsequent related or unrelated challenges (IFN- γ or TNF- α) (Figure 7D-F).

In addition to the experiments described above, we performed ChIP-qPCR analysis showing that IFN- γ recruits STAT3 and demethylase TET3 to the enhancer regions of *Cd74*, directly demonstrating the mechanistic link between IFN- γ signaling and epigenetic regulation (Figure 7B).

18. Fig. 6B: Please show complete 16S rRNA gene analysis. Lactobacilli, Clostridium (?), SFB and Bacteroides/Prevotella (also Fig. S7)) may not be representative for Gram+/- bacteria. Also, all taxa shown have very very low abundance in a substantial number of mice. A quantitative bacterial PCR would help to assess the effect of the LDP regimen.

We agree and now include longitudinal metagenomic analysis of fecal samples with or without low-dose penicillin (LDP) treatment to characterize the microbiomes and identify specific bacteria relevant to the observed epigenetic effects (Figure 6).

19. Fig. 6D: p values are only shown for the comparison between the SPF and LDP group. Is none of the LDP GF comparisons significantly different?

To avoid confusion, we now show only the comparison between mice with or without (w/o) LDP treatment (Figure 5B-F), removing GF data and SPF labels since all LDP-treated mice were housed under SPF conditions.

20. Fig. 6E: the MHCII staining is hard to interpret. The gate is placed in a very artificial way.

We reanalyzed the data using mean fluorescence intensity (MFI) to quantify MHC-II levels, which confirmed the original finding of reduced MHC-II expression in IECs under LDP treatment. Representative flow plots and the corresponding quantitative data are shown in Figure 5F.

21. Fig. 6F-H: I believe the authors imply that the effect of LDP on DSS susceptibility is MHCII dependent (based on the ref43). However, this is not formally shown. The enhanced susceptibility LPS could be due to other epigenetically modified genes or independent effects (e.g. of immune cell homing or IEC proliferation etc.). Please repeat the experiment with IEC MHCII KO mice to show that the difference is gone.

We thank the reviewer for the suggestion. However, using IEC-specific MHC-II knockout mice for LDP/DSS treatment would not directly address the role of DNA methylation. If the gene regulatory sequence is deleted, transcriptional changes driven by epigenetic modifications could not occur; therefore, the absence of differences could not be attributed to methylation.

22. The title is misleading. I don't think DSS colitis can formally be interpreted as adaptive immunity model.

We have revised the title to: "Weaning-driven gut microbiome reprograms intestinal stem cell epigenetics to establish epithelial-intrinsic immune memory". The title highlights our key finding that weaning-driven gut microbiota induce enhancer demethylation, priming long-lived ISCs to orchestrate immune-like transcriptional responses that link early-life microbial exposures to long-term mucosal immune function.

Minor points

1. L158: 15 week old mice are not young adult mice but adult mice.

We have made the changes as suggested.

2. The work from Dana Philpott (Tsang et al., Gut microbes, 2022) that describes expression differences between GF and SPF mice should be cited.

We have cited this work in the Introduction.

3. Fig. 6B. Please show quantitative data.

We now show quantitative data in Figure S7.

Reviewer #3 (Remarks to the Author):

The study entitled “Weaning-Driven Gut Microbiome Shapes Intestinal Stem Cell Epigenetics to Train Immunological Memory” by Yang et al, presents interesting findings on the potential role of the presence (or absence) of the gut microbiome in shaping intestinal stem cell (ISC) epigenetics during weaning. However, there are several areas where the study could be strengthened, both in terms of experimental design and data interpretation.

We thank the reviewer for recognizing the novelty of our findings on the role of the gut microbiome in shaping ISC epigenetics during weaning, and we appreciate the suggestions regarding experimental design and interpretation. In this revision, we have clarified our focus on MHC-II genes, expanded microbiome profiling, elucidated the IFN- γ -TET3-metabolite pathway, demonstrated the epigenetic regulatory role of the weaning-associated IFN- γ burst, and established the long-term functional consequences of this epigenetic memory. We believe these revisions address the reviewer’s concerns, and we respond to the specific comments in detail below.

Comments:

- The use of germ-free (GF) mice is appropriate for this study, but it is unclear if microbial products were also excluded (or tested to confirm they were not present), which could lead to activation of responses, and could occur even in the absence of live bacteria (this links to a comment on the organoid studies). The descriptive nature of the findings is significant, but the lack of further mechanistic exploration into what specific components of the microbiota are critical limits the depth of this study. Furthermore, there is no clear identification of which microbiota components (microbes/MAMPs/metabolites etc.) are driving the observed effects. It would be key to address whether microbial products might have been tested alongside live bacterial exposures to distinguish their individual impacts.

We agree that microbial products or components, even in the absence of live bacteria, can influence host responses. During weaning, microbiota colonization and diversification trigger an IFN- γ peak which, together with microbiota-derived metabolites, drives TET3-mediated enhancer demethylation in ISCs. This establishes a durable, immune-cell-like memory, enabling robust MHC-II induction upon subsequent challenges. Thus, our findings reveal how microbial signals give rise to stable epigenetic and transcriptional memory in innate epithelial cells.

- The study heavily relies on the presence or absence of microbiota, but it does not provide any detailed microbiota profiling data (only qPCR targeted a very limited number of ‘taxa’ and on a small number of samples). This reductionist approach limits the scope of the study and findings. Microbiota composition could significantly influence the epigenetic regulation of ISCs. Profiling at different stages of the experiment would provide insights into how specific microbial

populations contribute to the observed effects. Indeed, the article focuses largely on epigenetic changes, but it lacks a strong connection to microbial mechanisms.

While the epigenetic changes in response to weaning and microbiota are interesting, the lack of sequencing data limits the ability to link these changes to specific microbial species or community structures. A deeper microbiome analysis using shotgun metagenomics, and potentially metabolomics, would significantly add robustness to the conclusions.

We agree with the comments; thus, we have added microbiome analysis using shotgun metagenomics, which provides comprehensive profiling of microbial communities across different experimental stages. These data allowed us to identify specific bacterial taxa and their functional potential, which correlate with the observed epigenetic changes in ISCs. Specific results related to microbial colonization and antibiotic effects over time are explained after the next two comments.

- The authors claim that microbial exposure during early life impacts ISC epigenetics, but they do not explore how stable microbial colonization is achieved. What microbial species are colonizing, and how stable is the colonization over time (linked to point above about lack of detailed microbiome profiling)? Data on microbial stability or shifts would strengthen the paper's claims about long-term impacts.
- Moreover, while low-dose antibiotics were tested, the lack of detailed analysis on how these interventions affect later microbiota dynamics and subsequent epigenetic reprogramming limits the interpretation of the data. The authors focus on the 4-6-week window for epigenetic changes but fail to explore bacterial knockdown timelines or bacterial regrowth postintervention.

To determine the impact of the gut microbiome on epigenetic regulation, we analyzed fecal microbial composition longitudinally at 2, 4, 6, 8, 10, and 12 weeks of age in SPF mice with or without early-life low-dose penicillin (LDP) exposure. Mice exposed to LDP exhibited epigenetic dysregulation of MHC-II genes similar to that observed in GF mice, allowing us to link specific bacterial taxa and functional features to epigenetic alterations. Notably, several bacterial genera that normally expand during the post-weaning period (after 4 weeks) were persistently reduced in LDP-exposed mice into adulthood (10–12 weeks). These changes included a loss of SCFA-producing species and a decline in microbial functions that support TET-mediated demethylation, including α -KG production, iron availability, and DNMT inhibition. Our results are summarized in Figure 6.

- The authors suggest that juvenile gut microbiota transfer (GMT) is more effective in restoring epigenetic regulation, but they do not provide sufficient data on how the timing of microbial introduction affects colonization or stability. Further data on the longitudinal stability of microbiota following transfer would help elucidate whether the effects are transient or lasting.

We have addressed this concern in our response to Reviewer 1 (point 3) through microbiome profiling across groups and extended analysis of experiments in adult transplants.

- In the GF organoid experiments, the authors mention that LPS did not induce responses. However, they do not test other bacterial ligands and metabolites that might have elicited a different response, such as peptidoglycan, other types of LPS like Bacteroides-derived LPS, and SCFAs, which may have different effects.

In addition to IFN- γ , we evaluated the effects of LPS, TNF- α , and sodium butyrate on the methylation of enhancers of MHC-II genes in GF organoids. Individually, none of these factors had an effect, indicating that IFN- γ is the primary regulator in this context.

- The use of the term “dysbiosis” is over-simplified in this context. The authors should define the term more rigorously, particularly when referencing such complex microbiota-host interactions.

We agree and have replaced the term ‘dysbiosis’ in the context of LDP modeling with a precise description of changes in specific Gram-positive bacterial populations.

- Several studies have demonstrated impacts of the microbiota, including during pre- and postweaning within the murine gastrointestinal track with response to epithelial responses and these studies are currently not referenced. The authors should cite these works to provide context and acknowledge existing literature.

We appreciate the suggestion and have cited relevant prior studies in the Introduction and Discussion.

- As highlighted above - introducing specific bacterial strains or microbial consortia into GF mice could help identify key microbial players and confirm the protective or detrimental effects observed. Additionally, using an antibiotic cocktail to induce near GF conditions (would also represent a more ‘normally-programmed’ host than GF mice) and then reintroducing specific bacteria could provide more detailed insights into which microbes are crucial for ISC epigenetic regulation. This could be coupled with in-depth sequencing (and metabolomics) on microbial populations during the key stages of intervention, as this would reveal microbial dynamics and potential drivers of the observed epigenetic changes.

This study provides valuable data on the microbiome's role in shaping ISC epigenetics, particularly during the weaning period. However, the lack of microbiota profiling, mechanistic detail, and consideration of microbial products weakens the conclusions. Incorporating microbial profiling and addressing the concerns mentioned above would significantly strengthen the manuscript.

In summary, we have incorporated detailed microbiome profiling at both the taxonomic and functional levels into our experimental design and data analyses (Figure 3E, Figure S6, Figure 5B,C, and Figure 6). These additional analyses not only corroborate our original findings but also provide new insights into the role of specific bacterial taxa and microbial functions in influencing ISC epigenetics, particularly during the critical weaning period. We hope these enhanced microbiome analyses, conducted in response to the reviewer's suggestion, fully address the concerns.

We are delighted by the favorable responses from the Reviewers. We have addressed the one concern raised by Reviewer 3 (see below).

Reviewers Comments:

Reviewer #1 (Remarks to the Author):

The authors have addressed my concerns and I thank them for their work.

We thank the reviewer for the positive evaluation of our revised manuscript.

Reviewer #2 (Remarks to the Author):

The authors have adequately and in great detail responded to the reviewers questions and performed substantial new experiments (e.g. microbiota metagenomics, IFN γ inhibition in vivo). The revised manuscript presents a comprehensive study on the effect of the weaning microbiota induced IFN γ on the epigenetic control of the inducible MHCII gene expression with long-term consequences on disease susceptibility. With the changes made, I think the manuscript should be accepted for publication.

We appreciate the constructive input throughout the review process and are pleased that our study is now considered suitable for publication in Nature Microbiology.

Reviewer #3 (Remarks to the Author):

Thank you to the authors for the substantial revisions and for incorporating detailed microbiome analyses. These additions go a long way toward addressing the concerns raised in the previous submission. However, the microbiome component of the study remains largely descriptive, and the new analyses, while informative, still rely primarily on correlations between shotgun metagenomic profiles and GO-term-based functional predictions. This does not provide a direct mechanistic link or in vivo evidence demonstrating microbial causality.

We agree that the microbiome analyses are primarily based on functional inference from shotgun metagenomic data rather than direct measurements of microbial metabolites. We have clarified this limitation in the Discussion (line 355-359) and note that: In this study, metagenomic GO- and pathway-based analyses were used to identify microbiome-associated functional signatures that align with known epigenetic regulatory processes. Future studies incorporating metabolomics, targeted manipulation of specific bacterial species, and dietary modulation will be required to establish microbial causality and define the drivers of these long-lasting epigenetic programs.